# Variance-Reduced Gradient Estimation via Noise-Reuse in Online Evolution Strategies

**Oscar Li**[§1], **James Harrison**[◇], **Jascha Sohl-Dickstein**[◇], **Virginia Smith**[§], **Luke Metz**[◇2]
[§]Machine Learning Department, School of Computer Science    Carnegie Mellon University
[◇]Google DeepMind

## Abstract

Unrolled computation graphs are prevalent throughout machine learning but present challenges to automatic differentiation (AD) gradient estimation methods when their loss functions exhibit extreme local sensitivtiy, discontinuity, or blackbox characteristics. In such scenarios, online evolution strategies methods are a more capable alternative, while being more parallelizable than vanilla evolution strategies (ES) by interleaving partial unrolls and gradient updates. In this work, we propose a general class of unbiased online evolution strategies methods. We analytically and empirically characterize the variance of this class of gradient estimators and identify the one with the least variance, which we term Noise-Reuse Evolution Strategies (NRES). Experimentally[3], we show NRES results in faster convergence than existing AD and ES methods in terms of wall-clock time and number of unroll steps across a variety of applications, including learning dynamical systems, meta-training learned optimizers, and reinforcement learning.

## 1 Introduction

First-order optimization methods are a foundational tool in machine learning. With many such methods (e.g., SGD, Adam) available in existing software, ML training often amounts to specifying a computation graph of learnable parameters and computing some notion of gradients to pass into an off-the-shelf optimizer. Here, *unrolled computation graphs* (UCGs), where the same learnable parameters are repeatedly applied to transition a dynamical system's inner state, have found their use in various applications such as recurrent neural networks [1, 2], meta-training learned optimizers [3, 4], hyperpameter tuning [5, 6], dataset distillation [7, 8], and reinforcement learning [9, 10].

While a large number of automatic differentiation (AD) methods exist to estimate gradients in UCGs [11], they often perform poorly over loss landscapes with extreme local sensitivity and cannot handle black-box computation dynamics or discontinuous losses [12, 3, 13]. To handle these shortcomings, evolution strategies (ES) have become a popular alternative to produce gradient estimates in UCGs [14]. ES methods convolve the (potentially pathological or discontinuous) loss surface with a Gaussian distribution in the learnable parameter space, making it smoother and infinitely differentiable. Unfortunately, vanilla ES methods cannot be applied online[4] — the computation must reach the end of the graph to produce a gradient update, thus incurring large update latency for long UCGs. To address this, a recently proposed approach, Persistent Evolution Strategies [15] (PES), samples a new Gaussian noise in every truncation unroll and accumulates the past sampled noises to get rid of the estimation bias in its online application.

In this work, we investigate the coupling of the noise sampling frequency and the gradient estimation frequency in PES. By decoupling these two values, we arrive at a more general class of unbiased,

---

[1]Correspondence to: <oscarli@cmu.edu>. [2]Now at OpenAI.

[3]Code available at https://github.com/OscarcarLi/Noise-Reuse-Evolution-Strategies.

[4]*Online* here means a method can produce gradient estimates using only *a truncation window* of an unrolled computation graph *instead of the full graph*, thus allowing the interleaving of partial unrolls and gradient updates.

37th Conference on Neural Information Processing Systems (NeurIPS 2023).

**(a)**

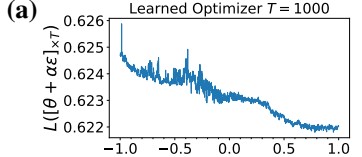

**(b)**

| method | online? | unbiased? | low variance? |
|---|---|---|---|
| FullES | ✗ | ✓ | ✓ |
| TES [3] | ✓ | ✗ | ✓ |
| PES [15] | ✓ | ✓ | ✗ |
| NRES (ours) | ✓ | ✓ | ✓ |

**Figure 1:** (a) The pathological loss surface in the learned optimizer task (Sec. 5.2) along a random $\epsilon$ direction; such surfaces are common in UCGs but can make automatic differentiation methods unusable, leading to the recent development of evolution strategies methods. (b) Comparison of properties of different evolution strategies methods. Unlike prior online ES methods, NRES produces both unbiased and low-variance gradient estimates.

online ES gradient estimators. Through a variance characterization of these estimators, we find that the one which provably has the lowest variance in fact reuses the same noise for the entire time horizon (instead of over a single truncation window as in PES). We name this method *Noise-Reuse Evolution Strategies* (NRES). In addition to being simple to implement, NRES converges faster than PES across a wide variety of applications due to its reduced variance. Overall, we make the following contributions:

- We propose a class of unbiased online evolution strategies gradient estimators for unrolled computation graphs that generalize Persistent Evolution Strategies [15].
- We analytically and empirically characterize the variance of this class of estimators and identify the lowest-variance estimator which we name Noise-Reuse Evolution Strategies (NRES).
- We identify the connection between NRES and the existing offline ES method FullES and show that NRES is a better alternative to FullES both in terms of parallelizability and variance.
- We demonstrate that NRES can provide optimization convergence speedups (up to 5-60×) over AD/ES baselines in terms of wall-clock time and number of unroll steps in applications of 1) learning dynamical systems, 2) meta-training learned optimizers, and 3) reinforcement learning.

## 2  Online Evolution Strategies: Background and Related Work

**Problem setup.** Unrolled computation graphs (UCGs) [15] are common in applications such as training recurrent neural networks, meta-training learned optimizers, and learning reinforcement learning policies, where the same set of parameters are repeatedly used to update the inner state of some system. We consider general UCGs where the inner state $s_t \in \mathbb{R}^p$ is updated with learnable parameters $\theta \in \mathbb{R}^d$ through transition functions: $\{f_t : \mathbb{R}^p \times \mathbb{R}^d \to \mathbb{R}^p\}_{t=1}^T$, $s_t = f_t(s_{t-1}; \theta)$ for $T$ time steps starting from an initial state $s_0$. At each time step $t \in \{1, \ldots, T\}$, the state $s_t$ incurs a loss $L_t^s(s_t)$. As the loss $L_t^s$ depends on $t$ applications of $\theta$, we make this dependence more explicit with a loss function $L_t : \mathbb{R}^{dt} \to \mathbb{R}$ and $L_t([\theta]_{\times t}) := L_t^s(s_t)^5$. We aim to minimize the average loss over all $T$ time steps unrolled under the same $\theta$, $\min_\theta L([\theta]_{\times T})$, where

$$L(\theta_1, \ldots, \theta_T) := \frac{1}{T} \sum_{t=1}^T L_t(\theta_1, \ldots, \theta_t). \tag{1}$$

**Loss properties.** Despite the existence of many automatic differentiation (AD) techniques to estimate gradients in UCGs [11], there are common scenarios where they are undesirable: **1)** *Loss surfaces with extreme local sensitiviy*: With large number of unrolls in UCGs, the induced loss surface is prone to high degrees of sharpness and many suboptimal local minima (see Figure 1). This issue is particularly prevalent when the underlying dynamical system is chaotic under the parameter ($\theta$) of interest [13], e.g. in model-based control [12] and meta-learning [3]. In such cases, naively following the gradient may either *a)* fail to converge under the normal range of learning rates (because of the conflicting gradient directions) or *b)* converge to highly suboptimal solutions using a tuned, yet much smaller learning rate. **2)** *Black-box or discontinuous losses*: As AD methods require defining a Jacobian-vector product (forward-mode) or a vector-Jacobian product (reverse-mode) for every elementary operation in the computation graph, they cannot be applied when the UCG's inner dynamics are inaccessible (e.g., model-free reinforcement learning) or the loss objectives (e.g. accuracy) are piecewise constant (zero gradients).

**Evolution Strategies.** Due to the issues with AD methods described above, a common alternative is to use evolution strategies (ES) to estimate gradients. Here, the original loss function is convolved with an isotropic Gaussian distribution in the space of $\theta$, resulting in an infinitely differentiable loss

---

$^5[\theta]_{\times a}$, $a \in \mathbb{Z}_{\geq 0}$ denotes $a$ copies of $\theta$, for example: $L_t([\theta]_{\times a}, [\theta']_{\times t-a}) := L_t(\underbrace{\theta, \ldots, \theta}_{a \text{ times}}, \underbrace{\theta', \ldots, \theta'}_{(t-a) \text{ times}})$.

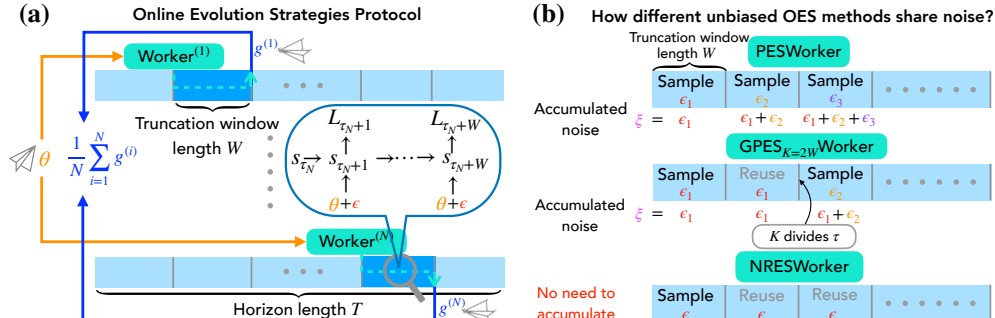

**Figure 2:** (a) Illustration of *step-unlocked* online ES workers working independently at different truncation windows. Here a central server sends $\theta$ (whose gradient to be estimated) to each worker and receives the estimates over partial unrolls from each. The averaged gradient can then be used in a first-order optimization algorithm. (b) Comparison of the noise sharing mechanisms of PES, GPES$_K$, and NRES (ours). Unlike PES (and GPES$_{K \neq T}$) which samples a new noise in every (some) truncation window and needs to accumulate the noise, NRES only samples noise once at the beginning of an episode and reuses the noise for the full episode.

function with lower sharpness and fewer local minima than before ($\sigma > 0$ is a hyperparameter):

$$\theta \mapsto \mathbb{E}_{\boldsymbol{\epsilon} \sim \mathcal{N}(\mathbf{0}, \sigma^2 I_{d \times d})} L([\theta + \boldsymbol{\epsilon}]_{\times T}). \tag{2}$$

An unbiased gradient estimator of (2) is given by the likelihood ratio gradient estimator [16]: $\frac{1}{\sigma^2} L([\theta + \boldsymbol{\epsilon}]_{\times T}) \boldsymbol{\epsilon}$. This estimator only requires the loss evaluation (hence is zeroth-order) but not an explicit computation of the gradient, thus being applicable in cases when the gradients are noninformative (chaotic or piecewise constant loss) or not directly computable (black-box loss). To reduce the variance, antithetic sampling is used and we call this estimator FullES (Algorithm 3 in the Appendix):

$$\text{FullES}(\theta) := \frac{1}{2\sigma^2} \left[ \frac{1}{T} \sum_{i=1}^{T} (L_i([\theta + \boldsymbol{\epsilon}]_{\times i}) - L_i([\theta - \boldsymbol{\epsilon}]_{\times i})) \right] \boldsymbol{\epsilon}. \tag{3}$$

The term Full highlights that this estimator can only produce a gradient estimate after a *full* sequential unroll from $t = 0$ to $T$. We call such a full unroll an *episode* following the reinforcement learning terminology. FullES can be parallelized [14] by averaging $N$ parallel gradient estimates using *i.i.d.* $\boldsymbol{\epsilon}$'s, but is not online and can result in substantial latency between gradient updates when $T$ is large.

**Truncated Evolution Strategies.** To make FullES online, Metz et al. [3] take inspiration from truncated backpropagation through time (TBPTT) and propose the algorithm TES (see Algorithm 6 in the Appendix). Unlike the stateless estimator FullES, TES is stateful: TES starts from a saved state $s$ (from the previous iteration) and draws a new $\boldsymbol{\epsilon}_i$ for antithetic unrolling. To make itself online, TES only unrolls for a truncation window of $W$ steps for every gradient estimate, thus reducing the latency from $O(T)$ to $O(W)$. Analytically,

$$\text{TES}(\theta) := \frac{1}{2\sigma^2 W} \sum_{i=1}^{W} \left[ L_{\tau+i}([\theta]_{\times \tau}, [\theta + \boldsymbol{\epsilon}_{(\tau/W)+1}]_{\times i}) - L_{\tau+i}([\theta]_{\times \tau}, [\theta - \boldsymbol{\epsilon}_{(\tau/W)+1}]_{\times i}) \right] \boldsymbol{\epsilon}_{(\tau/W)+1}. \tag{4}$$

Here, besides the Gaussian random variables $\boldsymbol{\epsilon}_i \big|_{i=1}^{T/W} \overset{\text{iid}}{\sim} \mathcal{N}(\mathbf{0}, \sigma^2 I_{d \times d})$, the time step $\boldsymbol{\tau}$ which TES starts from is also a random variable drawn from the uniform distribution $\boldsymbol{\tau} \sim \text{Unif}\{0, W, \ldots, T - W\}$[6]. It is worth noting that Vicol et al. [15] who also analyze online ES estimators do not take the view that the time step $\boldsymbol{\tau}$ an online ES estimator starts from is a random variable; as such their analyses do not fully reflect the "online" nature captured in our work. When multiple online ES Workers (e.g., TES workers) run in parallel, different workers will work at different *i.i.d.* time steps $\boldsymbol{\tau}_i$ (which we call *step-unlocked* workers) (see Figure 2(a)). We provide the pseudocode for creating step-unlocked workers and for general online ES learning in Algorithm 4 and 5 in the Appendix.

**TES is a biased gradient estimator of** (2)**.** Note that in (4), only the $\theta$'s in the current length-$W$ truncation window receive antithetic perturbations, thus ignoring the impact of the earlier $\theta$'s up to time step $\boldsymbol{\tau}$. Due to this bias, optimization using TES typically doesn't converge to optimal solutions.

**Persistent Evolution Strategies.** To resolve the bias of TES, Vicol et al. [15] recognize that TES samples a new noise in every truncation window and modifies the smoothing objective into:

$$\theta \mapsto \mathbb{E}_{\{\boldsymbol{\epsilon}_i\}} L([\theta + \boldsymbol{\epsilon}_1]_{\times W}, \ldots, [\theta + \boldsymbol{\epsilon}_{T/W}]_{\times W}). \tag{5}$$

---

[6]The random variable $\boldsymbol{\tau}$ will always be sampled from this uniform distribution. In addition, we will assume the time horizon $T$ can be evenly divided into truncation windows of length $W$, i.e. $T \equiv 0 \pmod{W}$.

| **Algorithm 1** Persistent Evolution Strategies [15] | **Algorithm 2** Noise-Reuse Evolution Strategies (ours) |
|---|---|

```
class PESWorker(OnlineESWorker):
    def __init__(self, W):
        self.τ = 0;  self.s⁺ = s₀;  self.s⁻ = s₀
        self.W = W;  self.ξ = 0 ∈ ℝᵈ

    def gradient_estimate(self, θ):
        # sample at every truncation window
        ε ~ 𝒩(0, σ²I_{d×d})  # this is ε_(τ/W)+1
        self.ξ += ε   # now self.ξ = Σ_{i=1}^{τ/W} εᵢ + ε_{τ/W+1}

        (s⁺, s⁻) = (self.s⁺, self.s⁻)
        L⁺_sum = 0;   L⁻_sum = 0
        for i in range(1, self.W+1):
            s⁺ = f_{self.τ+i}(s⁺, θ + ε)
            s⁻ = f_{self.τ+i}(s⁻, θ − ε)
            L⁺_sum += L^s_{self.τ+i}(s⁺)
            L⁻_sum += L^s_{self.τ+i}(s⁻)

        g = (L⁺_sum − L⁻_sum)/(2σ² · self.W) · self.ξ
        self.s⁺ = s⁺; self.s⁻ = s⁻
        self.τ = self.τ + W
        if self.τ ≥ T:  # reset at the end
            self.τ = 0; self.s⁺ = s₀; self.s⁻ = s₀
            self.ξ = 0
        return g
```

```
class NRESWorker(OnlineESWorker):
    def __init__(self, W):
        self.τ = 0;  self.s⁺ = s₀;  self.s⁻ = s₀
        self.W = W

    def gradient_estimate(self, θ):
        if self.τ == 0: # only sample at beginning
            ε ~ 𝒩(0, σ²I_{d×d})
            self.ε = ε        # reuse for this episode

        (s⁺, s⁻) = (self.s⁺, self.s⁻)
        L⁺_sum = 0;   L⁻_sum = 0
        for i in range(1, self.W+1):
            s⁺ = f_{self.τ+i}(s⁺, θ + self.ε)
            s⁻ = f_{self.τ+i}(s⁻, θ − self.ε)
            L⁺_sum += L^s_{self.τ+i}(s⁺)
            L⁻_sum += L^s_{self.τ+i}(s⁻)

        g = (L⁺_sum − L⁻_sum)/(2σ² · self.W) · self.ε
        self.s⁺ = s⁺; self.s⁻ = s⁻
        self.τ = self.τ + W
        if self.τ ≥ T:  # reset at the end
            self.τ = 0; self.s⁺ = s₀; self.s⁻ = s₀

        return g
```

They show that an unbiased gradient estimator of 5 is given by (see Algorithm 1):

$$\text{PES}(\theta) := \frac{1}{2\sigma^2 W} \sum_{i=1}^{W} \left[ L_{\tau+i}([\theta + \epsilon_1]_{\times W}, \dots, [\theta + \epsilon_{(\tau/W)+1}]_{\times i}) \right.$$
$$\left. - L_{\tau+i}([\theta - \epsilon_1]_{\times W}, \dots, [\theta - \epsilon_{(\tau/W)+1}]_{\times i}) \right] \left( \left(\sum_{j=1}^{\tau/W} \epsilon_j\right) + \epsilon_{\tau/W+1} \right), \quad (6)$$

with randomness in both $\{\epsilon_i\}$ and $\tau$. To eliminate the bias of TES, instead of multiplying only with the current epsilon $\epsilon_{\tau/W+1}$, PES multiplies it with the cumulative sum of all the different *iid* noise sampled so far (self.$\xi$ in PESWorker). As we shall see in the next section, this accumulation of noise terms provably results in higher variance, making PES less desirable in practice. We contrast the noise sampling properties of our proposed methods with PES in Figure 1(b).

**Hysteresis.** When online gradient estimators are used in training, they often suffer from *hysteresis*, or history dependence, as a result of the parameters $\theta$ changing between adjacent unrolls. That is, the parameter value $\theta_0$ that a worker uses in the current truncation window is not the same parameter $\theta_{-1}$ that was used in the previous window. This effect is often neglected [15], under an assumption that $\theta$ is updated slowly. To the best of our knowledge, [17] is the only work to analyze the convergence of an online gradient estimator under hysteresis. In the following theoretical analysis, we assume all online gradient estimates are computed without hysteresis in order to isolate the problem. However, in Section 5, we show empirically that even under the impact of hysteresis, our proposed online estimator NRES can outperform non-online methods (e.g., FullES) which don't suffer from hysteresis.

## 3 A New Class of Unbiased Online Evolution Strategies Methods

As shown in Section 2, TES and PES both sample a new noise perturbation $\epsilon$ for every truncation window to produce gradient estimates. Here we note that the *frequency of noise-sharing* (new noise every truncation window of size $W$) is fixed to the *frequency of gradient estimates* (a gradient estimate every truncation window of size $W$). However, the former is a choice of the smoothing objective (5), while the latter is often a choice of how much gradient update latency the user can tolerate. In this section *we break this coupling* by introducing a general class of gradient estimators that encompass PES. We then analyze these estimators' variance to identify the one with the least variance.

**Generalized Persistent Evolution Strategies (GPES).** For a given fixed truncation window size $W$, we consider all *noise-sharing periods* $K$ that are multiples of $W$, $K = cW$ for $c \in \mathbb{Z}^+$, $c \leq T/W$. $K$ being a multiple of $W$ ensures that within each truncation window, only a single $\epsilon$ is used. When

$K = W$, we recover the PES algorithm. However, when $K$ is larger than $W$, the same noise will be used across adjacent truncation windows (Figure 2(b)). With a new noise sampled every $K$ unroll steps, we define the $K$-*smoothed loss objective* as the function:

$$\theta \mapsto \mathbb{E}_{\{\boldsymbol{\epsilon}_i\}} L([\theta + \boldsymbol{\epsilon}_1]_{\times K}, \ldots, [\theta + \boldsymbol{\epsilon}_{\lceil T/K\rceil}]_{\times \mathrm{r}(T,K)}),^7 \tag{7}$$

where $\mathrm{r} : (\mathbb{Z}^+)^2 \to \mathbb{Z}^+$ is the modified remainder function such that $\mathrm{r}(x, y)$ is the unique integer $n \in [1, y]$ where $x = qy + n$ for some integer $q$. This extra notation allows for the possibility that $T$ is not divisible by $K$ and the last noise $\boldsymbol{\epsilon}_{\lceil T/K\rceil}$ is used for only $\mathrm{r}(T, K) < K$ steps.

We now give the analytic form of an unbiased gradient estimator of the resulting smoothed loss.[8]

**Lemma 1.** *An unbiased gradient estimator for the $K$-smoothed loss is given by*

$$\mathrm{GPES}_K(\theta) \coloneqq \frac{1}{2\sigma^2 W} \sum_{j=1}^{W} \Bigg[ L_{\boldsymbol{\tau}+j}([\theta + \boldsymbol{\epsilon}_1]_{\times K}, \ldots, [\theta + \boldsymbol{\epsilon}_{\lfloor \boldsymbol{\tau}/K\rfloor + 1}]_{\times \mathrm{r}(\boldsymbol{\tau}+j,K)})$$
$$- L_{\boldsymbol{\tau}+j}([\theta - \boldsymbol{\epsilon}_1]_{\times K}, \ldots, [\theta - \boldsymbol{\epsilon}_{\lfloor \boldsymbol{\tau}/K\rfloor + 1}]_{\times \mathrm{r}(\boldsymbol{\tau}+j,K)}) \Bigg] \cdot \left( \sum_{i=1}^{\lfloor \boldsymbol{\tau}/K\rfloor + 1} \boldsymbol{\epsilon}_i \right),$$

*with randomness in $\boldsymbol{\tau}$ and $\{\boldsymbol{\epsilon}_i\}_{i=1}^{\lceil T/K\rceil}$.*

**GPES$_K$ algorithm.** Here, for the truncation window starting at step $\boldsymbol{\tau}$, the noise $\boldsymbol{\epsilon}_{\lfloor \boldsymbol{\tau}/K\rfloor + 1}$ is used as the antithetic perturbation to unroll the system. If $\boldsymbol{\tau}$ is not divisible by $K$, then this noise has already been sampled at time step $t = \lfloor \boldsymbol{\tau}/K\rfloor \cdot K$ in an earlier truncation window. Therefore, to know what noise to apply at this truncation window, we need to remember the last used $\boldsymbol{\epsilon}$ and update it when $\boldsymbol{\tau}$ becomes divisble by $K$. We provide the algorithm for the $\mathrm{GPES}_K$ gradient estimator in Algorithm 7 in the Appendix. Note that $\mathrm{GPES}_{K=W}$ is the same as the PES algorithm.

**Variance Characterization of GPES$_K$.** With this generalized class of gradient estimators $\mathrm{GPES}_K$, one might wonder how to choose the value of $K$. Since each estimator is an unbiased gradient estiamtor with respect to its smoothed objective, we compare the variance of these estimators as a function of $K$. To do this analytically, we make some simplifying assumptions:

**Assumption 2.** For a given $\theta \in \mathbb{R}^d$ and $t \in [T] \coloneqq \{1, \ldots, T\}$, there exists a set of vectors $\{g_i^t \in \mathbb{R}^d\}_{i=1}^{t}$, such that for any $\{v_i \in \mathbb{R}^d\}_{i=1}^{t}$, the following equality holds:

$$L_t(\theta + v_1, \theta + v_2, \ldots, \theta + v_t) - L_t(\theta - v_1, \theta - v_2, \ldots, \theta - v_t) = 2 \sum_{i=1}^{t} (v_i)^\top (g_i^t) \tag{8}$$

**Remark 3.** This assumption is more general than the quadratic $L_t$ assumption made in [15] (explanation see Appendix D). Here one can roughly understand $g_i^t$ as time step $t$'s smoothed loss's partial derivative with respect to the $i$-th application of $\theta$. For notational convenience, we let $g^t \coloneqq \sum_{i=1}^{t} g_i^t$ (roughly the total derivative of smoothed step-$t$ loss with respect to $\theta$) and $g_{K,j}^t \coloneqq \sum_{i=K\cdot(j-1)+1}^{\min\{t,\, K\cdot j\}} g_i^t$ for $j \in \{1, \ldots, \lceil t/K\rceil\}$ (roughly the sum of partial derivatives of smoothed step-$t$ loss with respect to all $\theta$'s in the $j$-th noise-sharing window of size $K$ (the last window might be shorter)).

With this assumption in place, we first consider the case when $W = 1$ and $K = cW = c$ for $c \in [T]$. In this case, the $\mathrm{GPES}_K$ estimator can be simplified into the following form:

**Lemma 4.** *Under Assumption 2, when $W = 1$, $\mathrm{GPES}_{K=c}(\theta) = \frac{1}{\sigma^2} \sum_{j=1}^{\lfloor \boldsymbol{\tau}/c\rfloor + 1} \left( \sum_{i=1}^{\lfloor \boldsymbol{\tau}/c\rfloor + 1} \boldsymbol{\epsilon}_i \right) \boldsymbol{\epsilon}_j^\top g_{c,j}^{\boldsymbol{\tau}+1}.$*

With this simplified form, we can now characterize the variance of the estimator $\mathrm{GPES}_{K=c}(\theta)$. Since it's a random vector, we analytically derive its total variance (trace of covariance matrix) $\mathrm{tr}(\mathrm{Cov}[\mathrm{GPES}_{K=c}(\theta)])$.

**Theorem 5.** *When $W = 1$ and under Assumption 2, for integer $c \in [T]$,*

$$\mathrm{tr}(\mathrm{Cov}[\mathrm{GPES}_{K=c}(\theta)]) = \frac{(d+2)}{T} \sum_{t=1}^{T} (\|g^t\|_2^2) - \left\| \frac{1}{T} \sum_{t=1}^{T} g^t \right\|_2^2 + \frac{1}{T} \sum_{t=1}^{T} \left( \frac{d}{2} \sum_{j=1, j'=1}^{\lceil t/c\rceil} \|g_{c,j}^t - g_{c,j'}^t\|_2^2 \right). \tag{9}$$

---

[7] $\lceil x \rceil$ is smallest integer $\geq x$; $\lfloor x \rfloor$ is the largest integer $\leq x$.

[8] Proofs for all the Lemmas and Theorems are in Appendix D.

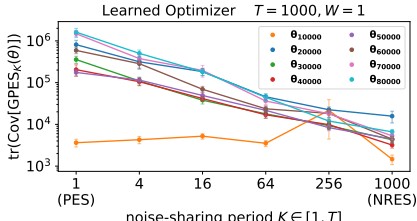

**Figure 3:** Total variance of $\mathrm{GPES}_K$ vs. noise-sharing period $K$ for different $\theta_i$'s from the learned trajectory of PES. $\mathrm{GPES}_{K=T}$ (NRES) has the lowest total variance among estimators of its class (including PES) for each $\theta_i$.

To understand how the value of $K = c$ changes the total variance, we notice that only the nonnegative third term in (9) depends on it. This term measures the pairwise squared distance between non-overlapping partial sums $g_{c,j}^t$ for all $j$. When $c = T$, for every $t \in [T]$, there is only a single such partial sum as $\lceil t/c \rceil = 1$. In this case, this third term reduces to its smallest value of 0. Thus:

**Corollary 6.** *Under Assumption 2, when $W = 1$, the gradient estimator $\mathrm{GPES}_{K=T}(\theta)$ has the smallest total variance among all $\{\mathrm{GPES}_K : K \in [T]\}$ estimators.*

**Remark 7.** To understand Corollary 6 intuitively, notice that at a given time step $t$ (i.e., a length-1 truncation window), any $\mathrm{GPES}_{K=c}$ gradient estimator ($c \in [T]$) aims to unbiasedly estimate the total derivative of the smoothed loss at this step with respect to $\theta$, which we have denoted by $g^t$. By applying a new Gaussian noise perturbation every $c < T$ steps, the $\mathrm{GPES}_{K=c}$ estimators *indirectly estimate* $g^t$ by first unbiasedly estimating the gradients inside each size-$c$ noise-sharing window: $\{g_{c,j}^t\}_{j=1}^{\lceil t/c \rceil}$ and then summing up the result (notice $g^t = \sum_{j=1}^{\lceil t/c \rceil} g_{c,j}^t$). To obtain this extra (yet unused) information about the intermediate partial derivatives, these estimators require more randomness and thus suffer from a larger total variance than the $\mathrm{GPES}_{K=T}$ estimator which directly estimates $g^t$.

**Experimental Verification of Corollary 6.** We empirically verify Corollary 6 on a meta-training learned optimizer task ($T = 1000$; see additional details in Section 5.2). Here we save a trajectory of $\theta_i$ learned by PES ($i$ denotes training iteration) and compute the total variance of the estimated gradients (without hysteresis) by GPES with different values of $K$ in Figure 3(a) ($W = 1$). In agreement with theory, $K = T$ has the lowest variance.

## 4 Noise-Reuse Evolution Strategies

**NRES has lower variance than PES.** As variance reduction is desirable in stochastic optimization [18], by Corollary 6, the gradient estimator $\mathrm{GPES}_{K=T}$ is particularly attractive and *can serve as a variance-reduced replacement for* PES. When $K = T$, we only need to sample a single $\epsilon$ once at the beginning of an episode (when $\tau = 0$) and reuse the same noise for the entirety of that episode before it resets. This removes the need to keep track of the cumulative applied noise ($\xi$) (Figure 2(b)), making the algorithm simpler and more memory efficient than PES. Due to its noise-reuse property, we name this gradient estimator $\mathrm{GPES}_{K=T}$ the *Noise-Reuse Evolution Strategies* (NRES) (pseudocode in Algorithm 2). Concurrent with our work, Vicol [19] independently proposes a similar algorithm with different analyses. We discuss in detail how our work differs from [19] in Appendix B. Despite Theorem 5 assuming $W = 1$, one can relax this assumption to any $W$ that divides the horizon length $T$. By defining a "mega" UCG whose single transition step is equivalent to $W$ steps in the original UCG, we can apply Corollary 6 to this mega UCG and arrive at the following result.

**Corollary 8.** *Under Assumption 2, when $W$ divides $T$, the NRES gradient estimator has the smallest total variance among all $\mathrm{GPES}_{K=cW}$ estimators $c \in [T/W]$.*

**NRES is a replacement for FullES.** By sharing the same noise over the entire horizon, the smoothing objective of NRES is the same as FullES's. Thus, we can think of NRES as the online counterpart to the offline algorithm FullES. Hence NRES can act as a drop-in replacement to FullES in UCGs. A single FullES worker runs $2T$ unroll steps for each gradient estimate, while a single NRES runs only $2W$ steps. Motivated by this, we compare the average of $T/W$ *i.i.d.* NRES gradient estimates with 1 FullES gradient estimate as they require the same amount of compute.

**NRES is more parallelizable than FullES.** Because the $T/W$ NRES gradient estimators are independent of each other, we can run them in parallel. Under perfect parallelization, the entire NRES gradient estimation would require $O(W)$ time to complete. In contrast, the single FullES gradient estimate has to traverse the UCG from start to finish, thus requiring $O(T)$ time. Hence, NRES is $T/W$ times more parallelizable than FullES under the same compute budget (Figure 4(a)).

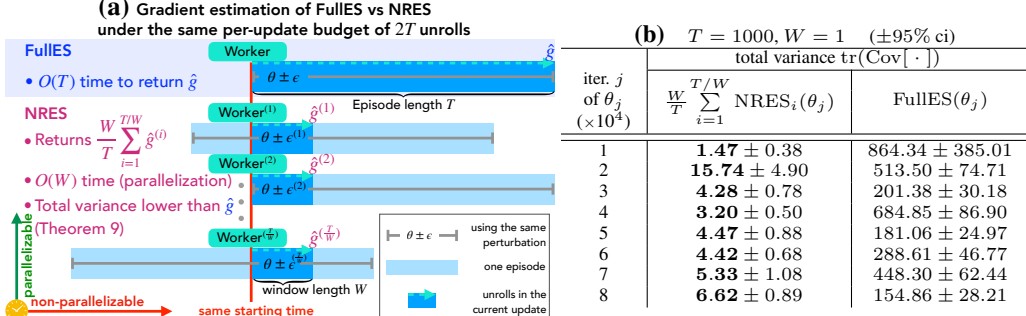

**Figure 4:** (a) Comparison of FullES and NRES gradient estimation under the same unroll budget. Unlike FullES which can only use a single noise perturbation $\epsilon$ to unroll sequentially for an entire episode of length $T$, NRES can use $T/W$ parallel step-unlocked workers each unrolling inside its random truncation windows of length $W$ with independent perturbations $\epsilon^{(i)}$. This results in a $T/W\times$ speed-up and variance reduction (Theorem 9) over FullES. (b) The total variance of NRES and FullES estimators under the same compute budget at the same set of $\theta_i$ checkpoints in Figure 3(a). NRES achieves significantly lower total covariance.

**NRES can often have lower variance than FullES.** We next compare the variance of the average of $T/W$ *i.i.d.* NRES gradient estimates with the variance of 1 FullES gradient estimate:

**Theorem 9.** *Under Assumption 2, for any $W$ that divides $T$, if*

$$\sum_{k=1}^{T/W}\left\|\sum_{t=W\cdot(k-1)+1}^{W\cdot k}g^t\right\|_2^2 \leq \frac{d+1}{d+2}\left\|\sum_{j=1}^{T/W}\sum_{t=W\cdot(k-1)+1}^{W\cdot k}g^t\right\|_2^2, \tag{10}$$

*then for iid $\{\mathrm{NRES}_i(\theta)\}_{i=1}^{T/W}$ estimators,* $\quad \mathrm{tr}(\mathrm{Cov}(\frac{1}{T/W}\sum_{i=1}^{T/W}\mathrm{NRES}_i(\theta)) \leq \mathrm{tr}(\mathrm{Cov}(\mathrm{FullES}(\theta))).$

**Remark 10.** To understand the inequality assumption in (10), we notice that it relates the sum of the squared 2-norm of vectors $\{\sum_{t=W\cdot(k-1)+1}^{W\cdot k}g^t\}_{k=1}^{T/W}$ with the squared 2-norm of their sum. When these vectors are pointing in similar directions, this inequality would hold (to see this intuitively, consider the more extreme case when all these vectors are exactly in the same direction). Because each term $\sum_{t=W\cdot(k-1)+1}^{W\cdot k}g^t$ can be understood as the total derivative of the sum of smoothed losses in the $k$-th truncation window with respect to $\theta$, we see that inequality (10) is satisfied when, roughly speaking, different truncation windows' gradient contributions are pointing in similar directions. This is often the case for real-world applications because if we can decrease the losses within a truncation window by changing the parameter $\theta$, we likely will also decrease other truncation windows' losses. At a high-level, Theorem 9 shows that for many practical unrolled computation graphs, NRES is not only *better than* FullES *due to its better parallelizability* but also *better due to its lower variance* given the same computation budget.

**Empirical Verification of Theorem 9.** We empirically verify Theorem 9 in Figure 4(b) using the same set up of the meta-training learned optimizer task used in Figure 3(a). Here we compare the total variance of averaging $T/W = 1000$ *i.i.d.* NRES estimators versus using 1 FullES gradient estimator (same total amount of compute). We see that NRES has a significantly lower total variance than FullES while also allowing $T/W = 1000$ times wall-clock speed up due to its parallelizability.

## 5 Experiments

NRES is particularly suitable for optimization in UCGs in two scenarios: 1) when the loss surface exhibits extreme local sensitivity; 2) when automatic differentiation of the loss is not possible/gives noninformative (e.g., zero) gradients. In this section, we focus on three applications exhibiting these properties: a) learning Lorenz system's parameters (sensitive), b) meta-training learned optimizers (sensitive), and c) reinforcement learning (nondifferentiable), and show that NRES outperforms existing AD and ES methods for these applications. When comparing online gradient estimation methods, we keep the number of workers $N$ used by all methods the same for a fair comparison. For the offline method FullES, we choose its number of workers to be $W/T\times$ the number of NRES workers on all tasks (in order to keep the same number of unroll steps per-update) except for the learned optimizer task in Section 5.2 where we show that NRES can solve the task faster while using much fewer per-update steps than FullES.

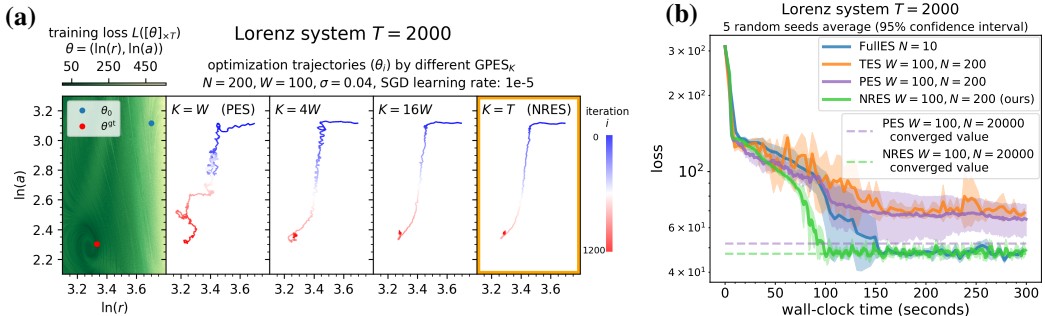

**Figure 5:** (a) The pathological training loss surface of the Lorenz system problem (left) and the optimization trajectory of different $\text{GPES}_K$ gradient estimators (right). NRES's trajectory is the smoothest because of its lowest variance. (b) Different ES methods' loss convergence on the same problem. NRES converges the fastest.

## 5.1 Learning dynamical system parameters

In this application, we consider learning the parameters of a Lorenz system, a canonical chaotic dynamical system. Here the state $s_t = (x_t, y_t, z_t) \in \mathbb{R}^3$ is unrolled with two learnable parameters $a, r$[9] with the discretized transitions ($dt = 0.005$) starting at $s_0 = (x_0, y_0, z_0) = (1.2, 1.3, 1.6)$:

$$x_{t+1} = x_t + a(y_t - x_t)dt; \quad y_{t+1} = y_t + [x_t \cdot (r - z_t) - y_t]dt; \quad z_{t+1} = z_t + [x_t \cdot y_t - 8/3 \cdot z_t]dt.$$

Due to the positive constraint on $r > 0$ and $a > 0$, we parameterize them as $\theta = (\ln(r), \ln(a)) \in \mathbb{R}^2$ and exponentiate the values in each application. We assume we observe the ground truth $z$-coordinate $z_t^{\text{gt}}$ for $t \in [T], T = 2000$ steps unrolled by the default parameters $(r^{\text{gt}}, a^{\text{gt}}) = (28, 10)$. For each step $t$, we measure the squared loss $L_t^s(s_t) := (z_t - z_t^{\text{gt}})^2$. Our goal is to recover the ground truth parameters $\theta_{\text{gt}} = (\ln(28), \ln(10))$ by optimizing the average loss over all time steps using vanilla SGD. We first visualize the training loss surface in the left panel of Figure 5(a) (also see Figure 8 in the Appendix) and notice that it has extreme sensitivity to small changes in the parameter $\theta$.

To illustrate the superior variance of NRES over other $\text{GPES}_K$ estimators, we plot in the right panels of Figure 5(a) the optimization trajectory of $\theta$ using gradient estimator $\text{GPES}_K$ with different values of $K$ under the same SGD learning rate. We see that NRES's trajectory exhibits the least amount of oscillation due to its lowest variance. In contrast, we notice that PES's trajectory is highly unstable, thus requiring a smaller learning rate than NRES to achieve a possibly slower convergence. Hence, we take extra care in tuning each method's constant learning rate and additionally allow PES to have a decay schedule. We plot the convergence of different ES gradient estimators in wall-clock time using the same hardware in Figure 5(b). (We additionally compare against automatic differentiation methods in Figure 9 in the Appendix; they all perform worse than the ES methods shown here.)

In terms of the result, we see that NRES outperforms **1)** TES, as NRES is unbiased and can better capture long-term dependencies; **2)** PES, as NRES has provably lower variance, which aids convergence in stochastic optimization; **3)** FullES, as NRES can produce more gradient updates in the same amount of wall clock time than FullES (with parallelization, each NRES update takes $O(W)$ time instead of FullES's $O(T)$ time). Additionally, we plot the asymptotically converged loss value when we train with a significantly larger number of particles ($N = 20000$) for PES and NRES. We see that by only using $N = 200$ particles, NRES can already converge around its asymptotic limit, while PES is still far from reaching its limit within our experiment time.

## 5.2 Meta-training learned optimizers

In this application [3], the meta-parameters $\theta$ of a learned optimizer control the gradient-based updates of an inner model's parameters. The inner state $s_t$ is the optimizer state which consists of both the inner model's parameters and its current gradient momentum statistics. The transition function $f_t$ computes an additive update vector to the inner parameters using $\theta$ and a random training batch and outputs the next optimizer state $s_{t+1}$. Each time step $t$'s meta-loss $L_t^s$ evaluates the updated inner parameters' generalization performance using a sampled validation batch.

We consider meta-training the learned optimizer model given in [3] ($d = 1762$) to optimize a 3-layer MLP on the Fashion MNIST dataset for $T = 1000$ steps. (We show results on the same task

---

[9]We don't learn the third parameter, fixed at $8/3$, so that we can easily visualize a 2-d loss surface.

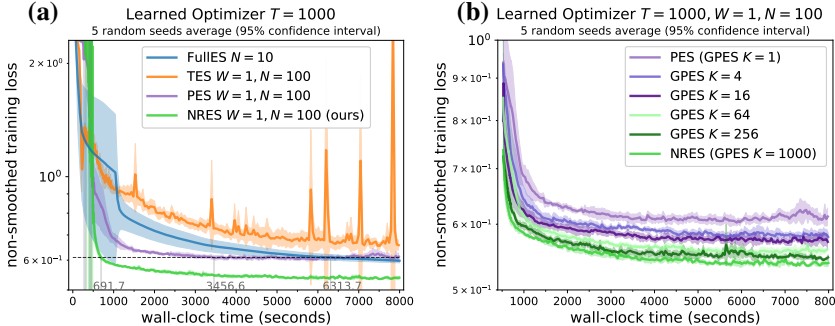

**Figure 6:** (a) Different ES gradient estimators' training loss convergence on the learned optimizer task in wall-clock time. NRES reaches the lowest loss fastest. (b) The loss convergence of $\text{GPES}_K$ gradient estimators with difference $K$ values on the same task. NRES converges the fastest due to its reduced variance.

with higher-dimension and longer horizon in Appendix E.1.2.) This task is used in the training task distribution of the state of the art learned optimizer VeLO [20][10]. The loss surface for this problem has high sharpness and many suboptimal minima as previously shown in Figure 1(a). We meta-train with Adam using different gradient estimation methods with the same hardware and tune each gradient estimation method's meta learning rate individually. Because AD methods all perform worse than the ES methods, we defer their results to Figure 10 in the Appendix and only plot the convergence of the ES methods in wall-clock time in Figure 6(a).

Here we see that NRES reaches the lowest loss value in the same amount of time. In fact, PES and FullES would require 5 and $9\times$ (respectively) longer than NRES to reach a loss NRES reaches early on during its training, while TES couldn't even reach that loss within our experiment time. It is worth noting that, for this task, NRES only require $2 \cdot N \cdot W = 200$ unrolls to produce an unbiased, low-variance gradient estimate, which is even smaller than the length of a single episode ($T = 1000$). In addition, we situate NRES's performance within our proposed class of $\text{GPES}_K$ estimators in Figure 6(b). In accordance with Corollary 8, NRES converges fastest due to its reduced variance.

### 5.3  Reinforcement Learning

It has been shown that ES is a scalable alternative to policy gradient and value function methods for solving reinforcement learning tasks [14]. In this application, we learn a linear policy [11] (following [21]) using different ES methods on the Mujoco [22] Swimmer ($d = 16$) and Half Cheetah task ($d = 102$). We minimize the average of negative per-step rewards over the horizon length of $T = 1000$, which is equivalent to maximizing the undiscounted sum rewards. Unlike [14, 21, 15], we don't use additional heuristic tricks such as **1)** rank conversion of rewards, **2)** scaling by loss standard deviation, or **3)** state normalization. Instead, we aim to compare the pure performance of different ES methods assuming perfect parallel implementations. To do this, we measure a method's performance as a function of the number of *sequential environment steps* it used. Sequential environment steps are steps that *have to* happen one after another (e.g., the environment steps within the same truncation window). However, steps that are parallelizable don't count additionally in the sequential steps. Hence, the wall-clock time under perfect parallel implementation is linear with respect to the number of sequential environment steps used. As all the methods we compare are iterative update methods, we additionally require that each method use the same number of environment steps per update when measuring each method's required number of sequential steps to solve a task. We tune the SGD learning rate individually for each method and plot their total rewards progression on both Mujoco tasks in Figure 7.

Here we see that TES fails to solve both tasks due to the short horizon bias [23], making it unable to capture the long term dependencies necessary to solve the tasks. On the other hand, PES, despite being unbiased, suffers from high variance, making it take longer (or unable in the case of Half Cheetah) to solve the task than NRES. As for FullES, despite using the same amount of compute per gradient update as NRES, it's much less parallelizable as discussed in Section 4 – it takes much longer time ($10\times$ and $60\times$) than NRES assuming perfect parallelization. In addition to the number

---

[10]We show the performance of ES methods on another task from this distribution in Appendix E.1.2.

[11]We additionally compare the ES methods on learning a non-linear ($d = 726$) policy on the Half-Cheetah task in Appendix E.1.3.

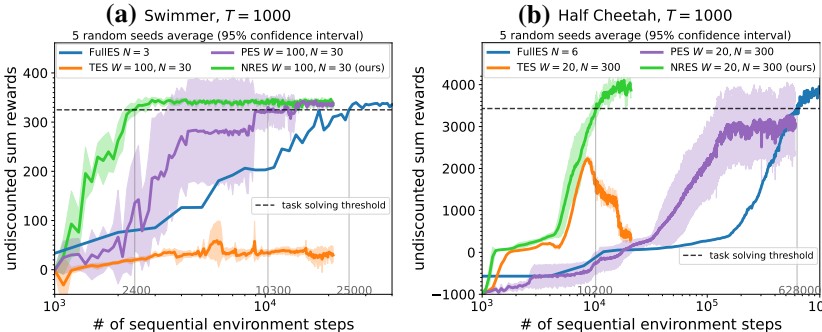

**Figure 7:** ES methods' performance vs. the number of sequential environment steps used in solving the Mujoco (a) Swimmer task and (b) Half Cheetah task. NRES solves the tasks fastest under perfect parallelization.

of sequential steps, we additionally show the total number of environment steps used by each method in Table 3 in the Appendix – NRES *also uses the least total number of steps* to solve both tasks, making it the most sample efficient.

# 6 Additional Related Work

Beyond the most related work in Section 2, in this section, we further position NRES relative to existing zeroth-order gradient estimation methods. We also provide additional related work on automatic differentiation (AD) methods for unrolled computation graphs in Appendix B.

**Zeroth-Order Gradient Estimators.** In this work, we focus on zeroth-order methods that can estimate continuous parameters' gradients to be plugged into any first-order optimizers, unlike other zeroth-order optimization methods such as Bayesian Optimization [24], random search [25], or Trust Region methods [26, 27]. We also don't compare against policy gradient methods [9], because they assume internal stochasticity of the unrolling dynamics, which may not hold for deterministic policy learning [e.g., 22]. Within the space of evolution strategies methods, many works have focused on improving the vanilla ES method's variance by changing the perturbation distribution [28–31], considering the covariance structure [32], and using control variates [33]. However, these works do not consider the unrolled structure of UCGs and are offline methods. In contrast, we reduce the variance by incorporating this unrolled aspect through online estimation and noise-reuse. As the aforementioned variance reduction methods work orthogonally to NRES, it is conceivable that these techniques can be used in conjunction with NRES to further reduce the variance.

# 7 Discussion, Limitations, and Future Work

In this work, we improve online evolution strategies for unbiased gradient estimation in unrolled computation graphs by analyzing the best noise-sharing strategies. By generalizing an existing unbiased method, Persistent Evolution Strategies, to a broader class, we analytically and empirically identify the best estimator with the smallest variance and name this method Noise-Reuse Evolution Strategies (NRES). We demonstrate the convergence benefits of NRES over other automatic differentiation and evolution strategies methods on a variety of applications.

**Limitations.** As NRES is both an online method and an ES method, it naturally inherits some limitations shared by all methods of these two classes, such as hysteresis and variance's linear dependence on the dimension $d$. We provide a detailed discussion of these limitations in Appendix F.

**Future Work.** There are some natural open questions: **1)** *choosing a better sampling distribution for* NRES. Currently the isotropic Gaussian's variance $\sigma^2$ is tuned as a hyperparameter. Whether there are better ways to leverage the sequential structure in unrolled computation graphs to automate the selection of this distribution is an open question. **2)** *Incorporating hysteresis.* Our analysis assumes no hysteresis in the gradient estimates and we haven't observed much impact of it in our experiments. However, understanding when and how to correct for hysteresis is an interesting direction.

# Acknowledgments

We thank Kevin Kuo, Jingnan Ye, Tian Li, and the anonymous reviewers for their helpful feedback.

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
