# Appendix

## Appendix Outline

# A   Notation

In this section we provide two tables (Table 1 and 2) that sumarize all the notations we use in this paper.

**Table 1:** Notations used in this paper (Part I)

| | |
|---|---|
| $T$ | the length of the unrolled computation graph. |
| $[T]$ | the set of integers $\{1, \ldots, T\}$. |
| $t \in \mathbb{Z} \cap [0, T]$ | a time step in the dynamical system. |
| $\theta \in \mathbb{R}^d$ | the learnable parameter that unrolls the dynamical system at each time step. |
| $\theta_i \in \mathbb{R}^d, i \in [T]$ | the parameter that unrolls the dynamical system at the $i$-th time step. |
| $d$ | dimension of the learnable unroll parameter $\theta$. |
| $s \in \mathbb{R}^p$ | an inner state of the dynamical system. |
| $s_t \in \mathbb{R}^p$ | the inner state of the dynamical system at time step $t$. |
| $p$ | the dimension of the inner state $s$ in the dynamical system. |
| $f_t : \mathbb{R}^p \times \mathbb{R}^d \to \mathbb{R}^p$ | the transition dynamics from state at time step $t-1$ to time step $t$. The state to be transitioned into at time step $t$ is $f_t(s_{t-1}, \theta)$. $f_t$ doesn't need to be the same for all $t \in [T]$. For example, different $f_t$ could implicitly use different data as part of the computation. |
| $L_t^s : \mathbb{R}^p \to \mathbb{R}$ | the loss function of the state $s_t$ at time step $t \in [T]$, which gives loss as $L_t^s(s_t)$. |
| $L_t : \mathbb{R}^{dt} \to \mathbb{R}$ | the loss at time step $t \in [T]$ as a function of all the $\theta_i$'s applied up to time step $t$, $L_t(\theta_1, \ldots, \theta_t)$. Here $\theta_i$ doesn't need to be all the same. |
| $L : \mathbb{R}^{dT} \to \mathbb{R}$ | the average loss over all $T$ steps incurred by unrolling the system from $s_0$ to $s_T$ using the sequence of $\{\theta_i\}_{i=1}^T$. $L(\theta_1, \ldots, \theta_T) := \frac{1}{T} \sum_{t=1}^{T} L_t(\theta_1, \ldots, \theta_t)$. |
| $L_t([\theta]_{\times a}, [\theta']_{\times t-a})$ | the loss incurred at time step $t$ by first unrolling with $\theta$ for $a$ steps, then unrolling with $\theta'$ for $t-a$ steps. $L_t([\theta]_{\times a}, [\theta']_{\times t-a}) := f(\underbrace{\theta, \ldots, \theta}_{a \text{ times}}, \underbrace{\theta', \ldots, \theta'}_{(t-a) \text{ times}})$. |
| $W$ | the length of an unroll truncation window (we always assume $T$ is divisible by $W$ for proof cleanness). |
| $I_{d \times d}$ | the $d$ by $d$ identity matrix. |
| $\sigma$ | a positive hyperparameter controlling the standard deviation in the isotropic Gaussian distribution $\mathcal{N}(\mathbf{0}, \sigma^2 I_{d \times d})$. |
| $\boldsymbol{\epsilon}$ | a random perturbation vector in $\mathbb{R}^d$ sampled from $\mathcal{N}(\mathbf{0}, \sigma^2 I_{d \times d})$. |
| $\boldsymbol{\epsilon}_i$ | the $i$-th Gaussian random vector sampled by an online evolution strategies worker in a given episode. The total number of $\epsilon_i$ in an episode might be strictly smaller than the number of truncation windows (which is $T/W$) for $\text{GPES}_K$ when $K > W$. |
| $\boldsymbol{\tau}$ | a random variable sampled from the uniform distribution $\text{Unif}\{0, W, \ldots, T - W\}$. $\tau$ denotes the starting time step of a truncation window by an online evolution strategies worker. |
| $K$ | the noise-sharing period for the algorithm $\text{GPES}_K$. $K$ is always a multiple of $W$, i.e. $K = cW$ for some positive integer $c$. |
| $c$ | the integer ratio $K/W$. |

| | |
|---|---|
| $\lceil x \rceil$ | the ceiling of $x \in \mathbb{R}$, the smallest integer $y \in \mathbb{Z}$ such that $y \geq x$ |
| $\lfloor x \rfloor$ | the floor of $x \in \mathbb{R}$, the largest integer $y \in \mathbb{Z}$ such that $y \leq x$ |
| $\mathrm{r} : \mathbb{Z}^+ \times \mathbb{Z}^+ \to \mathbb{Z}^+$ | the modified remainder function. $\mathrm{r}(x, y)$ is the unique integer $n \in [1, y]$ where $x = qy + n$ for some integer $q$. For example, if $T = 6W$ and $K = 3W$, $\mathrm{r}(T, K) = 3W$, while if $T = 6W$ and $K = 4W$, $\mathrm{r}(T, K) = 2W$. |
| $K$-smoothed loss | the loss function: $$\theta \mapsto \mathbb{E}_{\{\boldsymbol{\epsilon}_i\}} L([\theta + \boldsymbol{\epsilon}_1]_{\times K}, \dots, [\theta + \boldsymbol{\epsilon}_{\lceil T/K \rceil}]_{\times \mathrm{r}(T,K)}).$$ |
| $\{\{g_i^t \in \mathbb{R}^d\}_{i=1}^t\}_{t=1}^T$ | the classes of sets of vectors associated with a given fixed $\theta$ defined in Assumption 2. For any given $\theta$, there are $T$ such sets of vectors, one for each time step $t \in \{1, \dots, T\}$. Roughly speaking, $g_i^t$ is time step $t$'s smoothed loss's partial derivative with respect to the $i$-th application of $\theta$ |
| $g^t$ | $g^t := \sum_{i=1}^t g_i^t$ for any time step $t$. Roughly speaking, $g^t$ is time step $t$'s smoothed loss's total derivative with respect to the all the application of the same $\theta$. |
| $g_{K,j}^t$ | $g_{K,j}^t := \sum_{i=K \cdot (j-1)+1}^{\min\{t, \, K \cdot j\}} g_i^t$ for $j \in \{1, \dots, \lceil t/K \rceil\}$ and time step $t$. We can understand $g_{K,j}^t$ as the sum of partial derivatives of smoothed step-$t$ loss with respect to all $\theta$'s in the $j$-th noise-sharing window of size $K$. If $K$ doesn't divide $t$, the last such window will be shorter than $K$. |
| $g_{c,j}^t$ | $g_{K,j}^t$ when $K = c$ (used in the case of $W = 1$). |
| $\mathrm{tr}(A)$ | the trace (sum of diagonals) of a $d \times d$ matrix $A \in \mathbb{R}^{d \times d}$. |
| $\mathrm{Cov}[\boldsymbol{X}]$ | the $d \times d$ covariance matrix of random vector $\boldsymbol{X} \in \mathbb{R}^d$. |
| $\mathrm{FullES}(\theta)$ | a single FullES worker's gradient estimate given in Equation (3). To give a gradient estimate, the worker will run a total of $2T$ steps. The randomness comes from $\boldsymbol{\epsilon}$. |
| $\mathrm{TES}(\theta)$ | a single TES worker's gradient estimate given in Equation (4). This estimator keeps track of a single saved state. It samples a new noise in each truncation window and performs antithetic unrolling from the saved state. After computing the gradient estimate using the two antithetic states, the worker will run another $W$ steps from the saved state using $\theta$ without perturbation and record this as the new saved state. This estimator takes $3W$ unroll steps in total to produce a gradient estimate. The randomness comes from $\{\boldsymbol{\epsilon}_i\}_{i=1}^{T/W}$ and $\boldsymbol{\tau}$. |
| $\mathrm{PES}(\theta)$ | a single PES worker's gradient estimate given in Equation (6). This estimator keeps both a positive and negative inner state and samples a new noise perturbation at the beginning of every truncation window. It accumulates all the noise sampled in an episode to correct for bias. It runs a total of $2W$ steps to produce a gradient estimate. The randomness comes from $\{\boldsymbol{\epsilon}_i\}_{i=1}^{T/W}$ and $\boldsymbol{\tau}$. |
| $\mathrm{GPES}_K(\theta)$ | a single $\mathrm{GPES}_K(\theta)$ worker's gradient estimate given in Lemma 1. This estimator keeps both a positive and negative inner state and samples a new noise perturbation every $K$ steps in a given episode. It also accumulates past sampled noise for bias correction. To give a gradient estimate, the worker will run a total of $2W$ steps. The randomness comes from $\{\boldsymbol{\epsilon}_i\}_{i=1}^{\lceil T/K \rceil}$ and $\boldsymbol{\tau}$. |
| $\mathrm{NRES}(\theta)$ | a single NRES worker's gradient estimate. It is the same as $\mathrm{GPES}_{K=T}(\theta)$. This estimator keeps both a positive and negative inner state, and it only samples a noise perturbation once at the beginning of each episode. To give a gradient estimate, the worker will run a total of $2W$ steps. The randomness comes from $\boldsymbol{\epsilon}$ (single noise sampled at the beginning of an episode) and $\boldsymbol{\tau}$. |

# B   Additional Related Work

Beyond the related work we discuss in Section 2 and 6 on evolution strategies methods, in this section, we discuss additional related work in gradient estimation for unrolled computation graphs, including work on automatic differentiation (AD) (reverse mode and forward mode) methods and a concurrent work on online evolution strategies. Some of the AD methods described in this section are compared against as baselines in our experiments in Section 5.1 and 5.2.

**Reverse Mode Differentiation (RMD).**   When the loss function and transition functions in UCG is differentiable, the default method for computing gradients is backpropagation through time (BPTT). However, BPTT has difficulties when applied to UCGs: **1)** *memory issues*: the default BPTT implementations [e.g., 34, 35] store all activations of the graph in memory, making memory usage scale linearly with the length of the unrolled graph. There are works that improve the memory dependency of BPTT; however, they either require customized framework implementation [36, 37] or specially-designed reversible computation dynamics [5, 38]. **2)** *not online*: each gradient estimate using BPTT requires full forward and backward computation through the UCG, which is computationally expensive and incurs large latency between successive parameter updates. To alleviate the memory issue and allow online updates, a popular alternative is truncated backpropagation through time (TBPTT) which estimates the gradient within short truncation windows. However, this blocks the gradient flow to the parameters applied before the current window, making the gradient estimate biased and unable to capture long-term dependencies [23]. In contrast, NRES is memory efficient, online, and doesn't suffer from bias, while able to handle loss surfaces with extreme local sensitivity.

**Forward Mode Differentiation (FMD).**   An alternative to RMD in automatic differentiation is FMD, which computes gradient estimates through Jacobian-vector products alongside the actual forward computation, thus allowing for online applications. Among FMD methods, real-time recurrent learning (RTRL) [39] requires a computation cost that scales with the dimension of the learnable parameter, making it intractable for large problems. To alleviate the computation cost, stochastic approximations of RTRL have been proposed: DODGE [40] computes directional gradient along a certain direction; UORO [41] unbiasedly approximates the Jacobian with a rank-1 matrix; $KF - RTRL$ [42] and OK [43] uses Kronecker product decomposition to improve the gradient estimate's variance, but are specifically for RNNs. In Section 5.1 and 5.2, we experiment with forward mode methods DODGE (with standard Gaussian random directions) and UORO and demonstrate NRES's advantage over these two methods when the loss surfaces have high sensitivity to small changes in the parameter space.

**Concurrent work on online evolution strategies.**   Finally, we note that a concurrent work [19] on online evolution strategies proposes a similar algorithm (ES-Single) to the algorithm NRES proposed in our paper. However, their analyses and experiments differ from ours in a number of ways:

1. *Theoretical assumptions.* To capture the online nature of the online ES methods considered in our paper, we adopt a novel view which treats the random truncation window that an online ES method starts from as a random variable (which we denote by $\tau$). In contrast, this assumption is not made neither in the prior work on PES [15] nor in the concurrent work [19]. As such, these analyses cannot distinguish the theoretical difference between the estimator $\text{FullES}(\theta)$ and $\text{NRES}(\theta)$.

2. *Theoretical conclusions.* As a result of our novel viewpoint/theoretical assumption regarding the random truncation window used in online ES methods, we provide a precise variance characterization of our newly proposed class of $\text{GPES}_K$ gradient estimators. Two of the main theoretical contributions of our work are thus: *a)* showing that NRES provably has the lowest variance among the entire GPES class and *b)* identifying the conditions under which NRES can have lower variance than FullES with the same compute budget. In contrast, [19] do not make such contributions — in fact, in their theoretical analyses, they characterize their proposed method ES-Single as having the exact same variance as FullES, and as such they are unable to draw conclusions about the variance reduction benefits of their approach as we have done in our analyses.

3. *Experimental comparison against non-online method* FullES. In [15, 19], when comparing against ES methods, the authors primarily compare against online ES methods but not the canonical non-online ES method, FullES. In contrast, in our paper, we experimentally show that NRES can indeed provide significant speedup benefits over its non-online counterpart FullES. We believe these more complete results provide critical evidence which encourages evolution strategies users to consider switching to online methods (NRES) when working with unrolled computation graphs.

4. *Identifying the appropriate scenarios for the proposed method.* In our work, we precisely identify

problem scenarios that are most appropriate for the use of NRES (when the losses are extremely locally sensitive or blackbox) and provide experiments mirroring these scenarios to compare different gradient estimation (both ES and AD) methods. In contrast, [19] perform some of their experiments on problems where the loss surface might not have high sensitivity (e.g., LSTM copy task) and only show performance of ES methods (but not AD methods). However, for these scenarios where the loss surfaces are well-behaved, ES methods likely should not be used in the first place over traditional AD approaches (we discuss this further in Section F in the Appendix). In addition, [19] treats the application of their proposed method to blackbox losses (e.g., reinforcement learning) as future work, while we provide experiments demonstrating the effectiveness of NRES over other ES methods for this important set of applications in Section 5.3.

## C   Algorithms

In this section, we provide Python-style pseudocode for the gradient estimation algorithms discussed in this paper. We first provide the pseudocode for FullES. We then provide the pseudocode for general online evolution stratgies training. We finally provide the pseudocode for Truncated Evolution Strategies (TES) and Generalized Persistent Evolution Strategies (GPES).

### C.1   FullES Pseudocode

We show the pseudocode for the vanilla antithetic evolution strategies gradient estimation method FullES in Algorithm 3. Here we note that the FullESWorker is stateless (it has a boilerplate __init__() function). In addition, to produce a single gradient estimate, it needs to run from the beginning of the UCG ($s_0$) to the end of the graph after $T$ unroll steps. This is in contrast to the online ES methods which only unroll a truncation window of $W$ steps forward for each gradient estimate.

---
**Algorithm 3** Vanilla Antithetic Evolution Strategies (FullES)

---
**class** FullESWorker:
  **def** __init__(self,):
    # no need to initialize since FullES is stateless
    **pass**

  **def** gradient_estimate(self, $\theta$):
    $\epsilon \sim \mathcal{N}(\mathbf{0}, \sigma^2 I_{d \times d})$
    # FullES always starts from the beginning of an episode
    $(s^+, s^-) = (s_0, s_0)$
    $L^+_{\text{sum}} = 0; \quad L^-_{\text{sum}} = 0$
    # FullES always runs till the end of an episode
    **for** $t$ **in** range(1, $T+1$):
      $s^+ = f_t(s^+, \theta + \epsilon)$
      $s^- = f_t(s^-, \theta - \epsilon)$
      $L^+_{\text{sum}} \mathrel{+}= L^s_t(s^+)$
      $L^-_{\text{sum}} \mathrel{+}= L^s_t(s^-)$

    $g = (L^+_{\text{sum}} - L^-_{\text{sum}})/(2\sigma^2 \cdot T) \cdot \epsilon$
    **return** $g$

---

### C.2   Online Evolution Strategies Pseudocode

OnlineESWorker (Algorithm 4) is an abstract class (interface) that all the online ES methods will implement. The key functionality an OnlineESWorker provides is its worker.gradient_estimate($\theta$) function that performs unrolls in a truncation window of size $W$ and returns a gradient estimate based on the unroll. With this interface, we can train using online Evolution Strategies workers following Algorithm 5. The training takes two steps:

Step 1. Constructing independent *step-unlocked* workers to form a worker pool. This requires sampling different truncation window starting time step $\tau$ for different workers. During this stage, for simplicity and rigor, we only rely on the .gradient_estimate method call's side effect to alter the worker's saved states and discard the computed gradients. (We still count these environment steps for the reinforcement learning experiment in Experiment 5.3).)

Step 2. Training using the worker pool. At each outer iteration, we average all the worker's computed gradient estimates and pass that to any first order optimizer OPT_UPDATE (e.g. SGD or Adam) to update $\theta$ and repeat until convergence. Each worker's gradient_estimate method call can be parallelized.

---

**Algorithm 4** Online Evolution Strategies (OES) (a qbstract class)

---

**import** abc      `# abstract base class`
**class** OnlineESWorker(abc.ABC):
  $W$: int      `# the size of the truncation window`

  @abc.abstractmethod
  **def** __init__(self, $W$):
    `"""`

    `set up the saved states and other bookkeeping variables`
    `"""`

  @abc.abstractmethod
  **def** gradient_estimate(self, $\theta$):
    `"""`

    `Given a` $\theta$`,`
      `perform partial unroll in a truncation window of length` $W$ `and return a gradient estimate for` $\theta$
      `save the end inner state(s) and start off from the saved state(s)`
        `when` self.gradient_estimate `is called again`
      `if reach the end, reset to the initial state` $s_0$
    `"""`

---

**Algorithm 5** Training using Online Evolution Strategies

---

  $\theta = \theta_{\text{init}}$      `# start value of` $\theta$ `optimization.`

  `# Step 1:  Initialize online ES workers`
  worker_list = []
  **for** $i$ **in** range(N):                   `#` $N$ `is the number of workers, can be parallelized`
    new_worker = OnlineESWorker($W$)      `# replace with a real implementation of OnlineESWorker`
    `# the steps below make sure the workers are step-unlocked`
    `# i.e.  working independently at different truncation windows`
    $\boldsymbol{\tau} \sim \text{Unif}\{0, W, \ldots, T - W\}$
    **for** $t$ **in** range($\boldsymbol{\tau}/W$):
      _ = new_worker.gradient_estimate($\theta$)
      worker_list.append(new_worker)

  `# Step 2:  training`
  **while** not converged:
    $g_{\text{sum}} = \mathbf{0}$                     `#` $g_{\text{sum}}$ `is a vector in` $\mathbb{R}^d$
    **for** worker **in** worker_list:         `# can be parallelized`
      $g_{\text{sum}}$ += worker.gradient_estimate($\theta$)      `# accumulate this worker's gradient estimate`
    $g = g_{\text{sum}}/N$               `# average all workers' gradient estimates`
    $\theta = \text{OPT\_UPDATE}(\theta, g)$        `# updating` $\theta$ `with any first order optimizers`

---

## C.3 Truncated Evolution Strategies Pseudocode

In Section 2, we have described the biased online evolution strategies method Truncated Evolution Strategies (TES). We have provided its analytical form:

$$\frac{1}{2\sigma^2 W} \sum_{i=1}^{W} \left[ L_{\tau+i}([\theta]_{\times\tau}, [\theta + \epsilon_{(\tau/W)+1}]_{\times i}) - L_{\tau+i}([\theta]_{\times\tau}, [\theta - \epsilon_{(\tau/W)+1}]_{\times i}) \right] \epsilon_{(\tau/W)+1}. \qquad (11)$$

Here we provide the algorithm pseudocode for TES in Algorithm 6. It is important to note that after the antithetic unrolling using perturbed $\theta + \epsilon$ and $\theta - \epsilon$ for gradient estimates, another $W$ steps of unrolling is performed starting from the saved starting state using the unperturbed $\theta$. Because of this, a TESWorker requires a total of $3W$ unroll steps to produce a gradient estimate (unlike PES, GPES, and NRES which requires $2W$). The algorithm in this form is first introduced in [15].

---

**Algorithm 6** Truncated Evolution Strategies (TES)

---

**class** TESWorker(OnlineESWorker):
  **def** __init__(self, $W$):
    self.$\tau = 0$;  self.$s = s_0$
    self.$W = W$

  **def** gradient_estimate(self, $\theta$):
    `# sample at every truncation window`
    $\epsilon \sim \mathcal{N}(\mathbf{0}, \sigma^2 I_{d\times d})$

    $(s^+, s^-) = (\text{self}.s, \text{self}.s)$   `# unroll from the same state for the antithetic pair`
    $L_{\text{sum}}^+ = 0$;  $L_{\text{sum}}^- = 0$

    **for** $i$ **in** range(1, self.$W$+1):
      $s^+ = f_{\text{self}.\tau+i}(s^+, \theta + \epsilon)$
      $s^- = f_{\text{self}.\tau+i}(s^-, \theta - \epsilon)$
      $L_{\text{sum}}^+ \mathrel{+}= L_{\text{self}.\tau+i}^s(s^+)$
      $L_{\text{sum}}^- \mathrel{+}= L_{\text{self}.\tau+i}^s(s^-)$

    $g = (L_{\text{sum}}^+ - L_{\text{sum}}^-)/(2\sigma^2 \cdot \text{self}.W) \cdot \epsilon$

    **for** $i$ **in** range(1, self.$W$+1):   `# finally unroll using unperturbed `$\theta$
      self.$s = f_{\text{self}.\tau+i}(\text{self}.s,\ \theta)$;

    self.$\tau$ = self.$\tau + W$
    **if** self.$\tau \geq T$:        `# reset at the end of an episode`
      self.$\tau = 0$
      self.$s = s_0$

    **return** $g$

---

## C.4 Generalized Persistent Evolution Strategies Pseudocode

In section 3, we propose a new class of unbiased online evolution strategies methods which we name *Generalized Persistent Evolution Strategies* (GPES). It produces an unbiased gradient estimate of the $K$-smoothed loss objective defined in Equation 7 in the main paper. It samples a new Gaussian noise for perturbation every $K$ unroll steps ($K$ is a multiple of the truncation window size $W$). We provide the pseudocode for GPES in Algorithm 7.

---

**Algorithm 7** Generalized Persistent Evolution Strategies (GPES)

---

**class** GPESWorker(OnlineESWorker):

    **def** __init__(self, $W$, $K$):

        self.$\boldsymbol{\tau} = 0$;  self.$s^+ = s_0$;  self.$s^- = s_0$

        self.$W = W$;  self.$K = K$;  self.$\boldsymbol{\xi} = \mathbf{0} \in \mathbb{R}^d$

    **def** gradient_estimate(self, $\theta$):

        # only sample a new $\epsilon$ when self.$\tau$ is a multiple of self.$K$

        **if** self.$\boldsymbol{\tau}$ % self.$K$ == 0:

            $\epsilon_{\text{new}} \sim \mathcal{N}(\mathbf{0}, \sigma^2 I_{d \times d})$

            # keep track of $\epsilon_{\text{new}}$ to use for the next $K$ steps

            self.$\boldsymbol{\epsilon} = \boldsymbol{\epsilon}_{\text{new}}$

            self.$\boldsymbol{\xi}$ += $\boldsymbol{\epsilon}_{\text{new}}$

        # after the if statement above, self.$\epsilon$ is now $\epsilon_{\lfloor \text{self.}\tau/K \rfloor + 1}$

        # and self.$\xi$ is now $\sum_{i=1}^{\lfloor \tau/K \rfloor + 1} \epsilon_i$

        $(s^+, s^-) = (\text{self.}s^+, \text{self.}s^-)$

        $L_{\text{sum}}^+ = 0$;   $L_{\text{sum}}^- = 0$

        **for** $i$ **in** range(1, self.$W$+1):

            $s^+ = f_{\text{self.}\boldsymbol{\tau}+i}(s^+, \theta + \text{self.}\boldsymbol{\epsilon})$

            $s^- = f_{\text{self.}\boldsymbol{\tau}+i}(s^-, \theta - \text{self.}\boldsymbol{\epsilon})$

            $L_{\text{sum}}^+$ += $L_{\text{self.}\boldsymbol{\tau}+i}^s(s^+)$

            $L_{\text{sum}}^-$ += $L_{\text{self.}\boldsymbol{\tau}+i}^s(s^-)$

        $g = (L_{\text{sum}}^+ - L_{\text{sum}}^-)/(2\sigma^2 \cdot \text{self.}W) \cdot \text{self.}\boldsymbol{\xi}$

        self.$s^+ = s^+$; self.$s^- = s^-$

        self.$\boldsymbol{\tau}$ = self.$\boldsymbol{\tau} + W$

        **if** self.$\boldsymbol{\tau} \geq T$:        # reset at the end of an episode

            self.$\boldsymbol{\tau} = 0$; self.$s^+ = s_0$; self.$s^- = s_0$

            self.$\boldsymbol{\xi} = \mathbf{0}$

        **return** $g$

---

# D Theory and Proofs

In this section, we provide the proofs and interpretations of the lemmas, assumptions, and theorems presented in the main paper (we also restate these results for completeness). Before we begin, we set some background notation:

- We treat all real-valued function's gradient as column vectors.
- We use $[\theta]_{\times t}$ to denote $t$ copies to $\theta$ stacked together to form a $td$-dimensional column vector.

- For $\{v_i \in \mathbb{R}^d\}_{i=1}^m$, we use $\begin{bmatrix} v_1 \\ \vdots \\ v_m \end{bmatrix}$ to denote the column vector whose first $d$-dimensions is $v_1$ and

  so on and so forth. Similar we use $\begin{bmatrix} v_1 & \dots & v_m \end{bmatrix}$ to denote the transpose of the previous vector (thus a row vector).

- **Gradient notation** For any real-valued function whose input is more than one single $\theta$ (e.g., $L_t$ which takes in $t$ copies of $\theta$'s), we will use $\nabla_\Theta$ to describe the gradient with respect to the function's entire input dimension and similarly $\nabla_\Theta^2$ for Hessian. For such functions, we will use $\frac{\partial}{\partial \theta_i}$ to denote the partial derivative of the function with respect to the $i$-th $\theta$. We will use $\frac{d}{d\theta}$ to define the total derivative of some variable with respect to $\theta$ (e.g. when talking about $\frac{dL_{\text{avg}}}{d\theta}$ with $L_{\text{avg}}(\theta) := L([\theta]_{\times t})$ if $L$ is differentiable). This operator $\frac{d}{d\theta}$ will also produce the Jacobian matrix

when the applied function is vector-valued.

## D.1  Proof of Lemma 1

Define the $K$-smoothed loss objective as the function:

$$\theta \mapsto \mathbb{E}_{\{\epsilon_i\}} L([\theta + \epsilon_1]_{\times K}, \ldots, [\theta + \epsilon_{\lceil T/K \rceil}]_{\times \mathrm{r}(T,K)}). \tag{12}$$

**Lemma 1.** *An unbiased gradient estimator for the $K$-smoothed loss is given by*

$$\mathrm{GPES}_{K=cW}(\theta) \tag{13}$$

$$:= \frac{1}{2\sigma^2 W} \sum_{j=1}^{W} \left[ L_{\tau+j}([\theta + \epsilon_1]_{\times K}, [\theta + \epsilon_2]_{\times K}, \ldots, [\theta + \epsilon_{\lfloor \tau/K \rfloor + 1}]_{\times \mathrm{r}(\tau+j,K)}) \right.$$

$$\left. - L_{\tau+j}([\theta - \epsilon_1]_{\times K}, [\theta - \epsilon_2]_{\times K}, \ldots, [\theta - \epsilon_{\lfloor \tau/K \rfloor + 1}]_{\times \mathrm{r}(\tau+j,K)}) \right] \cdot \left( \sum_{i=1}^{\lfloor \tau/K \rfloor + 1} \epsilon_i \right), \tag{14}$$

*with randomness in $\tau \sim \mathrm{Unif}\{0, W, \ldots, T - W\}$ and $\{\epsilon_i\}_{i=1}^{\lceil T/K \rceil} \stackrel{\text{iid}}{\sim} \mathcal{N}(\mathbf{0}, \sigma^2 I_{d \times d})$.*

*Proof.*

For simplicity, we will denote $q = \lceil T/K \rceil$ and $r = \mathrm{r}(T, K)$.

First let's define a function $L^K : \mathbb{R}^{d \cdot q} \to \mathbb{R}$

$$L^K(\theta_1, \ldots, \theta_q) := L([\theta_1]_{\times K}, \ldots, [\theta_{q-1}]_{\times K}, [\theta_q]_{\times r}), \tag{15}$$

and a smoothed version of $L^K$ by $\widehat{L}^K : \mathbb{R}^{d \cdot q} \to \mathbb{R}$ with

$$\widehat{L}^K(\theta_1, \ldots, \theta_q) := \mathbb{E}_{\{\epsilon_i\}_{i=1}^q} L^K(\theta_1 + \epsilon_1, \ldots, \theta_q + \epsilon_q). \tag{16}$$

We notice that by definition, the $K$-smoothed loss function can be expressed as

$$\theta \mapsto \widehat{L}^K([\theta]_{\times q}). \tag{17}$$

In this form, the $K$-smoothed loss function is a simple composition of two functions: the first function maps $\theta$ to $q$-times repetition of $[\theta]_{\times q}$, while the second function is exactly $\widehat{L}^K$. For the first vector-valued function, we see that its Jacobian is given by:

$$\frac{d}{d\theta}(\theta \mapsto [\theta]_{\times q}) = \mathbf{1}_q \otimes I_{d \times d} \in \mathbb{R}^{qd \times d}, \tag{18}$$

where $\mathbf{1}_q \in \mathbb{R}^q$ is a vector of 1's and $\otimes(\cdot, \cdot)$ is the Kronecker product operator. For the second function, we see that by the score function gradient estimator trick [16],

$$\nabla_\Theta \widehat{L}^K(\theta_1, \ldots, \theta_q) = \frac{1}{\sigma^2} \mathbb{E}_{\{\epsilon_i\}_{i=1}^q} L^K(\theta_1 + \epsilon_1, \ldots, \theta_q + \epsilon_q) \begin{bmatrix} \epsilon_1 \\ \vdots \\ \epsilon_q \end{bmatrix}. \tag{19}$$

Now we are ready to compute the gradient of the $K$-smoothed loss using chain rule (recall that we assume the gradients are column vectors):

$$\nabla_\theta \left( \theta \mapsto \widehat{L}^K([\theta]_{\times q}) \right) \tag{20}$$

$$= \left[ \frac{d}{d\theta}(\theta \mapsto [\theta]_{\times q}) \right]^\top \nabla_\Theta \widehat{L}^K \Big|_{\Theta = [\theta]_{\times q}} \tag{21}$$

$$= (\mathbf{1}_q \otimes I_{d \times d})^\top \mathbb{E}_{\{\epsilon_i\}_{i=1}^q} \frac{1}{\sigma^2} L^K(\theta + \epsilon_1, \ldots, \theta + \epsilon_q) \begin{bmatrix} \epsilon_1 \\ \vdots \\ \epsilon_q \end{bmatrix} \tag{22}$$

$$= \frac{1}{\sigma^2} \mathbb{E}_{\{\epsilon_i\}_{i=1}^q} L^K(\theta + \epsilon_1, \ldots, \theta + \epsilon_q) \left( (\mathbf{1}_q \otimes I_{d \times d})^\top \begin{bmatrix} \epsilon_1 \\ \vdots \\ \epsilon_q \end{bmatrix} \right) \tag{23}$$

$$= \frac{1}{\sigma^2} \mathbb{E}_{\{\epsilon_i\}_{i=1}^q} L^K(\theta + \epsilon_1, \ldots, \theta + \epsilon_q)(\sum_{i=1}^q \epsilon_i). \tag{24}$$

Here the last step is by the algebra of Kronecker product. With this, we now consider the structure of $L^K$ as an average of losses over all time steps by converting this average into an expectation over truncation windows starting at $\tau$:

$$\nabla_\theta \left( \theta \mapsto \widehat{L}^K([\theta]_{\times q}) \right) \tag{25}$$

$$= \frac{1}{\sigma^2} \mathbb{E}_{\{\epsilon_i\}_{i=1}^q} L^K(\theta + \epsilon_1, \ldots, \theta + \epsilon_q)(\sum_{i=1}^q \epsilon_i) \tag{26}$$

$$= \frac{1}{\sigma^2} \mathbb{E}_{\{\epsilon_i\}_{i=1}^q} \frac{1}{T} \sum_{t=1}^T L_t([\theta + \epsilon_1]_{\times K}, [\theta + \epsilon_2]_{\times K} \ldots, [\theta + \epsilon_{\lceil t/K \rceil}]_{\times r(t,K)})(\sum_{i=1}^q \epsilon_i) \tag{27}$$

$$= \frac{1}{\sigma^2} \mathbb{E}_{\{\epsilon_i\}_{i=1}^q} \frac{1}{T/W} \sum_{b=0}^{T/W-1}$$
$$\left[ \frac{1}{W} \sum_{j=1}^W L_{Wb+j}([\theta + \epsilon_1]_{\times K}, [\theta + \epsilon_2]_{\times K} \ldots, [\theta + \epsilon_{\lceil (Wb+j)/K \rceil}]_{\times r(Wb+j,K)})(\sum_{i=1}^q \epsilon_i) \right] \tag{28}$$

$$= \frac{1}{\sigma^2} \mathbb{E}_{\{\epsilon_i\}_{i=1}^q} \frac{1}{T/W} \sum_{b=0}^{T/W-1}$$
$$\left[ \frac{1}{W} \sum_{j=1}^W L_{Wb+j}([\theta + \epsilon_1]_{\times K}, [\theta + \epsilon_2]_{\times K} \ldots, [\theta + \textcolor{blue}{\epsilon_{\lfloor Wb/K \rfloor + 1}}]_{\times r(Wb+j,K)})(\sum_{i=1}^q \epsilon_i) \right] \tag{29}$$

Here the last step we observe that $\lceil (Wb+j)/K \rceil = \lfloor Wb/K \rfloor + 1$ for any $1 \leq j \leq W \leq K$.

Here we notice that in Equation 29, for $i > \lfloor Wb/K \rfloor + 1$, there is independence between the random vector $\epsilon_i$ and the term $L_{Wb+j}([\theta + \epsilon_1]_{\times K}, [\theta + \epsilon_2]_{\times K} \ldots, [\theta + \epsilon_{\lfloor Wb/K \rfloor + 1}]_{\times r(Wb+j,K)})$. The expectation of the product between these independent terms is then $\mathbf{0}$ because $\mathbb{E}[\epsilon_i] = \mathbf{0}$. As a result, we have the further simplification:

$$\nabla_\theta \left( \theta \mapsto \widehat{L}^K([\theta]_{\times q}) \right) \tag{30}$$

$$= \frac{1}{\sigma^2} \mathbb{E}_{\{\epsilon_i\}_{i=1}^q} \frac{1}{T/W} \sum_{b=0}^{T/W-1}$$
$$\left[ \frac{1}{W} \sum_{j=1}^W L_{Wb+j}([\theta + \epsilon_1]_{\times K}, [\theta + \epsilon_2]_{\times K} \ldots, [\theta + \epsilon_{\lfloor Wb/K \rfloor + 1}]_{\times r(Wb+j,K)})( \sum_{i=1}^{\lfloor Wb/K \rfloor + 1} \epsilon_i) \right] \tag{31}$$

$$= \frac{1}{\sigma^2} \mathbb{E}_{\{\epsilon_i\}_{i=1}^q} \mathbb{E}_\tau \frac{1}{W} \sum_{j=1}^W L_{\tau+j}([\theta + \epsilon_1]_{\times K}, [\theta + \epsilon_2]_{\times K} \ldots, [\theta + \epsilon_{\lfloor \tau/K \rfloor + 1}]_{\times r(\tau+j,K)})( \sum_{i=1}^{\lfloor \tau/K \rfloor + 1} \epsilon_i). \tag{32}$$

Here the last step converts the average in Equation 31 into an expectation in Equation 32 by treating $Wb$ as the random variable $\tau$. By additionally averaging over the antithetic samples of the random variable in Equation 32, we arrive at the unbiased estimator given in the Lemma.

$\square$

## D.2 Interpretation of Assumption 2

**Assumption 2.** For a given fixed $\theta \in \mathbb{R}^d$, for any $t \in [T]$, there exists a set of vectors $\{g_i^t \in \mathbb{R}^d\}_{i=1}^t$, such that for any $\{v_i \in \mathbb{R}^d\}_{i=1}^t$, the following equality holds:

$$L_t(\theta + v_1, \theta + v_2, \dots, \theta + v_t) - L_t(\theta - v_1, \theta - v_2, \dots, \theta - v_t) = 2 \sum_{i=1}^t (v_i)^\top (g_i^t) \tag{33}$$

Here we show that when $L_t : \mathbb{R}^{dt} \to \mathbb{R}$ is a quadratic function, this assumption would hold with $g_i^t = \frac{\partial L_t}{\partial \theta_i}$.

If $L_t$ is a quadratic (assumption made in [15]), it can be expressed exactly as its second-order Taylor expansion. Then we have

$$L_t(\theta + v_1, \theta + v_2, \dots, \theta + v_t) = L_t([\theta]_{\times t}) + \left( \nabla_\Theta L_t \Big|_{\Theta=[\theta]_{\times t}} \right)^\top \begin{bmatrix} v_1 \\ \vdots \\ v_t \end{bmatrix} + \frac{1}{2} \begin{bmatrix} v_1 & \cdots & v_t \end{bmatrix} \nabla_\Theta^2 L_t \begin{bmatrix} v_1 \\ \vdots \\ v_t \end{bmatrix} \tag{34}$$

$$L_t(\theta - v_1, \theta - v_2, \dots, \theta - v_t) = L_t([\theta]_{\times t}) - \left( \nabla_\Theta L_t \Big|_{\Theta=[\theta]_{\times t}} \right)^\top \begin{bmatrix} v_1 \\ \vdots \\ v_t \end{bmatrix} + \frac{1}{2} \begin{bmatrix} v_1 & \cdots & v_t \end{bmatrix} \nabla_\Theta^2 L_t \begin{bmatrix} v_1 \\ \vdots \\ v_t \end{bmatrix} \tag{35}$$

Taking the difference of the above two equations, we have that

$$L_t(\theta + v_1, \theta + v_2, \dots, \theta + v_t) - L_t(\theta - v_1, \theta - v_2, \dots, \theta - v_t) = 2 \left( \nabla_\Theta L_t \Big|_{\Theta=[\theta]_{\times t}} \right)^\top \begin{bmatrix} v_1 \\ \vdots \\ v_t \end{bmatrix}. \tag{36}$$

We note that

$$\nabla_\Theta L_t \Big|_{\Theta=[\theta]_{\times t}} = \begin{bmatrix} \frac{\partial L_t}{\partial \theta_1} \\ \cdots \\ \frac{\partial L_t}{\partial \theta_t} \end{bmatrix}. \tag{37}$$

Plugging it into Equation (35), we have

$$L_t(\theta + v_1, \theta + v_2, \dots, \theta + v_t) - L_t(\theta - v_1, \theta - v_2, \dots, \theta - v_t) = 2 \sum_{i=1}^t (v_i)^\top \frac{\partial L_t}{\partial \theta_i}. \tag{38}$$

Thus we see in the case of quadratic $L_t$, $g_i^t$ is just the partial derivative of $L_t$ with respect to $\theta_i$ (i.e., $\frac{\partial L_t}{\partial \theta_i}$). Hence we see that our assumptions generalize those made in [15].

## D.3 Proof of Lemma 4

**Lemma 4.** *Under Assumption 2, when $W = 1$, $\mathrm{GPES}_{K=c}(\theta) = \frac{1}{\sigma^2} \sum_{j=1}^{\lfloor \tau/c \rfloor + 1} \left( \sum_{i=1}^{\lfloor \tau/c \rfloor + 1} \epsilon_i \right) \epsilon_j^\top g_{c,j}^{\tau+1}$, where the randomness lies in $\tau \sim \mathrm{Unif}\{0, 1, \dots, T-1\}$ and $\{\epsilon_i\} \overset{iid}{\sim} \mathcal{N}(\mathbf{0}, \sigma^2 I_{d \times d})$.*

*Proof.* We see that when $W = 1$, we have $K = cW = c$ and

$$\mathrm{GPES}_{K=c}(\theta) \tag{39}$$

$$= \frac{1}{2\sigma^2 W} \sum_{j=1}^{W=1} \left[ L_{\tau+j}([\theta + \epsilon_1]_{\times K}, [\theta + \epsilon_2]_{\times K}, \dots, [\theta + \epsilon_{\lfloor \tau/K \rfloor + 1}]_{\times r(\tau+j, K)}) \right.$$

$$\left. - L_{\tau+j}([\theta - \epsilon_1]_{\times K}, [\theta - \epsilon_2]_{\times K}, \dots, [\theta - \epsilon_{\lfloor \tau/K \rfloor + 1}]_{\times r(\tau+j, K)}) \right] \cdot \left( \sum_{i=1}^{\lfloor \tau/K \rfloor + 1} \epsilon_i \right) \tag{40}$$

$$= \frac{1}{2\sigma^2} \Bigg[ L_{\tau+1}([\theta + \epsilon_1]_{\times c}, [\theta + \epsilon_2]_{\times c}, \ldots, [\theta + \epsilon_{\lfloor \tau/c \rfloor + 1}]_{\times r(\tau+1,c)})$$

$$- L_{\tau+1}([\theta - \epsilon_1]_{\times c}, [\theta - \epsilon_2]_{\times c}, \ldots, [\theta - \epsilon_{\lfloor \tau/c \rfloor + 1}]_{\times r(\tau+1,c)}) \Bigg] \cdot \left( \sum_{i=1}^{\lfloor \tau/c \rfloor + 1} \epsilon_i \right) \tag{41}$$

Applying Assumption 2 to the difference in the last equation, we have

$$L_{\tau+1}([\theta + \epsilon_1]_{\times c}, [\theta + \epsilon_2]_{\times c}, \ldots, [\theta + \epsilon_{\lfloor \tau/c \rfloor + 1}]_{\times r(\tau+1,c)})$$
$$- L_{\tau+1}([\theta - \epsilon_1]_{\times c}, [\theta - \epsilon_2]_{\times c}, \ldots, [\theta - \epsilon_{\lfloor \tau/c \rfloor + 1}]_{\times r(\tau+1,c)}) \tag{42}$$

$$= 2 \sum_{j=1}^{\lfloor \tau/c \rfloor + 1} \left[ \epsilon_j^\top \left( \sum_{i=c\cdot(j-1)+1}^{\min\{\tau+1, c\cdot j\}} g_i^{\tau+1} \right) \right] \tag{43}$$

$$= 2 \sum_{j=1}^{\lfloor \tau/c \rfloor + 1} \epsilon_j^\top g_{c,j}^{\tau+1} \tag{44}$$

Plugging this equality in to Equation 41, we get

$$\text{GPES}_{K=c}(\theta) = \frac{1}{\sigma^2} \sum_{j=1}^{\lfloor \tau/c \rfloor + 1} \left( \sum_{i=1}^{\lfloor \tau/c \rfloor + 1} \epsilon_i \right) \epsilon_j^\top g_{c,j}^{\tau+1}$$

$\square$

### D.4 Proof of Theorem 5

**Theorem 5.** *When $W = 1$ and under Assumption 2, the total variance of $\text{GPES}_{K=c}(\theta)$ has the following form for any integer $c \in [1, T]$,*

$$\text{tr}(\text{Cov}[\text{GPES}_{K=c}(\theta)]) = \frac{(d+2)}{T} \sum_{t=1}^{T} \left( \|g^t\|_2^2 \right) - \left\| \frac{1}{T} \sum_{t=1}^{T} g^t \right\|_2^2 + \frac{1}{T} \sum_{t=1}^{T} \left( \frac{d}{2} \sum_{j=1,j'=1}^{\lceil t/c \rceil} \|g_{c,j}^t - g_{c,j'}^t\|_2^2 \right). \tag{45}$$

*Proof.* First we break down the total variance:

$$\text{tr}(\text{Cov}[\text{GPES}_{K=c}(\theta)]) \tag{46}$$

$$= \text{tr}(\mathbb{E}\left[ (\text{GPES}_{K=c}(\theta) - \mathbb{E}[\text{GPES}_{K=c}(\theta)]) (\text{GPES}_{K=c}(\theta) - \mathbb{E}[\text{GPES}_{K=c}(\theta)])^\top \right]) \tag{47}$$

$$= \text{tr}(\mathbb{E}\,\text{GPES}_{K=c}(\theta)\text{GPES}_{K=c}(\theta)^\top) - \text{tr}([\mathbb{E}\,\text{GPES}_{K=c}(\theta)][\mathbb{E}\,\text{GPES}_{K=c}(\theta)]^\top) \tag{48}$$

$$= (\mathbb{E}\,\text{tr}[\text{GPES}_{K=c}(\theta)\text{GPES}_{K=c}(\theta)^\top]) - [\mathbb{E}\,\text{GPES}_{K=c}(\theta)]^\top [\mathbb{E}\,\text{GPES}_{K=c}(\theta)] \tag{49}$$

$$= \mathbb{E}\,\|\text{GPES}_{K=c}(\theta)\|_2^2 - \|\mathbb{E}\,\text{GPES}_{K=c}(\theta)\|_2^2 \tag{50}$$

$$= \mathbb{E}_\tau \mathbb{E}_{\{\epsilon_i\}|\tau=t-1}\,\|\text{GPES}_{K=c}(\theta)\|_2^2 - \|\mathbb{E}\,\text{GPES}_{K=c}(\theta)\|_2^2 \tag{51}$$

$$= \frac{1}{T} \sum_{t=1}^{T} \underbrace{\mathbb{E}_{\{\epsilon_i\}|\tau=t-1}\,\|\text{GPES}_{K=c}(\theta)\|_2^2}_{\text{①}} - \underbrace{\|\mathbb{E}\,\text{GPES}_{K=c}(\theta)\|_2^2}_{\text{②}} \tag{52}$$

Thus to analytically express the trace of covariance, we need to separately derive ① for any $t \in \{1, \ldots, T\}$ and ②.

① **Expressing** $\mathbb{E}_{\{\epsilon_i\}|\tau=t-1}\,\|\text{GPES}_{K=c}(\theta)\|_2^2$.

Here we see that for a given $t \in [T]$, conditioning on $\tau = t - 1$, we have

$$\text{GPES}_{K=c}(\theta) \tag{53}$$

$$= \frac{1}{\sigma^2} \sum_{j=1}^{\lfloor \tau/c \rfloor + 1} \left( \sum_{i=1}^{\lfloor \tau/c \rfloor + 1} \epsilon_i \right) \epsilon_j^\top g_{c,j}^{\tau+1} \tag{54}$$

$$= \frac{1}{\sigma^2} \sum_{j=1}^{\lfloor (t-1)/c \rfloor +1} \left( \sum_{i=1}^{\lfloor (t-1)/c \rfloor +1} \boldsymbol{\epsilon}_i \right) \boldsymbol{\epsilon}_j^\top g_{c,j}^t \tag{55}$$

$$= \frac{1}{\sigma^2} \sum_{j=1}^{\lceil t/c \rceil} \left( \sum_{i=1}^{\lceil t/c \rceil} \boldsymbol{\epsilon}_i \right) \boldsymbol{\epsilon}_j^\top g_{c,j}^t, \tag{56}$$

where in the last step we use the fact that for any integer $t$, $\lfloor (t-1)/c \rfloor + 1 = \lceil t/c \rceil$. Because we are deriving expression ① for every value of $t$ separately, to simplify the notation, we define $n := \lceil t/c \rceil$ and $a_j := g_{c,j}^t$ for a given fixed $t$.

Then the expression in Equation (56) can be simplified as $\frac{1}{\sigma^2} \sum_{j=1}^n \left( \sum_{i=1}^n \boldsymbol{\epsilon}_i \right) \boldsymbol{\epsilon}_j^\top a_j$.

As a result, term ① can be expressed as

$$\mathbb{E}_{\{\boldsymbol{\epsilon}\}|\tau=t-1} \| \text{GPES}_{K=c}(\theta) \|_2^2 \tag{57}$$

$$= \mathbb{E}_{\boldsymbol{\epsilon}} \left\| \frac{1}{\sigma^2} \sum_{j=1}^n \left( \sum_{i=1}^n \boldsymbol{\epsilon}_i \right) \boldsymbol{\epsilon}_j^\top a_j \right\|_2^2 \tag{58}$$

$$= \frac{1}{\sigma^4} \mathbb{E}_{\boldsymbol{\epsilon}} \left[ \sum_{i=1}^n \left( \sum_{k=1}^n \boldsymbol{\epsilon}_k \right) \boldsymbol{\epsilon}_i^\top a_i \right]^\top \left[ \sum_{j=1}^n \left( \sum_{l=1}^n \boldsymbol{\epsilon}_l \right) \boldsymbol{\epsilon}_j^\top a_j \right] \tag{59}$$

$$= \frac{1}{\sigma^4} \sum_{i=1}^n \sum_{j=1}^n a_i^\top \mathbb{E}_{\boldsymbol{\epsilon}} \left[ \boldsymbol{\epsilon}_i \left( \sum_{k=1}^n \boldsymbol{\epsilon}_k \right)^\top \left( \sum_{l=1}^n \boldsymbol{\epsilon}_l \right) \boldsymbol{\epsilon}_j^\top \right] a_j \tag{60}$$

$$\tag{61}$$

From Equation (60), we see that term ① is a quadratic in $\{a_i\}_{i=1}^n$ with each bilinear form's matrix determined by an expectation. Thus we break into different cases to evaluate the expectation $\left[ \sum_{k=1}^n \sum_{l=1}^n \mathbb{E}_{\boldsymbol{\epsilon}} \, \boldsymbol{\epsilon}_i \boldsymbol{\epsilon}_k^\top \, \boldsymbol{\epsilon}_l \boldsymbol{\epsilon}_j^\top \right]$ for different values of $i, j, k, l$.

**Begin of Cases**

**Case (I)** $i = j$.

**(I.1)** If $k \neq i, l \neq i$,

$$\mathbb{E}_{\boldsymbol{\epsilon}} \, \boldsymbol{\epsilon}_i \boldsymbol{\epsilon}_k^\top \, \boldsymbol{\epsilon}_l \boldsymbol{\epsilon}_j^\top$$
$$= \mathbb{E}_{\boldsymbol{\epsilon}_i, \boldsymbol{\epsilon}_k, \boldsymbol{\epsilon}_l} \, \boldsymbol{\epsilon}_i \boldsymbol{\epsilon}_k^\top \, \boldsymbol{\epsilon}_l \boldsymbol{\epsilon}_i^\top$$
$$= \mathbb{E}_{\boldsymbol{\epsilon}_i} \, \mathbb{E}_{\boldsymbol{\epsilon}_k, \boldsymbol{\epsilon}_l} \, \boldsymbol{\epsilon}_i \boldsymbol{\epsilon}_k^\top \, \boldsymbol{\epsilon}_l \boldsymbol{\epsilon}_i^\top$$
$$= \mathbb{E}_{\boldsymbol{\epsilon}_i} \, \boldsymbol{\epsilon}_i [\mathbb{E}_{\boldsymbol{\epsilon}_k, \boldsymbol{\epsilon}_l} \, \boldsymbol{\epsilon}_k^\top \, \boldsymbol{\epsilon}_l] \boldsymbol{\epsilon}_i^\top$$

**(I.1.a)** If $k \neq i, l \neq i$, and $k \neq l$, then $\mathbb{E}_{\boldsymbol{\epsilon}_k, \boldsymbol{\epsilon}_l} \, \boldsymbol{\epsilon}_k^\top \, \boldsymbol{\epsilon}_l = 0$ and $\mathbb{E}_{\boldsymbol{\epsilon}} \, \boldsymbol{\epsilon}_i \boldsymbol{\epsilon}_k^\top \boldsymbol{\epsilon}_l \boldsymbol{\epsilon}_j^\top = \mathbf{0}$.

**(I.1.b)** If $k \neq i, l \neq i$, and $k = l$.

$$\mathbb{E}_{\boldsymbol{\epsilon}_i} \, \boldsymbol{\epsilon}_i [\mathbb{E}_{\boldsymbol{\epsilon}_k, \boldsymbol{\epsilon}_l} \, \boldsymbol{\epsilon}_k^\top \, \boldsymbol{\epsilon}_l] \boldsymbol{\epsilon}_i^\top$$
$$= \mathbb{E}_{\boldsymbol{\epsilon}_i} \, \boldsymbol{\epsilon}_i [\mathbb{E}_{\boldsymbol{\epsilon}_k} \, \boldsymbol{\epsilon}_k^\top \, \boldsymbol{\epsilon}_k] \boldsymbol{\epsilon}_i^\top$$
$$= (d\sigma^2) \sigma^2 I_d$$
$$= d\sigma^4 I_d$$

**(I.2)** If $k = i, l \neq i$,

$$\mathbb{E}_{\boldsymbol{\epsilon}} \, \boldsymbol{\epsilon}_i \boldsymbol{\epsilon}_k^\top \, \boldsymbol{\epsilon}_l \boldsymbol{\epsilon}_j^\top$$
$$= \mathbb{E}_{\boldsymbol{\epsilon}} \, \boldsymbol{\epsilon}_i \boldsymbol{\epsilon}_i^\top \, \boldsymbol{\epsilon}_l \boldsymbol{\epsilon}_i^\top$$
$$= \mathbb{E}_{\boldsymbol{\epsilon}_i} \, \boldsymbol{\epsilon}_i \boldsymbol{\epsilon}_i^\top \, \mathbb{E}_{\boldsymbol{\epsilon}_l} [\boldsymbol{\epsilon}_l] \boldsymbol{\epsilon}_i^\top$$
$$= \mathbf{0}$$

**(I.3)** If $k \neq i, l = i$, similarly as **(I.2)**,

$$\mathbb{E}_{\boldsymbol{\epsilon}} \, \boldsymbol{\epsilon}_i \boldsymbol{\epsilon}_k^\top \, \boldsymbol{\epsilon}_l \boldsymbol{\epsilon}_j^\top$$
$$= \mathbb{E}_{\boldsymbol{\epsilon}} \, \boldsymbol{\epsilon}_i \boldsymbol{\epsilon}_k^\top \, \boldsymbol{\epsilon}_i \boldsymbol{\epsilon}_i^\top$$
$$= \mathbb{E}_{\boldsymbol{\epsilon}_i} \, \boldsymbol{\epsilon}_i \, \mathbb{E}_{\boldsymbol{\epsilon}_k} [\boldsymbol{\epsilon}_k^\top] \boldsymbol{\epsilon}_i \boldsymbol{\epsilon}_i^\top$$
$$= \mathbf{0}$$

**(I.4)** If $k = i, l = i$, by Isserlis' theorem (derivation see Supplementary material A.2 in [29]),

$$\mathbb{E}_{\boldsymbol{\epsilon}} \, \boldsymbol{\epsilon}_i \boldsymbol{\epsilon}_k^\top \, \boldsymbol{\epsilon}_l \boldsymbol{\epsilon}_j^\top$$
$$= \mathbb{E}_{\boldsymbol{\epsilon}_i} \, \boldsymbol{\epsilon}_i \boldsymbol{\epsilon}_i^\top \, \boldsymbol{\epsilon}_i \boldsymbol{\epsilon}_i^\top$$
$$= (d + 2)\sigma^4 I_d$$

Combining **(I.1)** to **(I.4)**, we see that for the case of $i = j$,

$$\sum_{k=1}^n \sum_{l=1}^n \mathbb{E}_{\boldsymbol{\epsilon}} \, \boldsymbol{\epsilon}_i \boldsymbol{\epsilon}_k^\top \, \boldsymbol{\epsilon}_l \boldsymbol{\epsilon}_j^\top = \left[ \sum_{k \neq i} d\sigma^4 I_d \right] + (d + 2)\sigma^4 I_d = (nd + 2)\sigma^4 I_d \tag{62}$$

**Case (II)** $i \neq j$.

**(II.1)** $k \neq i, k \neq j$.

**(II.1.a)** If $k \neq i, k \neq j$, and additionally $k \neq l$,

$$\mathbb{E}_{\boldsymbol{\epsilon}} \, \boldsymbol{\epsilon}_i \boldsymbol{\epsilon}_k^\top \, \boldsymbol{\epsilon}_l \boldsymbol{\epsilon}_j^\top$$
$$= \mathbb{E}_{\boldsymbol{\epsilon}_i, \boldsymbol{\epsilon}_j, \boldsymbol{\epsilon}_l} \, \boldsymbol{\epsilon}_i (\mathbb{E}_{\boldsymbol{\epsilon}_k} [\boldsymbol{\epsilon}_k])^\top \boldsymbol{\epsilon}_l \boldsymbol{\epsilon}_j^\top$$
$$= \mathbf{0}$$

**(II.1.b)** If $k \neq i, k \neq j$, and $k = l$,

After pulling out $\mathbb{E}_{\boldsymbol{\epsilon}_k} [\boldsymbol{\epsilon}_k^\top \boldsymbol{\epsilon}_k]$, because $i \neq j$, we still have the expectation $= 0$.

$$\mathbb{E}_{\boldsymbol{\epsilon}} \, \boldsymbol{\epsilon}_i \boldsymbol{\epsilon}_k^\top \, \boldsymbol{\epsilon}_l \boldsymbol{\epsilon}_j^\top$$
$$= \mathbb{E}_{\boldsymbol{\epsilon}} \, \boldsymbol{\epsilon}_i \, \mathbb{E}[\boldsymbol{\epsilon}_k^\top \boldsymbol{\epsilon}_k] \boldsymbol{\epsilon}_j^\top$$
$$= [\mathbb{E}_{\boldsymbol{\epsilon}_i, \boldsymbol{\epsilon}_j} \, \boldsymbol{\epsilon}_i \boldsymbol{\epsilon}_j^\top] \cdot \mathbb{E}[\boldsymbol{\epsilon}_k^\top \boldsymbol{\epsilon}_k]$$
$$= \mathbf{0} \cdot \mathbb{E}[\boldsymbol{\epsilon}_k^\top \boldsymbol{\epsilon}_k]$$
$$= \mathbf{0}$$

**(II.2)** $k = i$,

**(II.2.a)** if $l \neq i, l \neq j$. This is similar to **(II.1)** as we can swap the position of $k$ and $j$:

$$\boldsymbol{\epsilon}_i \boldsymbol{\epsilon}_k^\top \, \boldsymbol{\epsilon}_l \boldsymbol{\epsilon}_j^\top = \boldsymbol{\epsilon}_i \boldsymbol{\epsilon}_l^\top \, \boldsymbol{\epsilon}_k \boldsymbol{\epsilon}_j^\top.$$

Thus we have $\mathbb{E}_{\boldsymbol{\epsilon}} \, \boldsymbol{\epsilon}_i \boldsymbol{\epsilon}_k^\top \, \boldsymbol{\epsilon}_l \boldsymbol{\epsilon}_j^\top = \mathbf{0}$.

**(II.2.b)** if $l = i$,

$$\mathbb{E}_{\boldsymbol{\epsilon}} \, \boldsymbol{\epsilon}_i \boldsymbol{\epsilon}_k^\top \, \boldsymbol{\epsilon}_l \boldsymbol{\epsilon}_j^\top$$
$$= E_{\boldsymbol{\epsilon}} \, \boldsymbol{\epsilon}_i \boldsymbol{\epsilon}_i^\top \, \boldsymbol{\epsilon}_i \boldsymbol{\epsilon}_j^\top$$
$$= E_{\boldsymbol{\epsilon}_i} \, \boldsymbol{\epsilon}_i \boldsymbol{\epsilon}_i^\top \, \boldsymbol{\epsilon}_i \, \mathbb{E}_{\boldsymbol{\epsilon}_j} \, \boldsymbol{\epsilon}_j^\top$$
$$= \mathbf{0}$$

**(II.2.c)** if $l = j$,

$$\mathbb{E}_{\boldsymbol{\epsilon}} \, \boldsymbol{\epsilon}_i \boldsymbol{\epsilon}_k^\top \, \boldsymbol{\epsilon}_l \boldsymbol{\epsilon}_j^\top$$
$$= \mathbb{E}_{\boldsymbol{\epsilon}_i, \boldsymbol{\epsilon}_j} \, \boldsymbol{\epsilon}_i \boldsymbol{\epsilon}_i^\top \, \boldsymbol{\epsilon}_j \boldsymbol{\epsilon}_j^\top$$
$$= \mathbb{E}_{\boldsymbol{\epsilon}_i} [\boldsymbol{\epsilon}_i \boldsymbol{\epsilon}_i^\top] \, \mathbb{E}_{\boldsymbol{\epsilon}_j} [\boldsymbol{\epsilon}_j \boldsymbol{\epsilon}_j^\top]$$
$$= \sigma^4 I_d$$

**(II.3)** $k = j$,

**(II.3.a)** if $l \neq i, l \neq j$. Again similar to **(II.1)** by swapping $k$ and $j$, we have $\mathbb{E}_{\boldsymbol{\epsilon}}\, \boldsymbol{\epsilon}_i \boldsymbol{\epsilon}_k^\top \boldsymbol{\epsilon}_l \boldsymbol{\epsilon}_j^\top = \mathbf{0}$.

**(II.3.b)** if $l = i$. By swapping the position, we have

$$\mathbb{E}_{\boldsymbol{\epsilon}}\, \boldsymbol{\epsilon}_i \boldsymbol{\epsilon}_k^\top \boldsymbol{\epsilon}_l \boldsymbol{\epsilon}_j^\top$$
$$= \mathbb{E}_{\boldsymbol{\epsilon}_i, \boldsymbol{\epsilon}_j}\, \boldsymbol{\epsilon}_i (\boldsymbol{\epsilon}_j^\top \boldsymbol{\epsilon}_i) \boldsymbol{\epsilon}_j^\top$$
$$= \mathbb{E}_{\boldsymbol{\epsilon}_i, \boldsymbol{\epsilon}_j}\, \boldsymbol{\epsilon}_i (\boldsymbol{\epsilon}_i^\top \boldsymbol{\epsilon}_j) \boldsymbol{\epsilon}_j^\top$$
$$= \mathbb{E}_{\boldsymbol{\epsilon}_i}[\boldsymbol{\epsilon}_i \boldsymbol{\epsilon}_i^\top]\, \mathbb{E}_{\boldsymbol{\epsilon}_j}[\boldsymbol{\epsilon}_j \boldsymbol{\epsilon}_j^\top]$$
$$= \sigma^4 I_d$$

**(II.3.c)** $l = j$.

$$\mathbb{E}_{\boldsymbol{\epsilon}}\, \boldsymbol{\epsilon}_i \boldsymbol{\epsilon}_k^\top \boldsymbol{\epsilon}_l \boldsymbol{\epsilon}_j^\top$$
$$= \mathbb{E}_{\boldsymbol{\epsilon}}\, \boldsymbol{\epsilon}_i \boldsymbol{\epsilon}_j^\top \boldsymbol{\epsilon}_j \boldsymbol{\epsilon}_j^\top$$
$$= \mathbb{E}_{\boldsymbol{\epsilon}_i}\, \boldsymbol{\epsilon}_i\, \mathbb{E}_{\boldsymbol{\epsilon}_j}\, \boldsymbol{\epsilon}_j^\top \boldsymbol{\epsilon}_j \boldsymbol{\epsilon}_j^\top$$
$$= \mathbf{0}$$

As a result, when we have $i \neq j$, the total sum over all the cases **(II.1)** - **(II.3)** is

$$\sum_{k=1}^{n}\sum_{l=1}^{n} \mathbb{E}_{\boldsymbol{\epsilon}}\, \boldsymbol{\epsilon}_i \boldsymbol{\epsilon}_k^\top \boldsymbol{\epsilon}_l \boldsymbol{\epsilon}_j^\top = 2\sigma^4 I_d \tag{63}$$

**End of all Cases**.

Using the result in (62) and (63), we see that for the fixed time step $t$ and $n := \lceil t/c \rceil$, we have

$$\mathbb{E}_{\{\boldsymbol{\epsilon}\}|\tau=t-1}\, \|\text{GPES}_{K=c}(\theta)\|_2^2 \tag{64}$$

$$= \frac{1}{\sigma^4} \left( \sum_{i=1}^{n}(nd+2)\sigma^4\|a_i\|_2^2 + \sum_{i\neq j} 2\sigma^4 a_i^\top a_j \right) \tag{65}$$

$$= \sum_{i=1}^{n}(nd+2)\|a_i\|_2^2 + \sum_{i\neq j} 2a_i^\top a_j \tag{66}$$

$$= \left( \sum_{i=1}^{n}(d+2)\|a_i\|_2^2 + \sum_{i=1}^{n}(n-1)d\|a_i\|_2^2 \right) + \left( \sum_{i\neq j}(d+2)[a_i]^\top a_j - \sum_{i\neq j} da_i^\top a_j \right) \tag{67}$$

$$= \left( \sum_{i=1}^{n}(d+2)\|a_i\|_2^2 + \sum_{i\neq j}(d+2)[a_i]^\top a_j \right) + \left( \sum_{i=1}^{n}(n-1)d\|a_i\|_2^2 - \sum_{i\neq j} d[a_i]^\top a_j \right) \tag{68}$$

$$= (d+2)\left\|\sum_{i=1}^{n} a_i\right\|_2^2 + \frac{d}{2}\left( \sum_{i=1}^{n} 2(n-1)\|a_i\|_2^2 - \sum_{i\neq j} 2a_i^\top a_j \right) \tag{69}$$

$$= (d+2)\left\|\sum_{i=1}^{n} a_i\right\|_2^2 + \frac{d}{2}\left( \sum_{i=1}^{n}(n-1)\|a_i\|_2^2 + \sum_{j=1}^{n}(n-1)\|a_j\|_2^2 - \sum_{i\neq j} 2a_i^\top a_j \right) \tag{70}$$

$$= (d+2)\left\|\sum_{i=1}^{n} a_i\right\|_2^2 + \frac{d}{2}\left( \sum_{i=1}^{n}\sum_{j\neq i}\|a_i\|_2^2 + \sum_{j=1}^{n}\sum_{i\neq i}\|a_j\|_2^2 - \sum_{i\neq j} 2a_i^\top a_j \right) \tag{71}$$

$$= (d+2)\left\|\sum_{i=1}^{n} a_i\right\|_2^2 + \frac{d}{2}\left( \sum_{i=1,j=1,i\neq j}^{n}\|a_i\|_2^2 + \|a_j\|_2^2 - 2a_i^\top a_j \right) \tag{72}$$

$$= (d+2)\left\|\sum_{i=1}^{n} a_i\right\|_2^2 + \frac{d}{2}\sum_{i=1,j=1,i\neq j}^{n}\|a_i - a_j\|_2^2 \tag{73}$$

Now we substitute $a_i$ with $g_{c,i}^t$ and $n$ back with $\lceil t/c \rceil$, we have

$$\mathbb{E}_{\{\boldsymbol{\epsilon}\}|\tau=t-1}\, \|\text{GPES}_{K=c}(\theta)\|_2^2 \tag{74}$$

$$=(d+2)\left\|\sum_{i=1}^{\lceil t/c\rceil} g_{c,i}^t\right\|_2^2 + \frac{d}{2}\sum_{i=1,j=1,i\neq j}^{\lceil t/c\rceil}\|g_{c,i}^t - g_{c,j}^t\|_2^2 \tag{75}$$

$$=(d+2)\|g^t\|_2^2 + \frac{d}{2}\sum_{j=1,j'=1}^{\lceil t/c\rceil}\|g_{c,j}^t - g_{c,j'}^t\|_2^2 \tag{76}$$

This completes the derivation for ①.

② **Expressing** $\|\mathbb{E}\,\text{GPES}_{K=c}(\theta)\|_2^2$

This amounts to computing the expectation of the $\text{GPES}_{K=c}(\theta)$ estimator:

$$\mathbb{E}\,\text{GPES}_{K=c}(\theta) \tag{77}$$

$$=\mathbb{E}_{\boldsymbol{\tau}}\,\mathbb{E}_{\boldsymbol{\epsilon}}\,\text{GPES}_{K=c}(\theta) \tag{78}$$

$$=\frac{1}{T}\sum_{t=1}^{T}\mathbb{E}_{\boldsymbol{\epsilon}|\boldsymbol{\tau}=t-1}\,\text{GPES}_{K=c}(\theta) \tag{79}$$

$$=\frac{1}{T}\sum_{t=1}^{T}\frac{1}{\sigma^2}\mathbb{E}_{\boldsymbol{\epsilon}}\sum_{j=1}^{\lceil t/c\rceil}\left(\sum_{i=1}^{\lceil t/c\rceil}\boldsymbol{\epsilon}_i\right)\boldsymbol{\epsilon}_j^\top g_{c,j}^t \tag{80}$$

$$=\frac{1}{T}\sum_{t=1}^{T}\frac{1}{\sigma^2}\sum_{j=1}^{\lceil t/c\rceil}\sum_{i=1}^{\lceil t/c\rceil}\mathbb{E}_{\boldsymbol{\epsilon}}[\boldsymbol{\epsilon}_i\boldsymbol{\epsilon}_j^\top]g_{c,j}^t \tag{81}$$

$$=\frac{1}{T}\sum_{t=1}^{T}\frac{1}{\sigma^2}\sum_{j=1}^{\lceil t/c\rceil}\sigma^2 g_{c,j}^t \tag{82}$$

$$=\frac{1}{T}\sum_{t=1}^{T}\sum_{j=1}^{\lceil t/c\rceil} g_{c,j}^t \tag{83}$$

$$=\frac{1}{T}\sum_{t=1}^{T} g^t \tag{84}$$

Thus we have

$$\|\mathbb{E}\,\text{GPES}_{K=c}(\theta)\|_2^2 = \left\|\frac{1}{T}\sum_{t=1}^{T} g^t\right\|_2^2. \tag{85}$$

This completes the derivation for ②.

Combining the result we have for ① (Equation (76)) and ② (Equation (85)), we have

$$\text{tr}(\text{Cov}[\text{GPES}_{K=c}(\theta)]) \tag{86}$$

$$=\frac{1}{T}\sum_{t=1}^{T}\mathbb{E}_{\{\boldsymbol{\epsilon}_i\}|\boldsymbol{\tau}=t-1}\|\text{GPES}_{K=c}(\theta)\|_2^2 - \|\mathbb{E}\,\text{GPES}_{K=c}(\theta)\|_2^2 \tag{87}$$

$$=\frac{(d+2)}{T}\sum_{t=1}^{T}\left(\|g^t\|_2^2\right) - \left\|\frac{1}{T}\sum_{t=1}^{T} g^t\right\|_2^2 + \frac{1}{T}\sum_{t=1}^{T}\left(\frac{d}{2}\sum_{j=1,j'=1}^{\lceil t/c\rceil}\|g_{c,j}^t - g_{c,j'}^t\|_2^2\right) \tag{88}$$

This completes the proof for Theorem 5. $\qquad\square$

### D.5 Proof of Corollary 6

**Corollary 6.** *Under Assumption 2, when $W = 1$, the gradient estimator $\text{GPES}_{K=T}(\theta)$ has the smallest total variance among all $\{\text{GPES}_K : K \in [T]\}$ estimators.*

*Proof.* We notice that in Equation 88, the only term that depends on $c$ is the third term:

$$\frac{1}{T}\sum_{t=1}^{T}\left(\frac{d}{2}\sum_{j=1,j'=1}^{\lceil t/c\rceil}\|g_{c,j}^t - g_{c,j'}^t\|_2^2\right). \tag{89}$$

This term can be made zero by having $c = T$: in this case, $\lceil t/T \rceil = 1$ for any $t \in [T]$, hence there is only one $g_{T,1}^t$ to compare against itself. Thus having $c = T$ minimizes the total variance among the class of $\text{GPES}_{K=c}$ estimators. $\qquad \square$

### D.6 Proof of Corollary 8

**Corollary 7.** *Under Assumption 2, when $W$ divides $T$, the NRES gradient estimator has the smallest total variance among all $\text{GPES}_{K=cW}$ estimators $c \in \mathbb{Z} \cap [1, T/W]$.*

*Proof.* Here the key idea is that because $W$ divides $T$, we can define a mega unrolled computation graph and apply Corollary 6.

Specifically, we consider a mega UCG where the inner state is represented by all the states within a size $W$ truncation window. The total horizon length of this mega UCG is $T' = T/W$. The initial state of the mega UCG is given by

$$S_0 := (s_0, \ldots, s_0) \in \mathbb{R}^{Wp}. \tag{90}$$

For time step $t' \in \{1, \ldots, T/W\}$ in this mega UCG, the mega-state $S_{t'}$ is given by the concatenation of all the states in a given original truncation window

$$S_{t'} := (s_{(t'-1) \cdot W+1}, \ldots, s_{t' \cdot W}) \in \mathbb{R}^{Wp}, \tag{91}$$

The learnable parameter in this problem is still $\theta \in \mathbb{R}^d$. The transition dynamics $u_{t'} : \mathbb{R}^{Wp} \times \mathbb{R}^d \to \mathbb{R}^{Wp}$ works as follows:

$$u_{t'} : \left( S_{t'} = \begin{bmatrix} s_{(t'-1) \cdot W+1} \\ s_{(t'-1) \cdot W+2} \\ \vdots \\ s_{t' \cdot W} \end{bmatrix}, \theta \right) \mapsto S_{t'+1} := \begin{bmatrix} f_{t'W+1}(s_{t' \cdot W}, \theta) \\ f_{t'W+2}(f_{t'W+1}(s_{t' \cdot W}, \theta), \theta) \\ \vdots \\ f_{(t'+1)W}(\ldots(f_{t'W+2}(f_{t'W+1}(s_{t' \cdot W}, \theta), \theta) \ldots, \theta). \end{bmatrix} \tag{92}$$

Here, the mega transition only look at the last time step's original state in $S_{t'}$ and unroll it forward $W$ steps to form the next mega state.

The mega loss function for step $t'$ is defined as $\ell_{t'}^s : \mathbb{R}^{Wp} \to \mathbb{R}$:

$$\ell_{t'}^s((s_{(t'-1) \cdot W+1}, \ldots, s_{t' \cdot W})) = \frac{1}{W} \sum_{i=1}^{W} L_{(t'-1)W+i}^s(s_{(t'-1)W+i}), \tag{93}$$

which simply averages each original inner state's loss within the truncation window.

Here, when we consider a truncation window of size $W$ in the original graph, it is equivalent to a truncation window of size 1 in the mega UCG. To apply Corollary 6 to this graph, we only need to make sure Assumption 2 holds for this mega graph. Here we see that, by Assumption 2 on this original graph, we have

$$\ell_{t'}(\theta + v_1, \ldots, \theta + v_{t'}) - \ell_{t'}(\theta - v_1, \ldots, \theta - v_{t'})$$

$$= \frac{1}{W} \sum_{j=1}^{W} \left[ L_{W(t'-1)+j}([\theta + v_1]_{\times W}, \ldots, [\theta + v_{t'}]_{\times j}) - L_{W(t'-1)+j}([\theta - v_1]_{\times W}, \ldots, [\theta - v_{t'}]_{\times j}) \right]$$

$$= \frac{1}{W} \sum_{j=1}^{W} 2 \sum_{i'=1}^{t'} [v_{i'}]^\top g_{W,i'}^{W(t'-1)+j}$$

$$= 2 \sum_{i'=1}^{t'} [v_{i'}]^\top \left( \frac{1}{W} \sum_{j=1}^{W} g_{W,i'}^{W(t'-1)+j} \right)$$

Thus for this mega graph, the set of vectors satisfying Assumption 2 is $g_{i'}^{t'} = \left( \frac{1}{W} \sum_{j=1}^{W} g_{W,i'}^{W(t'-1)+j} \right)$.

With the mega graph satisfying the assumption in Corollary 6, we see that when the truncation window is of size 1 in the mega graph, the gradient estimator NRES on this mega graph has smaller

trace of covariance than other $\text{GPES}_K = c$ estimators on this mega graph. Here, running NRES on the mega-graph is equivalent to runing NRES on the original graph, while the gradient estimator $\text{GPES}_{K=c}$ on the mega-graph is equivalent to the gradient estimator $\text{GPES}_{K=cW}$ on the original UCG. Thus we have proved that the NRES gradient estimator on the original graph has the smallest total variance among all $\text{GPES}_{K=cW}$ estimators, thus completing the proof.

$\square$

### D.7 Proof of Theorem 9

**Theorem 8.** *Under Assumption 2, for any $W$ that divides $T$, if*

$$\sum_{k=1}^{T/W} \left\| \sum_{t=W\cdot(k-1)+1}^{W\cdot k} g^t \right\|_2^2 \le \frac{d+1}{d+2} \left\| \sum_{j=1}^{T/W} \sum_{t=W\cdot(k-1)+1}^{W\cdot k} g^t \right\|_2^2, \tag{94}$$

*then* $\text{tr}(\text{Cov}(\frac{1}{T/W} \sum_{i=1}^{T/W} \text{NRES}_i(\theta)) \le \text{tr}(\text{Cov}(\text{FullES}(\theta))$ *where* $\text{NRES}_i(\theta)$ *are iid NRES estimators.*

*Proof.* We first analytically express the FullES gradient estimator. Under the Assumption 2,

$$\text{FullES}(\theta) \tag{95}$$

$$= \frac{1}{2\sigma^2} \left[ L([\theta + \epsilon]_{\times T}) - L([\theta - \epsilon]_{\times T}) \right] \epsilon \tag{96}$$

$$= \frac{1}{\sigma^2} \epsilon \epsilon^\top \left( \frac{1}{T} \sum_{t=1}^{T} g^t \right) \tag{97}$$

Because $\mathbb{E}_\epsilon \, \epsilon \epsilon^T \epsilon \epsilon^T = (d+2)\sigma^2 I_{d\times d}$, we can see that

$$\text{tr}(\text{Cov}[\text{FullES}(\theta)]) = (d+2) \left\| \frac{1}{T} \sum_{t=1}^{T} g^t \right\|_2^2 - \left\| \frac{1}{T} \sum_{t=1}^{T} g^t \right\|_2^2 \tag{98}$$

$$= (d+1) \left\| \frac{1}{T} \sum_{t=1}^{T} g^t \right\|_2^2 \tag{99}$$

From Theorem 5 and Corollary 8, we can see that the trace of covariance for a single NRES when $W \ge 1$ is given by

$$\text{tr}(\text{Cov}(\text{NRES}(\theta))) = \frac{(d+2)}{T/W} \sum_{k=1}^{T/W} \left( \left\| \frac{1}{W} \sum_{t=W\cdot(k-1)+1}^{W\cdot k} g^t \right\|_2^2 \right) - \left\| \frac{1}{T} \sum_{t=1}^{T} g^t \right\|_2^2 \tag{100}$$

When we average over $T/W$ *i.i.d.* NRES workers, the trace of covariance of the average is scaled by $\frac{1}{T/W}$:

$$\text{tr}(\text{Cov}(\frac{1}{T/W} \sum_{i=1}^{T/W} \text{NRES}_i(\theta)) \tag{101}$$

$$= \frac{1}{T/W} \text{tr}(\text{Cov}(\text{NRES}(\theta))) \tag{102}$$

$$= \frac{1}{T/W} \frac{(d+2)}{T/W} \frac{1}{W^2} \sum_{k=1}^{T/W} \left( \left\| \sum_{t=W\cdot(k-1)+1}^{W\cdot k} g^t \right\|_2^2 \right) - \frac{1}{T/W} \left\| \frac{1}{T} \sum_{t=1}^{T} g^t \right\|_2^2 \tag{103}$$

$$\le \frac{(d+2)}{T^2} \sum_{k=1}^{T/W} \left( \left\| \sum_{t=W\cdot(k-1)+1}^{W\cdot k} g^t \right\|_2^2 \right) \tag{104}$$

$$\leq \frac{(d+2)}{T^2} \frac{d+1}{d+2} \left\| \sum_{k=1}^{T/W} \sum_{t=W\cdot(k-1)+1}^{W\cdot k} g^t \right\|_2^2 \quad \text{(using the condition in Theorem 9)} \tag{105}$$

$$= \frac{(d+1)}{T^2} \left\| \sum_{k=1}^{T/W} \sum_{t=W\cdot(k-1)+1}^{W\cdot k} g^t \right\|_2^2 \tag{106}$$

$$= \frac{(d+1)}{T^2} \left\| \sum_{t=1}^{T} g^t \right\|_2^2 \tag{107}$$

$$= (d+1) \left\| \frac{1}{T} \sum_{t=1}^{T} g^t \right\|_2^2 \tag{108}$$

$$= \mathrm{tr}(\mathrm{Cov}[\mathrm{FullES}(\theta)]) \tag{109}$$

This completes the proof.

$\square$

## E  Experiments

In this section, we first provide additional experiment results for each of the three applications we consider in the main paper. We next describe the experiment details and hyperparameters used for these experiments. We finally describe the computation resources needed to run these experiments and our implementation.

### E.1  Additional Experiment Results

#### E.1.1  Learning dynamical system parameters

**Visualizing the Lorenz system parameter learning loss surface.**  We have plotted the training loss surface of the Lorenz system parameter learning problem in the left panel of Figure 5(a) in the main paper. Here we provide a larger version of the same figure in Figure 8(a). In addition, we plot the losses along the line segment connecting the groundtruth $\theta_{\mathrm{gt}}$ and $\theta_{\mathrm{init}}$ in Figure 8(b). (Notice that this is just a demonstration of the sensitivity of the loss surface; the optimization of $\theta$ are not constrained to this line segment.)  We see that the loss surface have many local fluctuations and suboptimal local minima. In this case, the direction of the negative gradient is often non-informative for global optimization — it would frequently point in a direction opposite to the global direction of loss decrease.

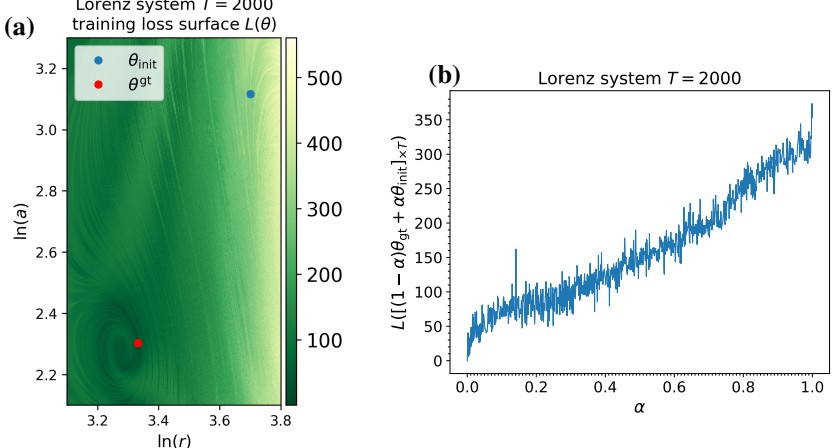

**Figure 8:** (a) The extremely locally sensitive training loss surface in the Lorenz system parameter learning problem. (b) The training loss on the line segment in the parameter space connecting the groundtruth $\theta_{\mathrm{gt}}$ and the initialization $\theta_{\mathrm{init}}$. Because of the high sharpness and existence of many suboptimal local minima, automatic differentiation methods are ineffective to optimize this loss.

**Measuring progress on the Lorenz system's loss surface.** As we have seen in Figure 8 that the training loss surface is highly non-smooth, it is difficult to visually compare different gradient estimation methods' performance through their non-smoothed training loss convergences as the losses have significant fluctuation for all methods. Instead, we measure the test loss (denoted by `loss` in Figure 5(b)) instead of the non-smoothed training loss by sampling the random initial state $s_0 \sim \mathcal{N}((1.2, 1.3, 1.6), 0.01\, I_{3\times3})$. Because this test loss considers a distribution of initial states, it is much smoother and helps with better visual comparisons. Besides, the test loss is a better indicator of predictive generalization to novel initial state conditions.

**AD methods perform worse than ES methods.** In Figure 5(b) in the main paper, we have shown that NRES outperforms other evolution strategies baselines on the Lorenz system learning task. Here we additionally include the performance of four popular automatic differentiation methods BPTT, TBPTT, UORO, and DODGE (discussed in Section B in the Appendix) on the same task in Figure 9. We notice that these 4 AD methods all perform worse than the 4 ES methods. Thus, NRES is still the best among all the methods considered in this paper.

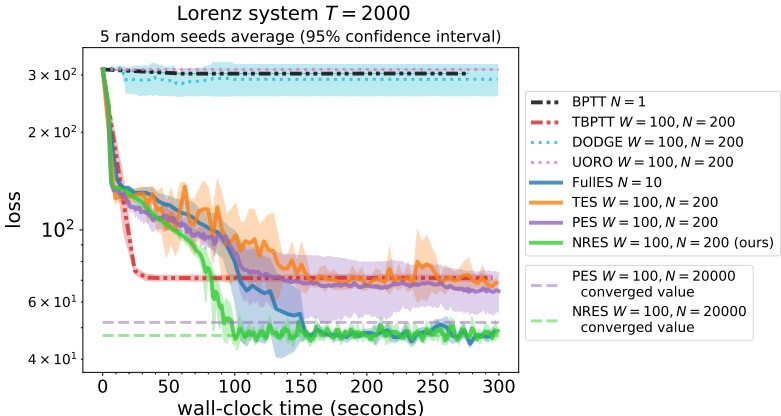

**Figure 9:** Different automatic differentiation and evolution strategies methods' loss convergence on the Lorenz system parameter learning problem. AD methods (BPTT, TBPTT, UORO, DODGE) all perform worse than ES methods, justifying the need for ES methods on problems with chaotic loss surfaces. Among all the methods, our proposed method NRES converges the fastest.

### E.1.2 Meta-training learned optimizers

**AD methods perform worse than ES methods.** In Figure 6(b) in the main paper, we have shown that NRES outperforms other ES baselines on the task to meta-learn a learned optimizer model to train a 3-layer MultiLayer Perceptron (MLP) on the Fashion MNIST dataset. Here we additionally include the performance of four popular automatic differentiation methods BPTT, TBPTT, UORO, and DODGE (discussed in Section B in the Appendix) on the same task in Figure 10. It is worth noting that we focus on the training loss range $[0.5, \ln(10)]$ in Figure 6(a) in the main paper to make the comparisons among different ES methods more visually obvious. ($\ln(10)$ is chosen because random guessing on Fashion MNIST yields a validation loss of $\ln(10)$.) However, the non-smoothed training loss of the initialization $\theta_{\text{init}}$ for this problem is at around $84.4$. Thus we show the loss range of $[0.5, 100]$ on the y-axis in Figure 10. Regardless of the y-axis range choice, our proposed method NRES performs the best among all the gradient estimation methods considered in this paper.

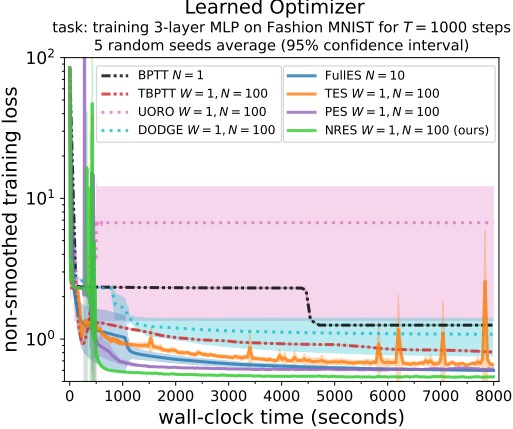

**Figure 10:** Different automatic differentiation and evolution strategies methods' loss convergence on the meta-learning a learned optimizer to train a 3-layer MLP on Fashion MNIST for $T = 1000$ steps. AD methods (BPTT, TBPTT, UORO, DODGE) all perform worse than ES methods, justifying the need for ES methods on problems with such highly-sensitive loss surfaces. Among all the methods, our proposed method NRES converges the fastest. We notice that the two best performing methods PES and NRES experience some loss fluctuation right below 500 seconds. Because such fluctuation never occurs for any other tasks we experiment with, we believe this is a property of this specific application itself but not the problem of the two methods.

**Results on the learned optimizer task with a higher dimension.** In Section 5.2, the learned optimizer we have considered has a parameter dimension $d = 1762$. Here we compare ES gradient estimators on the same meta-training task but with an $11\times$ larger learnable parameter ($\theta$) dimension in Figure 11(a). To increase the parameter dimension, we increase the width of the multilayer perceptron used by the learned optimizer. On this higher-dimensional problem, NRES can still provide a $3.8\times$ wall clock time speed up over PES and a $9.7\times$ wall clock time speed up over FullES to reach a loss value which NRES reaches early on during the meta-training.

**Results on the learned optimizer task with a longer horizon.** In Section 5.2, the learned optimizer we have considered has an inner-problem training steps of $T = 1000$. Here we make the problem horizon $10\times$ longer and show the training convergence of different ES estimators in Figure 11(b). For this problem, NRES still achieves a $2.1\times$ wall clock speed up over PES and a $3.5\times$ speed up over FullES to reach a loss value which NRES reaches early on during the meta-training.

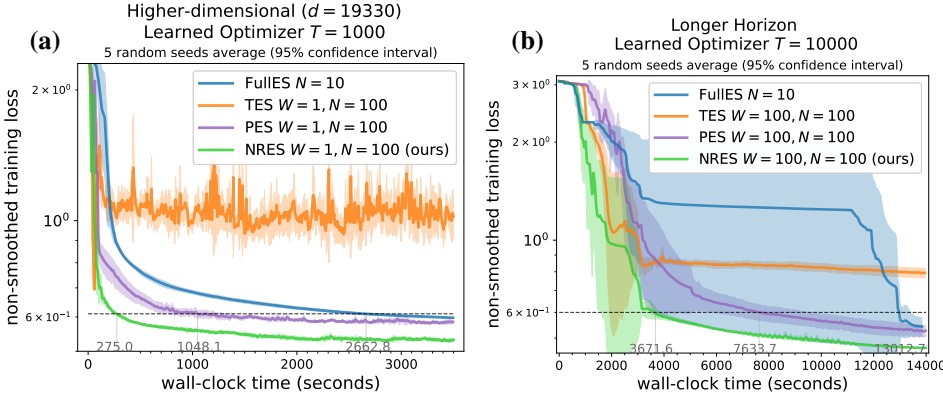

**Figure 11:** (a) Comparing different ES gradient estimators' training loss convergence (in wall-clock time) on a $11\times$ higher-dimensional learned optimizer task ($d = 19330$) than Figure 6(a) in the main paper. (b) Comparing different ES gradient estimators' training loss convergence (in wall-clock time) on a learned optimizer task with $10\times$ longer horizon $T = 10000$. For both cases NRES still converges much faster than other methods.

**Results on another learned optimizer task.** In addition to the learned optimizer task shown in the main paper, we experiment with another task from the training task distribution of VeLO [20] to further compare the performance of different evolution strategies methods. Here we meta-learn the learned optimizer model from [3] to train a 4-layer fully Convolutional Neural Network on CIFAR-10 [44]. As shown in Figure 12, we see that TES fails to converge to to the same loss level as the other three methods. Among the rest of the methods, NRES can achieve a speed up of more than $2\times$ and

$5\times$ over PES and FullES respectively to reach the non-smoothed training loss of $1.55$ given perfect parallel implementations of each method.

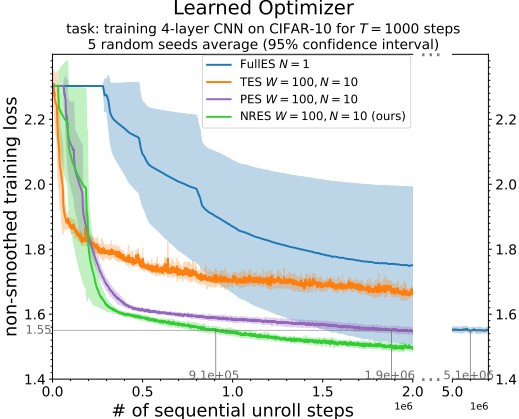

**Figure 12:** Different automatic differentiation and evolution strategies methods' loss convergence on the meta-learning learned optimizer to train a 4-layer fully convolutional neural network on CIFAR-10 for $T = 1000$ steps. Because FullES takes a lot more sequential steps than PES and NRES to reach the loss value of $1.55$, we make the x-axis skip the values between $[2.0 \times 10^6, 5.0 \times 10^6]$ to show the number of steps FullES reaches the same value. Among all the ES methods, our proposed method NRES converges the fastest by using the smallest number of sequential unroll steps to reach the same loss value.

### E.1.3  Reinforcement Learning

**Total number of environment steps to solve the Mujoco tasks.**  We show in Figure 7 that NRES can solve the two Mujoco tasks using the least number of sequential environment steps, which indicates NRES can solve the tasks using the shortest amount of wall clock time under perfect parallelization. Here we additionally show the *total* number of environment steps used by each ES methods to solve the two tasks in Table 3. We see that NRES uses the least total number of environment steps and thus is the most sample efficient among the methods compared.

**Table 3:** Total number of environment steps used by each ES method considered in this paper. TES is unable to solve either task, while PES struggles to solve the Half Cheetah task despite we allowing it to use a significantly larger number of environment steps. In contrast, NRES (ours) solve both tasks using the least number of total environment steps, being 26% and 22% more sample efficient than FullES for the two tasks respectively.

| | total number of environment steps | | | |
| | used to solve the Mujoco task | | | |
| | (averaged over 5 random seeds) | | | |
| Mujoco task | FullES | TES | PES | NRES (ours) |
|---|---|---|---|---|
| Swimmer | $1.50 \times 10^5$ | not solved | $5.85 \times 10^5$ | $\mathbf{1.11 \times 10^5}$ |
| Half Cheetah | $7.54 \times 10^6$ | not solved | $> 3.60 \times 10^8$ | $\mathbf{5.81 \times 10^6}$ |

**Results on non-linear policy learning on the Half Cheetah task.**  In Section 5.3, we compare ES gradient estimators on training linear policies on the Mujoco tasks. Here we compare these estimators' performance in training a non-linear ($d = 726$) policy network on the Mujoco Half Cheetah task under a fixed budget of total number of environment steps in Figure 13. Only our proposed method NRES solves the task under the computation budget.

**Ablation on the impact of $N$ (number of workers) on NRES's performance.**  We perform an ablation study on the impact of the number of NRES workers on its performance in solving the Mujoco Swimmer task in Figure 14. Here, increasing $N$ can help NRES use fewer sequential steps to solve the task but at a larger per sequential step compute cost.

**Ablation on the impact of $\sigma$ on NRES and FullES's performance.**  We perform an ablation study on the impact of the noise variance $\sigma^2$ on the performance of our proposed method NRES and FullES in solving the Mujoco Half Cheetah task in Figure 15. While setting $\sigma$ too small provides insufficient amount of smoothing and makes both methods fail to to solve the task, there still exists a range of larger $\sigma$ values under which both methods can solve the task successfully. For these cases, NRES always achieves a more than $50\times$ reduction in the number of sequential steps used over FullES.

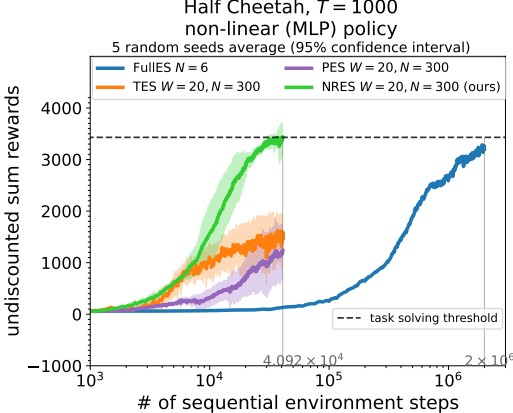

**Figure 13:** Comparing different ES methods' performance in training the nonlinear MLP policy architecture used in TRPO [10] on the Mujoco Half Cheetah task under a fixed number of total environment steps. Among all the ES methods, only NRES solves the task within the budget while also offering close to $50\times$ parallelization speed up compared to FullES.

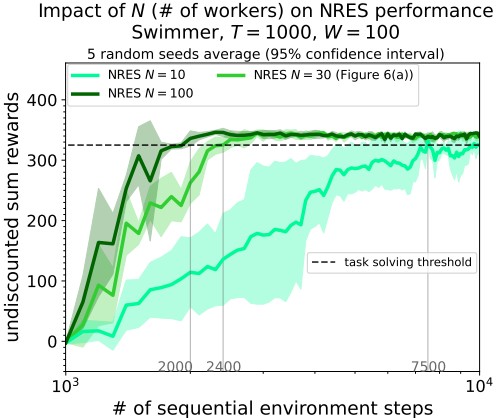

**Figure 14:** The impact of the number of parallel workers $N$ on NRES's performance on the Mujoco Swimmer task. We see that using more parallel workers can reduce the number of sequential environment steps needed to solve the task because the optimization benefits from a reduced gradient estimate's variance which scales with $1/N$.

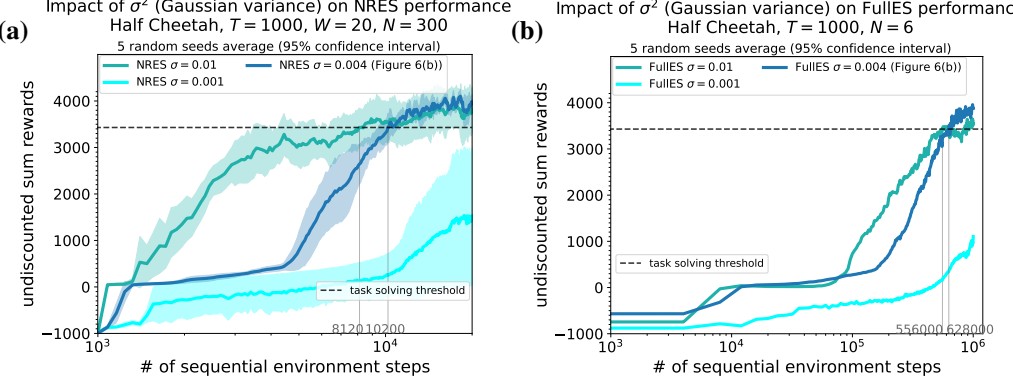

**Figure 15:** The impact of the hyperparameter $\sigma^2$ (smoothing Gaussian distribution's isotropic variance) on (a)NRES's and (b)FullES's performance on the Mujoco Half Cheetah task. Although insufficient amount of loss smoothing ($\sigma = 0.001$) could lead to slow convergence, there exists a range of hyperparameters ($\sigma = 0.004$ and $\sigma = 0.01$) that can allow both ES methods to solve the task. For both such cases, NRES improves significantly over FullES by more than $50\times$.

## E.2 Experiment details and hyperparameters

### E.2.1 Learning Lorenz dynamical system parameters

On the Lorenz system parameter learning task, we use the vanilla SGD optimizer to update the parameter $\theta = (\ln(r), \ln(a))$ starting at $\theta_{\text{init}} = (\ln(r_{\text{init}}), \ln(a_{\text{init}})) = (3.7, 3.116)$ with the episode length at $T = 2000$.

- For non-online methods, we use only $N = 1$ worker for BPTT to compute the non-smoothed true gradient since we only have one example sequence to learn from. For FullES, we use $N = 10$ workers.
- For all the online methods, we use $N = 200$ workers and a truncation window of size $W = 100$. This relationship exactly matches the condition considered in Theorem 9, and the total amount of computation for FullES and NRES to produce one gradient estimate is roughly the same.
- For all the ES methods, we use the smoothing standard deviation $\sigma = 0.04$ chosen by first tuning it on FullES.

For each gradient estimation methods, we tune its SGD constant learning rate from the following set

$$\{10^{-3}, 3 \times 10^{-4}, 10^{-4}, 3 \times 10^{-5}, 10^{-5}, 10^{-6}, 10^{-7}, 10^{-8}, 10^{-9}, 10^{-10}, 10^{-11}, 10^{-12}, 10^{-13}\}. \quad (110)$$

Here we choose the learning rate to ensure that **1)** there is no NaN in gradients/loss value due to the exploding gradient from the loss surface; **2)** the learning rate doesn't result in a significant increase in the loss. The learning rate range is made so wide because for all the unbiased AD methods, *these two issues would occur unless we use a trivially small learning rate*. Among the learning rates that pass these two requirements, we choose the best learning rate such that the optimization metric of interest decreases the fastest. For each method, the tuned learning rate used for Figure 5(a) is given in Table 4. As PES couldn't afford to use the same learning rate as NRES because of the unstable convergence (see Figure 5(a)), we hand-tuned a learning rate decay schedule for PES to maximally allow for its convergence.

**Table 4:** Learning rates (schedules) used for different gradient estimators on the Lorenz system parameter learning task

| method name | SGD learning rate (schedule) |
|---|---|
| BPTT $N = 1$ | $10^{-8}$ |
| TBPTT $N = 200, W = 100$ | $3 \times 10^{-4}$ |
| DODGE $N = 200, W = 100$ | $10^{-10}$ |
| UORO $N = 200, W = 100$ | $10^{-13}$ |
| FullES $N = 10$ | $3 \times 10^{-5}$ |
| TES $N = 200, W = 100$ | $3 \times 10^{-4}$ |
| PES $N = 200, W = 100$ | $10^{-5}$ for the first 1000 updates; $10^{-6}$ afterwards |
| NRES $N = 200, W = 100$ | $10^{-5}$ |

To generate Figure 5(a), we use a constant SGD learning rate of $10^{-5}$ to isolate the variance properties of different $\text{GPES}_K$ estimators.

### E.2.2 Meta-training learned optimizers

**Meta-training learned optimizer to train 3-layer MLP on Fashion MNIST.** The inner problem model is a three-layer GeLU-activated [45] MLP with 32 dimensions for each hidden layer. We choose GeLU instead of ReLU because GeLU is infinitely differentiable. By this design choice, the meta-training loss function is infinitely differentiable with respect to $\theta$, making it possible to consider automatic Differentiation methods for this task. We downsize the dataset Fashion MNIST [46] to 8 by 8 images to make both the unrolled computation graph small enough to fit onto a single GPU and also to make training fast. We split the Fashion MNIST training's first 80% image for inner problem training and the last 20% images for inner problem validation. During the inner training, we randomly sample a training batch of size 128 to compute the training gradient using the cross entropy loss. After updating the inner model with the learned optimizer, we evaluate the meta-loss (our optimization objective) using the cross entropy on a randomly sampled validation batch of size 128. We fix the data sequence in inner training and inner evaluation to focus on the optimization problem on a single unrolled computation graph. One inner problem training lasts $T = 1000$ steps of updates using the learned optimizer. We aim to minimize the average meta loss over $T = 1000$ steps.

- For offline methods, we only need $N = 1$ worker for BPTT to compute the gradient exactly. For FullES, we tuned its number of workers among the set $N \in \{1, 3, 10\}$ and choose $N = 10$.
- For all the online methods, we use the smallest possible truncation window size of $W = 1$ to test the online algorithms to the extreme. We use a total of $N = 100$ workers for all the online methods. Therefore, to produce one gradient update, our proposed online ES method NRES only runs $2 \cdot N \cdot W/T = 100 \cdot 1/1000 = 20\%$ of an episode in total.
- For all the ES methods, we use the smoothing standard deviation at $\sigma = 0.01$ inspired by the hyperparameter choice made in [3].

When meta-training the learned optimizer, we use Adam with the default parameters (other than a tuneable learning rate) to optimize the meta-training loss. For each gradient estimation method, we tune their Adam learning rate in the following range

$$\{3 \times 10^{-2}, 1 \times 10^{-2}, 3 \times 10^{-3}, 10^{-3}, 3 \times 10^{-4}, 10^{-4}, 3 \times 10^{-5}\} \tag{111}$$

After choosing a learning rate for a particular method over a specific random seed, we also check its performance on another seed to ensure its consistency (otherwise we lower the learning rate and try again). For each method, the tuned learning rate is given in Table 5.

**Table 5:** Tuned Learning rates used for different AD and ES gradient estimators on the meta-training learned optimizer task to train 3-layer MLP on Fashion MNIST for $T = 1000$ steps.

| method name | Adam learning rate (schedule) |
|---|---|
| BPTT $N = 1$ | $3 \times 10^{-4}$ |
| TBPTT $N = 100, W = 1$ | $10^{-2}$ |
| DODGE $N = 100, W = 1$ | $10^{-5}$ |
| UORO $N = 100, W = 1$ | $3 \times 10^{-5}$ |
| FullES $N = 10$ | $3 \times 10^{-2}$ |
| TES $N = 100, W = 1$ | $10^{-3}$ |
| PES $N = 100, W = 1$ | $3 \times 10^{-4}$ |
| GPES$_K$ $N = 100, W = 1, K = 4$ | $3 \times 10^{-4}$ |
| GPES$_K$ $N = 100, W = 1, K = 16$ | $3 \times 10^{-4}$ |
| GPES$_K$ $N = 100, W = 1, K = 64$ | $3 \times 10^{-4}$ |
| GPES$_K$ $N = 100, W = 1, K = 256$ | $3 \times 10^{-4}$ |
| NRES $N = 100, W = 1$ | $3 \times 10^{-4}$ |

**Meta-training a learned optimizer to train 4-layer CNN on CIFAR-10.** The inner problem model is a 4-layer ReLU-activated Fully Convolutional Neural Networks with 32 channels per layer produced by $(3, 3)$ filters with stride size of 1 and same padding. The convolutional layer is followed with a $(2, 2)$ average pooling layer with stride size of 2 and valid padding. After the 4-th convolutional block, we flatten the features and apply an affine transformation to predict the logits over the 10 classes in CIFAR-10. We split the CIFAR-10 [44] dataset with the first 80% images from its training set for inner problem training and the last 20% images for inner problem validation. During the inner training, we randomly sample a training batch of size 32 to compute the training gradient using the cross entropy loss. After updating the inner model with the learned optimizer, we evaluate the meta-loss (our objective) using cross entropy on a randomly sampled validation batch of size 32. Following [47], we apply thresholding to each step's meta loss to cap it at $1.5 \times \ln(10)$. Without this thresholding, the meta losses could become greater than $10^4$. This choice of thresholding introduces discontinuity to the loss function, making automatic differentiation inappropriate for this task. One inner problem training lasts $T = 1000$ steps of updates using the learned optimizer. We aim to minimize the average meta validation loss over $T = 1000$ steps. We fix the sequence of training and validation batches to isolate the problem to learning on a single unrolled computation graph instead of a distribution of UCGs. We compare among different ES methods on this task:

- For FullES, we set the number of workers $N = 1$ as we observe that it could already optimize the meta losses efficiently.
- For all the online ES methods, we use the truncation window size of $W = 100$. To match the total number of unroll steps used by NRES with FullES, we use $N = 10$ for all online ES methods (TES, PES, NRES).
- For all the ES methods, we use the smoothing standard deviation at $\sigma = 0.01$ inspired by the hyperparameter choice made in [3].

When meta-training the learned optimizer, we use Adam with the default parameters (other than a tuneable learning rate) to optimize the meta-training loss. For each gradient estimation method, we tune the learning rate in the following range

$$\{10^{-3}, 3 \times 10^{-4}, 10^{-4}, \}$$
(112)

For each ES method, the tuned learning rate is given in Table 6.

**Table 6:** Tuned Learning rates used for different ES gradient estimators on the meta-training learned optimizer task to train 4-layer CNN on CIFAR-10 for $T = 1000$ steps

| method name | Adam learning rate (schedule) |
|---|---|
| FullES $N = 1$ | $10^{-3}$ |
| TES $N = 10, W = 100$ | $10^{-3}$ |
| PES $N = 10, W = 100$ | $3 \times 10^{-4}$ |
| NRES $N = 10, W = 100$ | $3 \times 10^{-4}$ |

### E.2.3 Reinforcement Learning

We use the Mujoco tasks Swimmer-v4 and Half Cheetah-v4 from Open AI gym (gymnasium version) [48] with default settings. Here, following [21], we learn a deterministic linear policy that maps the observation space to the action space: for Swimmer-v4, this amounts to mapping an 8-dimensional observation space to a 2-dimensional action space ($d = 16$), while for Half Cheetah-v4, this amounts to mapping a 17-dimensional observation space to a 6-dimensional action space ($d = 102$). Because we can't differentiate through the dynamics, we only consider ES methods for this task.

Regarding the hyperparameters used for the Swimmer task,

- We use $N = 3$ FullES workers.
- For all the online ES methods, we fix the truncation window size at $W = 100$ and use $N = 30$ workers. This choice makes NRES and PES use the same number of total environment steps as FullES.
- We choose $\sigma = 0.3$ for all ES methods after tuning it for FullES.

For Swimmer-v4, we tune the learning rate used by the SGD algorithm in the following range:

$$\{10^2, 3 \times 10^1, 10^1, 3 \times 10^0, 10^0, \}$$
(113)

The tuned learning rates for each method on the Swimmer task are given in Table 7.

**Table 7:** Tuned learning rates used for different ES gradient estimators on the Mujoco Swimmer-v4 task

| method name | SGD learning rate |
|---|---|
| FullES $N = 3$ | $1 \times 10^0$ |
| TES $N = 30, W = 100$ | $3 \times 10^1$ |
| PES $N = 30, W = 100$ | $1 \times 10^0$ |
| NRES $N = 30, W = 100$ | $3 \times 10^0$ |

Regarding the hyperparameters used for the Half Cheetah task,

- We use $N = 6$ FullES workers.
- For the online ES methods, we fix the truncation window size at $W = 20$ and use $N = 300$ workers. This choice makes NRES and PES use the same number of total environment steps as FullES.
- We choose $\sigma = 0.004$ for all ES methods after tuning it for FullES.

For Half Cheetah-v4, we tune the learning rate used by the SGD algorithm in the following range:

$$\{10^{-4}, 3 \times 10^{-5}, 10^{-5}, 3 \times 10^{-6}\}$$
(114)

The tuned learning rates for each method on the Half Cheetah task are given in Table 8. We notice that PES's learning rate is tuned to be much smaller than the other methods because it suffers from high variance.

**Table 8:** Tuned learning rates used for different ES gradient estimators on the Mujoco Half Cheetah-v4 task

| method name | SGD learning rate |
|---|---|
| FullES $N = 6$ | $3 \times 10^{-5}$ |
| TES $N = 100, W = 100$ | $3 \times 10^{-5}$ |
| PES $N = 100, W = 100$ | $3 \times 10^{-6}$ |
| NRES $N = 100, W = 100$ | $3 \times 10^{-5}$ |

### E.3 Computation resources

For any gradient estimation method on each of the three applications, we can train on a GPU machine with a single Nvidia GeForce RTX 3090 GPU. As shown in the Figure 5 (Lorenz) and Figure 6(a) (learned optimizer) , NRES can converge in wall-clock time in less than $150$ and $2500$ seconds respectively. For the reinforcement learning mujoco tasks Swimmer and Half Cheetah, because the environment transitions are computed on CPUs in OpenAI Gym, we can run the entire experiment on CPU and it takes less than 2 hours and 10 hours respectively to finish training for all the ES methods. We believe these tasks are valuable experiment set ups to analyze new online evolution strategies methods in the future.

### E.4 Experiment implmentation

Our codebase is inspired by the high level logic in the codebase by [47]. We provide our code implementation in JAX [49] at https://github.com/OscarcarLi/Noise-Reuse-Evolution-Strategies.

## F  Broader Impacts and Limitations of NRES

As we have shown through multiple applications in Section 5, NRES can be an ideal choice of gradient estimator for unrolled computation graphs when the loss surface has extreme local sensitivity or is black-box/non-differentiable. In addition to being simple to implement, we show that NRES results in faster convergence than existing AD and ES methods in terms of wall-clock time and number of unroll steps across a variety of applications, including learning dynamical systems, meta-training learned optimizers, and reinforcement learning. These efficiency gains and lower resource requirements can in turn reduce the environmental impact and cost required to enable such applications.

However, just as any other useful tool in the modern machine learning toolbox, we note that NRES should only be used in appropriate scenarios. Here, we additionally discuss limitations of NRES (which are often limitations shared by a class of methods NRES belongs to) and in what scenarios it might not be the first choice to use.

**Limitations of** NRES **as an evolution strategies method.** Here we discuss the limitations of NRES as an ES method (both limitations below are common to all ES methods).

- **[Dependence on $d$]** As all the evolution strategies methods only have access to zeroth-order information, their variance has a linear dependence on the parameter dimension $d$ (see the first term in Equation (9) in Theorem 5). As a result, the most ideal use cases of ES methods (including NRES) are when the parameter dimension $d$ is lower than $100,000$. As we have seen in Section 5, many interesting applications would have parameter dimensions in this range. However, even in cases when $d$ is larger than this range, if the loss surface is non-differentiable or has high local sensitivity, one might still need to use NRES because AD methods might either be unable to compute the gradient or give noninformative gradients for effective optimization. On the other hand, AD methods (specifically Reverse Mode Differentiation methods BPTT and TBPTT) should be the first choice when the parameter dimension is large and the loss surface is differentiable and well-behaved, because their variances do not suffer from a linear dependence on $d$.

- **[Dependence on $\sigma$]** The variance $\sigma$ of the smoothing isotropic Gaussian distribution for $\epsilon$ is an extra hyperparameter ES methods introduce. In contrast, stochastic forward mode method DODGE doesn't have such a hyperparameter and can instead always sample from the standard Gaussian distribution (i.e. $\sigma = 1$). Assuming the loss surface is locally quadratic, differentiable, and well-behaved, DODGE (with standard Gaussian distribution) would have similar variance properties as NRES and is a preferable choice than NRES among stochastic gradient estimation methods due to its lack of hyperparameters to tune. On the other hand, as we have seen in Figure 9 and 10, when the loss surface are non-smooth and have many local fluctuations, DODGE would

perform much worse than NRES, because DODGE unbiasedly estimates the pathological loss surface's true gradient without smoothing.

**Limitations of** NRES **as an online method.** Next we discuss two limitations of NRES as an *online* gradient estimation method for unrolled computation graphs. The limitations are not specific to NRES but apply more broadly to classes of online methods.

- **[Hysteresis]** Any online gradient estimation methods (including TBPTT, DODGE, UORO, TES, PES, and NRES) would have bias due to hysteresis (see Section 2) because the parameter $\theta$ changes across adjacent partial online unrolls in the same UCG episode. Although we do not observe much impact of this bias in the applications and hyperparamter combinations considered in this paper, it is conceivable for some scenarios it could become an issue. In those cases, one should consider the tradeoffs of using NRES versus using FullES – NRES allows more frequent gradient updates than FullES while FullES avoids the bias from hysteresis.

- **[$T$'s potential dependence on $\theta$]** In this paper, we assume the unrolled computation graph is of length $T$, where $T$ doesn't depend on the value of the parameter $\theta$. However, for some UCGs, an episode's length might be conditioned on what value of $\theta$ is used. For example, consider a robotic control problem where the robot could physically break when some action is taken. In this case, a prespecified episode length might not be reached. In Mujoco, these types of termination are called unhealthy conditions. This is potentially problematic for antithetic online ES methods (including TES, PES, and NRES) when only one of the two antithetic trajectories terminate early but not the other. In this case, one heuristic would be to directly reset both particles, but by doing this, one would ignore this useful information about the difference in antithetic directions. Besides, the early termination of some workers might also make the distribution of workers' truncation windows not truly uniform over the entire episode's length $[T]$. How to gracefully handle these cases (through better heuristics/principled approaches) would be an interesting question of future work to make online antithetic ES methods (especially NRES) more applicable for these problems.