# OpenReview forum: "Variance-Reduced Gradient Estimation via Noise-Reuse in Online Evolution Strategies"
_NeurIPS.cc/2023/Conference — NeurIPS 2023 poster_

### Official Review · Reviewer_w7MW · 2023-06-28

**Soundness:** 3 good
**Presentation:** 2 fair
**Contribution:** 2 fair
**Rating:** 4
**Confidence:** 3

**Summary:**

The study considers numerically estimating gradients for online (reinforcement) learning.
The parameters are perturbed and from the performance of the perturbed models the gradient is estimated.
In an online setting, the proposed method decouples the number of steps between gradient estimates (which are then used by a first order gradient based optimiser) and the number of steps between changing the perturbation.
Two aspects are relevant in the paper: The variance of the gradient estimate and the ability to parallelise the methods.


**Strengths:**

I could not find anything that I would call wrong in the manuscript.



**Weaknesses:**

Allmost all research is to a certain degree incremental. The main problem of the research presented in the paper that is that the increment is rather small, it lacks originality and novel „surprising“ insights.

The problem setting is not new.
The proposed algorithm, as properly stated in the manuscript, can be viewed as a slight generalisation of „Persistent evolution strategies“ [13].
The canonical way to apply a Monte Carlo method such as ES to non-episodic RL tasks is to split the state-action-reward sequence into „pseudo-episodes“ . This is also done/suggested in other other non-ES RL algorithms, e..g., in 	Proximal Policy Optimization. This is also sometimes referred to as truncation.
The problem of how many (pseudo-)episodes are needed to evaluate a perturbation to get a reliable signal to be exploited by ES for RL has also been studied, for example see
Heidrich-Meisner and Igel. Hoeffding and Bernstein Races for Selecting Policies in Evolutionary Direct Policy Search. ICML 2009

Two aspects are relevant in the paper: The variance of the gradient estimate and the ability to parallelise the methods. The first one leads to a non-surprising result, which was simultaneously discovered in another work, properly cited by the authors: [19].
In my evaluation, the patalllization, while being useful in practice, is rather technicality.

The discussion the (parallel) runtime is partly not convincing.
I understand that
1. FullES takes O(T) time, it sees the states from s^+_0 to s^+_T (and s^-_0 to s^-_T)
2. calling Algorithm 2 takes O(W) time, it sees the states s^+_tau to s^+_tau+W (and s^-_tau to s^-_tau+W)
3. T/W independent calls of Algorithm 2 can be fully parallelised, so on T/W nodes taking the average over T/W independent calls starting from the same state s+ / s- also takes O(W) time
(Please correct me if I am wrong).

So I think „Under perfect parallelization, the entire  NRES gradient estimation would require O(W) time to complete. In contrast, the single FullES  gradient estimate has to traverse the UCG from start to finish, thus requiring O(T) time. Hence, NRES is T /W times more parallelizable than FullES with the same compute budget.“ is a bit misleading. Because a single FullES  gradient estimate sees all steps from 0 to T, while Algorithm 2 looks at several „parallel“ trajectories starting from the same state running for W steps. I think this is qualitatively different. In the RL setting, you cannot in general not replace   information collected over a time period with information collected from an ensemble where the ensemble members start from the same state. Think of an RL task where the rewards are always zero in the first 0.99*T steps.

The „Instead, we aim to compare the pure performance of different ES methods   assuming perfect parallel implementations. To do this, we measure a method’s performance as a  function of the number of sequential environment steps it used. Sequential environment steps are    steps that have to happen one after another (e.g., the environment steps within the same truncation window). However, steps that are parallelizable don’t count additionally in the sequential steps.“
This means that global random search, which evaluates 10^50 independent random policies has only a cost of one?
This would then be the ultimate baseline method, outperforming all proposed algorithms.

The manuscript lacks clarity. For example, some statements about „chaotic loss surfaces“ are confusing. Being chaotic is a well defined mathematical property.
It starts to make sense in the experiments. The Lorenz system is chaotic, which inmates the first place refers to perturbations of the system state.
The experiments always start from the same initial state. Optimised are the coefficients of the Lorenz system. What is the exact definition of a  „chaotic loss surfaces“? Where is the proof that the loss surface is really chaotic when changing r and a (citation missing).

A comparison with an evolution strategy that uses the perturbations also for the update, such as (1+1) CMA-ES with restricted covariance matrix) would have been interesting.

Hyperparameter tuning seems to be crucial. The description in the appendix is appreciated.
The Lorenz training suffers from instabilities. There are countermeasures, like gradient clipping.
One could also move to an optimiser that decouple gradient magnitude and update step length, for example Adam. Adam is used later, but for the
Lorenz experiments the manuscript sticks to SGD. How are the results with Adam (and corresponding learning rate tuning)?
This may be important because the experiments lead to to strong statements „AD methods perform worse than ES methods“ (Appendix) and one has to rule out that the problem is not the optimiser but really the AD as such.



Minor:

* Instead of talking about a trick and adding a reference, simply state that you use \nabla_theta E_x~p(x|theta) [ f(x) ] = E_x~p [ f(x)  \nabla_theta \ln p(x|theta), that is clearer (referring to this as a trick makes me cringe).

* The embedding in the existing literature is weak. One may not need no go back to Rechenberg and Schwefel when talking about evolution strategies (although it does not harm), but citing [12] as the only source for evolution strategies for RL is in my evaluation not OK.
* The first epsion in line 73should be in bold font.
* The hyperparameters are relevant, the main paper should point out to the appendix where the hyperparameters selection is discussed more clearly.



**Questions:**


* What is the exact definition of a  „chaotic loss surfaces“? Where is the proof that the Lorenz „loss surface“ is really chaotic when changing r and a?
* Under the used „sequential environment steps“ measure, would a  global random search algorithm that  which evaluates 10^50 independent random policies only have a cost of one? Would this be the ultimate baseline method, outperforming all proposed algorithms?




**Limitations:**

OK

---

> ### Author Rebuttal · Authors · 2023-08-09
>
> **Originality of contribution**.
> It is correct that NRES, while being simpler to implement than PES, is not a large algorithmic deviation from PES. However, the insight that reusing noise in truncation windows can both theoretically and empirically achieve significant variance reduction for unbiased online ES methods is a major and novel contribution of our paper. We want to emphasize that it is the combination of 1) noise reuse and 2) truncation that makes our contribution unique and novel. In contrast, the references the reviewer provides only consider either of these two ideas individually: on the one hand, it is true that the non-ES literature has explored the idea of truncation windows before, but such prior works have not considered the noise reuse aspect over multiple such windows. On the other hand, although the paper by Heidrich-Meisner and Igel evaluates a perturbation ($\theta$ plus noise) multiple times (similar to the idea of noise reuse), every such evaluation is an independent sample over the entire episode but not over a random truncation window. We believe the existence of these prior works on each concept alone does not diminish the contribution of our work. In addition, we note that we have provided a detailed discussion in Section B (line 507-541) of the various aspects our work differs and improves over the contemporaneous work of [19] in analyzing online ES algorithms.
>
> **Understanding the parallelization argument**.
> Unfortunately, the reviewer’s summary of how NRES is parallelized is incorrect. The reviewer assumes that all the parallel $T/W$ NRES workers would start unrolling from the same pair of antithetic states $s_+$ and $s_{-}$, thus covering the same truncation window. However, as shown in Figure 2(a), for an NRES update, each NRES worker would each maintain its own pair of antithetic states (see Algorithm 2) and would independently work at different truncation windows (we call these independent workers _step-unlocked_ (line 90-92)). By independence, different workers’ truncation windows should largely be non-overlapping and collectively cover almost all of the possible truncation windows in an episode. Thus the reviewer’s question of having no NRES worker cover the last $1\\%$ of an episode length for multiple updates would not be possible under our assumption and implementation.
>
> **Measuring the number of sequential steps of global random search**.
> In our RL Mujoco experiments, we have constrained the total number of environment steps per gradient update to be equal for all the ES methods (line 289). For example, on the Swimmer task, both FullES and NRES use a total of 6000 steps per update. This would have prevented the reviewer’s degenerate case of using $10^{50}$ of parallel workers in global random search at the cost of only a single episode worth of sequential steps $T=1000$. Besides, using $10^{50}$ episodes would also compare unfavorably in the total number of environment steps needed to solve a task, where we also show NRES is the best among all the ES methods.
>
> **What does _chaotic_ mean?** Mathematically, a function being chaotic means the function’s output is extremely sensitive to small changes in the function’s input. However, chaotic can be used to describe different functions in different contexts: when we say the Lorenz system is chaotic, the chaotic function is the dynamical system which maps from an initial state to a future state evolved under its dynamics. In contrast, _when we say the Lorenz system has a chaotic loss surface, the chaotic function is the loss function, which maps from the learnable parameter $\theta$ to its loss value_. Prior works on TES and PES have both used the term “chaotic” to describe the loss surfaces arising from unrolled computations. Thus we believe we use the term “chaotic” clearly and in a manner consistent with prior work. We have also shown that the Lorenz task’s loss surface is indeed chaotic through the visualizations in Figure 4(a) (left panel) and Figure 7. For example, in Figure 7(b), we see that the loss has extreme oscillations on the line segment connecting the initialization and ground truth parameters. If the reviewer believes there are other portions of our paper that lack clarity, please let us know.
>
> **Comparison against (1 + 1) CMA-ES with restricted covariance matrix**.
> Our paper focuses on online evolution strategies that use truncation windows for gradient estimation. We have compared against the offline method FullES because the online ES methods stem from it and prior works haven’t theoretically and experimentally analyzed the benefit of online ES over FullES. However, beyond FullES, we believe other offline ES methods (such as (1+1) CMA-ES) do not fit into the comparison scope of our current paper. We leave holistic comparisons of different classes of ES methods as future work.
>
> **Adam or gradient clipping cannot help automatic differentiation methods learn on the Lorenz task**.
> When the loss surface has extreme sensitivity and many suboptimal local minima, not only can the gradient’s magnitude become extremely large, but the gradient’s direction can also become non-informative. The alternative methods the reviewer proposed (gradient clipping or Adam optimizer) can potentially handle the cases of extreme gradient magnitude, but are not suitable to handle such non-informative gradient direction problems. Experimentally, we have tried using Adam with learning rates spanning over 9 different orders of magnitude ($10^{-1}$ to $10^{-9}$) and also gradient clipping over 3 orders of magnitude of thresholds. Under all of these optimizer hyperparameters, we are unable to make the de facto automatic differentiation methods backprop through time (BPTT) to reach a loss of lower than 300 (initialization loss is at 312) on the Lorenz task (where NRES and FullES can reach a loss lower than 50), thus proving the issue really lies in the AD gradient estimation methods but not in the chosen optimizers.

---

> > ### Comment · Reviewer_w7MW · 2023-08-16
> >
> > Coming back to my questions:
> >
> > Question 1: It is clear and well defined what a chaotic system is; it is clear what the Lorentz system is and for which values of its parameters it is a chaotic system.
> > The chaotic behaviour refers to the there state variables (often denoted x, y, z) of the system.
> > The objective/fitness function is a function of (some of the) the parameters (typically denoted by sigma, rho, and beta) of the Lorentz system.
> > How the properties of the Lorentz system carries over to the objective function is not clear to me.
> > Now, what defines a  „chaotic loss surfaces“?
> >
> >
> > Question 2: „Under the used „sequential environment steps“ measure, would a global random search algorithm that which evaluates 10^50 independent random policies only have a cost of one?“ So, the answer to this question is yes?

---

> > > ### Author Response · Authors · 2023-08-17
> > > **Further Explanation to the Reviewer's Questions**
> > >
> > > Thanks for your reply. We believe our rebuttal has answered these questions (see paragraphs on “__What does chaotic mean?__” and “__Measuring the number of sequential steps of global random search__”). However we discuss these questions in more detail below in case our previous response was unclear.
> > >
> > > __Answer to Question 1__: The reviewer asks how the chaotic property of the Lorenz system carries over to the objective function. We provide a detailed explanation in the paragraph below. However, before doing so, we note that this question is not the focus of our work: the aspect that is relevant to our experiments is that this dynamical system parameter learning problem indeed has _a loss objective function that exhibits extreme sensitivity (oscillations) with respect to small changes in its input learnable parameters_ (__this is what we refer to as a chaotic loss surface__). This terminology is consistent with its usage in prior works on online evolution strategies methods [1,2], and we have empirically shown the Lorenz loss objective function is chaotic as it has high degrees of fluctuation when small changes occur in the parameter space through Figure 4(a) and Figure 7.
> > >
> > > In terms of why the chaotic property of the Lorenz system carries over to the objective function, we summarize the discussion in [3], which has explained this relationship. Since the Lorenz system is a chaotic dynamical system, the state variable at a later time $s_t$ is highly sensitive to changes in the state variable at a prior time $s_i$, $i < t$. This means that the Jacobian matrix between the two states ($\frac{d s_t}{d s_i}$) has some large singular values. Because the total derivative of the loss function $L_{\textrm{avg}}$ with respect to the Lorenz system parameter $\theta$ is related to the Jacobian matrix through the following relationship (equation (7) and (8) in [3]):
> > >
> > > $\frac{dL_{\textrm{avg}}}{d\theta} = \frac{1}{T} \sum_{t=1}^T [\frac{\partial L_t}{\partial \theta} + \sum_{i=1}^t \frac{\partial L_t}{\partial s_t} {\color{blue} \frac{d s_t}{d s_i}} \frac{\partial s_i}{\partial \theta} ],$
> > >
> > > the magnitude of the loss gradient $\frac{dL_{\textrm{avg}}}{d\theta}$ will also be large due to the existence of the large singular values in the Jacobian matrices $\\{{\color{blue} \frac{d s_t}{d s_i}}\\}$. Having these high magnitude gradients implies that the loss values would have extreme sensitivity to small changes in the system parameters ($\theta$), thus resulting in the chaotic behavior of the loss function.
> > >
> > > [1] Metz, L., Maheswaranathan, N., Nixon, J., Freeman, D., & Sohl-Dickstein, J. Understanding and correcting pathologies in the training of learned optimizers. ICML, 2019.
> > >
> > > [2] Vicol, P., Metz, L., & Sohl-Dickstein, J. Unbiased gradient estimation in unrolled computation graphs with persistent evolution strategies. ICML, 2021.
> > >
> > > [3] Metz, L., Freeman, C. D., Schoenholz, S. S., & Kachman, T. Gradients are not all you need. arXiv:2111.05803, 2021.
> > >
> > >
> > > __Answer to Question 2__: The answer to this question is “no”, as the cost would be significantly larger for global random search in our empirical setup. To see this, we note that we have constrained all the methods in our experiments to use the same number of environment steps per gradient update (line 289) for fair comparison. If the reviewer insists on a comparison to global random search (which is not a gradient-based local update method, as we focus on in this work) in our experiment setup, to appropriately account for costs we should similarly consider __the cost of global random search as an iterative update method__: global random search keeps track of the best $\theta_*$ (with the lowest loss value) seen so far, and for each update, it randomly samples a number of parameters $\\{{\theta_i}\\}\_{i=1}^N$, evaluates their individual loss in parallel, and updates $\theta_*$ with the best $\theta_i$ if it has an even lower loss. From this perspective, we need to constrain global random search to also __use the same number of environment steps per update__ just as we have constrained the ES methods. For example, on the Swimmer task, because we make both FullES and NRES use only 6000 environment steps (a total of 6 episodes) per update, global random search will also be allowed to evaluate only $6$ different randomly drawn $\theta_i$ (instead of $10^{50}$) in parallel for an update (which would cost an episode-length ($T=1000$) number of sequential unrolls). Thus to evaluate $10^{50}$ episodes in total, global random search would require $10^{50} / 6 \approx 1.67 \times 10^{49}$ updates and a total of about $1.67 \times 10^{52}$ sequential steps. Global random search would therefore be expected to perform quite poorly against the methods considered in our paper.
> > >
> > >
> > > Thank you for engaging with us during the rebuttal period. Please let us know if you have any further questions, and in particular if there are any other points from your initial review that have not yet been addressed.

---

> > > > ### Comment · Reviewer_w7MW · 2023-08-18
> > > >
> > > > Question 2: If I am not overlooking something, either the paper is wrong or the rebuttal answer.
> > > > The paper states: „To do this, we measure a method’s performance as a function of the number of sequential environment steps it used. Sequential environment steps are steps that have to happen one after another (e.g., the environment steps within the same truncation window). However, steps that are parallelizable don’t count additionally in the sequential steps.“  Global random search can evaluate random search points in parallel - the do not depend on each other. Thus, a global random search algorithm that evaluates N independent random policies/search points only has a constant cost independent N of when under the suggested performance measure as defined in the submission (e.g., the answer to my original question should briefly be "yes").
> > > >
> > > > This is not just nitpicking. The performance measure is important for the evaluation of the different approaches. The performance measure was picked to stress a particular (and important) property of the proposed algorithm - which makes it look better compared to other methods that do not have this property. But if „sequential environment steps“ as defined in the manuscript is the relevant measure, then trivial methods seem perform at least on par with the suggest algorithm. I guess you need to introduce an additional constraint on the number of environmental steps per iteration. This would make a hyperparameter in the performance evaluation explicit - and methods will rank differently based on the choice.
> > > >
> > > > Question 1: The chaotic behaviour is stressed in the manuscript (the word occurs three times on the first page), therefore my question. The characterisation „a loss objective function that exhibits extreme sensitivity with respect to small changes in its input learnable parameter“ makes sense, but is not necessarily sufficient for the underlying system being chaotic in the strict mathematical sense. While I would suggest a different wording, I am happy with the explanation you gave.
> > > > (In your explanation in the reply, note that the Lorentz system is only chaotic for certain choices of parameters, in particular the „default“ parameters that are the optimum of the considered optimization task. This means that during optimization the behaviour of the underlying dynamical system need not be chaotic during the whole search process - but it will be around the optimum.)

---

> > > > > ### Author Response · Authors · 2023-08-18
> > > > >
> > > > > Thanks very much for your prompt response. We follow-up below:
> > > > >
> > > > > __Question 1__: Thank you for acknowledging our explanation. We originally chose the term “chaotic” following its usage in related works (TES, PES), but we agree with the reviewer that the word has different meanings depending on the context, and so we will further clarify its usage to describe loss functions in unrolled computation graph in our revision. We also agree that the underlying dynamical system doesn’t need to be chaotic to have a chaotic loss function on its induced unrolled computation graphs (nor do we claim this in the paper). For example, we do not claim that the dynamical system in the learned optimization application (where the state variables are the inner model parameters) is a chaotic dynamical system. However, as we have shown through Figure 1(a), the loss surface of this learned optimization task has extreme sensitivity to small changes in its learned optimizer’s parameters, and thus we refer to this loss surface as chaotic.
> > > > >
> > > > > __Question 2__: Similar to what the reviewer mentioned, _we have indeed introduced an additional constraint on the number of environmental steps per update_ when we measure the number of sequential steps for all the ES methods. __Thus there is nothing wrong or inconsistent with our paper/rebuttal response.__ For example, as we have explained on Line 289, FullES and NRES use the same number of steps ($6000$) on the Swimmer task. In our revision we will discuss this constraint earlier (Line 279-282), when we introduce the concept of _sequential steps_, to make the constraint more clear.
> > > > >
> > > > > The reviewer claims that under different values of this constraint, the methods will rank differently. Here we wish to make a few clarifications:
> > > > >
> > > > > 1. __How would global random search perform under different values of the constraint?__. It is true that when the limit approaches infinity (e.g. $\ge 10^{50}$ episodes per update), global random search can dominate any iterative update method that only explores locally (including the methods we consider but also other methods like CMA-ES) by using only one update and thus $T$ sequential steps. However, under practical values of this constraint (e.g., $\le 10^{3}$ episodes per update), _global random search is not a favorable method compared to the ES methods we have considered as it suffers from the curse of dimensionality_ — the total number of evaluations needs to be __exponential__ in the search space dimension $d$ to thoroughly evaluate the space, thus requiring a significantly larger number of updates and sequential steps than the ES methods need to solve the tasks. We are unaware of prior work that uses global random search for the tasks we have considered likely due to this issue.
> > > > >
> > > > > 2. __How do the methods we focus on compare under different values of the constraint?__. For the ES methods we consider in our paper (which do not include global random search), our proposed method _NRES would still outperform the other three methods_  __regardless of the size of this constraint__. This is because:
> > > > >
> > > > >     - NRES outperforms FullES because it uses $T/W$ times fewer sequential steps per update under any fixed constraint value due to its extra ability to parallelize over the time-step dimension (line 191-195).
> > > > >
> > > > >     - NRES outperforms PES because NRES gradient estimates will always have a lower variance (line 171-184) than PES estimates whenever both methods use the same number of workers, which would be true under the same number of environment steps per update.
> > > > >
> > > > >     - NRES outperforms TES because TES’s estimated gradients suffer from truncation bias (line 94-96), making it unable to converge to an optimal solution regardless of the number of steps used per update.
> > > > > Finally, we note that we have picked the current constraint values for each experiment (e.g. 6000 steps per update on Swimmer) by tuning on the method FullES (but not on our proposed method NRES). Hence this shouldn’t give NRES an unfair advantage in our experiments.
> > > > >
> > > > > __Summary__: Thanks again for engaging with us in this discussion and for your questions and feedback. To summarize:
> > > > >
> > > > > * We will make the definition “chaotic loss surface” more explicit in the paper when we describe __Loss properties__ in line 58.
> > > > >
> > > > > * We will highlight earlier on that when comparing iterative update methods in terms of sequential steps, we constrain the number of steps used per update by __adding the following sentences after line 282__: “As all the methods we compare are iterative update methods, we additionally require that each method use the same number of environment steps per update when measuring each method’s required number of sequential steps to solve a task.”
> > > > >
> > > > > * We emphasize that the _conclusion that NRES offers improvements over the other ES methods considered in our paper would still hold regardless of the number of steps allowed per update_. We will add the discussion/explanation provided above on this in our revision.

---

### Official Review · Reviewer_hYKW · 2023-07-01

**Soundness:** 3 good
**Presentation:** 2 fair
**Contribution:** 3 good
**Rating:** 5
**Confidence:** 3

**Summary:**

This paper generalizes PES based on noise-reuse, generating a more general class of unbiased online ES gradient estimators. The authors analytically characterize the variance of the estimators and identify the lowest-variance estimator named Noise-Reuse Evolution Strategies (NRES). Experiments on learning dynamical systems, meta-training learned optimizers, and reinforcement learning show that NRES results in faster convergence than existing AD and ES methods in terms of wall-clock time and number of unroll steps.

**Strengths:**

1. The paper proposes a general framework GPES by generalizing the existing PES algorithm based on noise-reuse.
2. The paper gives a theoretical analysis of unbiasedness and variance for GPES, and identifies the lowest-variance gradient estimator under this setting named NRES, which always reuses noise.
3. The paper proves that under some reasonable conditions, NRES has lower variance than FullES.


**Weaknesses:**

1. The proposed GPES is a simple and direct generalization of the existing PES. The novelty is not very strong.

2. The experiments need to be improved, e.g., adding the empirical comparison on longer sequences and higher dimension problems.

3. Why GPES can be better than PES? An intuitive explanation is needed.

Minor issue: Step-unlocked is used without explanation.


**Questions:**

1. In Theorem 8, it's said that the required assumption often holds in real-world applications. Can you give more explanation and evidences?

2. In the experiments, the x-axis of figures is measured by wall-clock time. Why not use the number of iterations?

**Limitations:**

Yes

---

> ### Author Rebuttal · Authors · 2023-08-09
>
> **Novelty of the contribution**. We agree with the reviewer that our proposed class of unbiased online ES gradient estimators GPES is a simple, intuitive generalization of PES. (In fact, we view the simplicity of the method as an important practical benefit.) However, _we respectfully disagree that our work lacks novelty_, as the main contributions of our paper aren’t only in introducing this class but more importantly in:
> 1. providing a theoretical characterization of the total variance of this class of estimators;
> 2. provably identifying the estimator NRES has the least total variance within this class;
> 3. empirically verifying the improvement of NRES over PES and other GPES estimators.
>
> **More experiments on longer sequences and higher-dimensional problems**.
> We believe our current experiments’ scale is already sufficient to prove NRES’s advantage over other ES baselines, and note that the scale is comparable to the scale considered in the most relevant prior work PES (which also uses $d \le 2000$ and $T$ around $1000$). However, to further demonstrate our algorithm NRES’s ability in handling higher-dimensional and longer sequence problems, we provide more experiments in the common response (Figure 1 in the attached pdf). To summarize the result, we increase the parameter dimension $d$ and sequence length $T$ each by (more than) $10\times$ on the learned optimizer task. For both cases, our proposed method still achieves significant wall-clock savings over all other ES methods.
>
> **Intuitive explanation of why GPES can be better than PES?**
> We do not claim that any GPES estimator is better than PES. Instead, we prove through Theorem 5 and Corollary 6 that NRES is better than other GPES estimators (including PES) due to its least amount of total variance. We thus assume the reviewer is asking for the intuition of _why NRES has lower variance than PES_ (please let us know if that’s not the case):
>
> Let’s consider size $W=1$ truncation window for simplicity. When PES and NRES unroll over the same time step $t$, both methods aim to estimate the **total derivative** of step $t$’s smoothed loss ($\widehat{L_t}$) with respect to $\theta$: $\frac{d\widehat{L_t}([\theta]\_{\times t})}{d\theta}$ (notice $\theta$ has been repeatedly applied $t$ times to compute this loss). This total derivative is equal to the sum of partial derivatives $\frac{d\widehat{L_t}}{d\theta} = \sum_{i=1}^t \frac{\partial \widehat{L_t}}{\partial \theta_i}$, where each term is with respect to the $i$-th time step’s application of $\theta$. By applying a new gaussian noise at every time step, PES can produce an unbiased estimate of each partial derivative. In contrast, because NRES uses the same gaussian noise for all time steps, it cannot individually estimate each partial but only their sum. However, to optimize $\theta$, we only need this sum (i.e. the total derivative) but not the individual terms. _The cost PES pays for having a separate yet unused estimation for each partial derivative is a larger variance than NRES_, since it needs more randomness to obtain this extra information. We will add this intuitive explanation to the paper as a remark.
>
> **Understanding why Theorem 8’s assumption often holds in real world**.
> To understand our claim that Theorem 8’s assumption often holds in the real world, we can reorganize our explanation in line 200-207 in three steps:
> 1. In many real world applications of unrolled computation graphs, if we update the parameter $\theta$ using the loss gradient from a specific truncation window, the losses in other truncation windows will also decrease. This is because there is a correlation between performing well at different time steps. For example, in a learned optimization task, the inner model parameters gradually improve during the training by a learned optimizer. Here, if the inner model over the last truncation window achieves lower validation losses (thus improved generalization), the earlier windows’ inner model snapshots will likely also generalize better and have lower validation losses.
> 2. Given Step 1’s observation, we can conclude that different truncation window losses’ total derivatives with respect to the same $\theta$ should largely point in the same direction. (Otherwise Step 1 would not be observed).
> 3. When different truncation windows’ gradients are pointing in similar directions, the assumption (Equation (10)) in Theorem 8 would hold true. To see this intuitively, consider the extreme case where each window’s gradient (the vector $\sum_{t=W(k-1) + 1}^{Wk} g^t$) lies exactly in the same direction (on the same line). In this case, Equation (10) is almost trivially satisfied.
>
> With the above three steps, we see that Theorem 8’s assumption would often hold true in the real world. We have also provided empirical verification of Theorem 8’s conclusion under this assumption in Figure 3(b) in the main paper, which provides further evidence that such assumptions would hold in real applications.
>
> **Why use wall-clock time instead of number of iterations?**
> Since our work focuses on parallel ES methods, we choose measures of computation that can allow us to compare the parallelizability of these algorithms in terms of actual/theoretical time they take to run. Number of iterations is not such a measure, because _different algorithms could require significantly different amounts of time to run while still using the same number of update iterations and total computation_. For example, given the same computation budget of $2T$ unrolls per update, FullES can only afford to run 1 worker from start to finish, thus taking $T$ units of time per update. In contrast, NRES can simultaneously run $T/W$ independent workers each over a truncation window of length $W$, in total taking only $W$ units of time because of the parallelization. In this case, NRES can achieve a $T/W\times$ time speed up over FullES even if they use the number of iterations to converge (see line 191-195).

---

> > ### Comment · Area_Chair_PTqC · 2023-08-16
> > **Follow up hYKW**
> >
> > Dear Reviewer hYKW,
> >
> > We would appreciate if you would you be so kind as to acknowledge and respond to the authors' rebuttal. This is crucial to ensure the reviewing process is conducted adequately.
> >
> > AC

---

> > ### Comment · Reviewer_hYKW · 2023-08-19
> > **Thanks for your detailed response.**
> >
> > Most of my concerns have been addressed. I will keep my evaluation.

---

### Official Review · Reviewer_FJdn · 2023-07-04

**Soundness:** 4 excellent
**Presentation:** 4 excellent
**Contribution:** 3 good
**Rating:** 7
**Confidence:** 2

**Summary:**

This work proposes a method for optimizing unrolled computation graphs (e.g. recurrent networks, etc.). When using ES to optimize a computation graph, the graph must be fully rolled out. Recent methods (PES) have examined using a truncated window to optimize, so that optimization can occur without a full unroll. PES aims to unbias the estimator by accumulating truncation noise during an online rollout, however, it produces high variance.

This paper presents a generalization to PES, GPES, that decouples the frequency of noise injection and gradient estimation. A specific instantiation, named NRES, samples the noise only once per episode. This reduces variance as it removes the need for noise accumulation.

Experiments are presented on a two-parameter chaotic Lorenz system, where NRES outperforms previous work. An RNN-based meta-learning setup is also evaluated wherein the transition function defines an update over inner parameters, along with a reinforcement learning setup to solve Mujoco tasks.

**Strengths:**

This paper provides an extremely thorough investigation of their proposed framework. The framework includes a generalization of previous work, along with a theoretically-justified instantiation with desirable properties on unbiased estimation and low variance. This is an original idea that is promising. The quality and clarity of the writing is excellent. The work sets up for additional work in investigating the properties of GPES methods, including variations on the proposed NRES.

**Weaknesses:**

The final experiments on meta-learning and reinforcement learning are lacking detail. A more precise experimental setup would strengthen the argument of the paper. In addition, comparisons to non-ES based methods would give more context on how such methods compare to other solutions in the field. The experiments present wall-clock time as the unified axis, but discussion in the paper mentions total computation budget -- a figuring comparing the methods using some consistent measure of computation would strengthen the paper.

**Questions:**

How does NRES compare to FullES in terms of pure computational budget? What are the scaling patterns of additional parallelization?
How do non-ES methods perform on the Mujoco tasks?

Additionally, ablations on hyperparameters such as the noise variance and # of workers would grant insight into the nuance of the method.


**Limitations:**

Limitations are discussed in section 7 -- they largely relate to ES in general.

---

> ### Author Rebuttal · Authors · 2023-08-09
>
> We thank the reviewer for recognizing the quality and clarity of our writing and the originality of our idea.
>
> **Experiment details**.
> We have provided detailed descriptions of the experimental set ups and how we have tuned the hyperparameters for each experiment in Appendix E. We will make this more obvious in the main paper.
>
> **Comparison against non-ES based methods**.
> The primary non-ES methods we compare against are automatic differentiation gradient estimation methods. As we have explained in line 58-63, AD methods struggle to be useful in problems with chaotic loss surfaces. For these problems, we have already compared against $4$ AD methods on the Lorenz task and the learned optimizer task in our experiments. Because AD methods all perform worse than ES methods (line 241 and 261), we show their results in Figure 8 and 9 in the Appendix.
>
> **Measure of computation**.
> There are naturally two ways to measure the amount of computation an optimization algorithm needs to reach a given performance level on unrolled computation graphs:
> * **Measure 1**:  the total number of unrolls, which characterizes the total amount of work (energy) used by the algorithm. In reinforcement learning, this measure is also treated as the empirical sample complexity.
> * **Measure 2**: the number of sequential unrolls, which is linearly proportional to the theoretical wall clock time a parallel algorithm needs assuming good implementation and computing hardware.
>
> Since we focus on parallel ES algorithms, our primary measure of computation is through Measure 2 (we use Measure 2 for the RL experiments in the paper). Empirically, we have also compared ES algorithms by the actual wall-clock time and calculated how many times NRES improves over other methods. This actual wall-clock improvement is a conservative estimate of NRES’s theoretical wall-clock improvement captured by Measure 2 because we only use a single GPU card (which has a limited amount of parallelization ability). However, the actual wall-clock speed up should approach the theoretical speed up (Measure 2) as we scale up on the amount of compute hardware.
>
> **Comparing NRES and FullES in terms of pure computation budget**. Despite our focus on Measure 2, we also observe an improvement in Measure 1 (total number of unrolls) because of the variance reduction benefit of NRES over FullES (Theorem 8). We have provided the statistics of both measures for the RL experiments in Table 3 in the Appendix. In the Table below, we redisplay these results for RL and additionally provide these measures for the meta-training learned optimizer task in Figure 5(a). Here we see that NRES is not only significantly more parallelizable than FullES (Measure 2), but also reduces the total amount of compute/sample (Measure 1).
>
> || reach 0.61 loss in learned optimizer||solve Mujoco Swimmer||solve Mujoco Half Cheetah||
> |-|-|-|-|-|-|-|
> ||# of sequential unrolls (Measure 1)|# of total unrolls (Measure 2)|# of sequential unrolls|# of total unrolls|# of sequential unrolls|# of total unrolls|
> | FullES  | $2.46 \times 10^6$  | $4.92 \times 10^7$ | $2.50 \times 10^4$ | $1.50 \times 10^5$  |  $6.28 \times 10^5$  | $7.54 \times 10^6$  |
> | NRES | $\mathbf{1.95 \times 10^4}$  | $\mathbf{4.00 \times 10^6}$  |  $\mathbf{2.40 \times 10^3}$ |  $\mathbf{1.11 \times 10^5}$ |  $\mathbf{1.02 \times 10^4}$  | $\mathbf{5.81 \times 10^6}$ |
>
> **Non-ES methods on the Mujoco task**.
> As the Mujoco task’s transition dynamics do not support automatic differentiation, the most relevant methods are policy gradient methods, which require having stochastic policy and apply likelihood ratio (LR) gradient estimation in the action space. (In contrast, ES applies LR in the parameter ($\theta$) space and can handle deterministic policy). We compare the reported results by Rajeswaran et al which have used natural policy gradient to train linear policies on the two Mujoco tasks we have considered in this paper:
> ||Number of environment steps to solve||
> |-|-|-|
> ||Swimmer|Half Cheetah|
> |(Rajeswaran et al) + stochastic linear policy|$1.45 \times 10^6$|$1.13 \times 10^7$|
> |NRES (ours) + deterministic linear policy|$\mathbf{1.11 \times 10^5}$|$\mathbf{5.81 \times 10^6}$|
>
> As we can see, NRES improves over this policy gradient method in solving these Mujoco tasks.
>
> Rajeswaran, A., Lowrey, K., Todorov, E. V., & Kakade, S. M. (2017). Towards generalization and simplicity in continuous control. Advances in Neural Information Processing Systems, 30.
>
>
> **Ablation on the noise variance $\sigma$**.
> As we have described in line 946, all ES methods require tuning the hyperparameter $\sigma$ which controls the variance of the smoothing distribution: on the one hand, setting $\sigma$ too small might not provide sufficient loss smoothing, making the optimization difficult to converge. On the other hand, setting $\sigma$ too large might shift the global optimum of the smoothed surface away from the true minimum. Following the reviewer’s suggestion for an ablation, we provide experiments on the impact of $\sigma$ on both NRES and FullES’s performance on the Mujoco Half Cheetah task in Figure 2 in the common response pdf (where we have tuned the learning rate for each value of $\sigma$ and each method separately). Here we see that although insufficient amount of loss smoothing ($\sigma = 0.001$) could lead to slow convergence, there exists a range of hyperparameters ($\sigma=0.004$ and $\sigma=0.01$) that can allow both ES methods to solve the task. For both cases, NRES improves significantly over FullES.
>
> **Ablation on the number of workers $N$**.
> As the NRES workers are independent, the variance of the average of their gradient estimates scales with $1/N$. We provide an experiment on the impact of $N$ on NRES’s performance on the Mujoco Swimmer task in Figure 3(a) in the common response pdf. We see that having more workers can reduce the number of sequential steps to converge but also require a greater cost per sequential step.

---

> > ### Comment · Reviewer_FJdn · 2023-08-13
> > **Acknowledgement of Rebuttal**
> >
> > Thank you for the detailed response and the additional experimental results. It would be great to include these results in a revised version of the paper as well, for future reference. Given that the score already indicates accept, I will maintain this score.

---

> > > ### Author Response · Authors · 2023-08-14
> > > **Response to Reviewer's Acknowledgement**
> > >
> > > Thanks for your prompt reply! We will make sure to incorporate these added results in our revision.
> > >
> > > If you believe we have answered your questions to the degree that you'd feel comfortable raising your _confidence_ score, we would really appreciate it, but in any case we want to express our gratitude again for your positive feedback and suggestions to improve our work.

---

### Official Review · Reviewer_77pb · 2023-07-05

**Soundness:** 3 good
**Presentation:** 3 good
**Contribution:** 4 excellent
**Rating:** 7
**Confidence:** 3

**Summary:**

In this paper, the author(s) extended the well-known Persistent Evolution Strategies via noise-reuse and proposed an improved version with reduced variance. The main contribution of this paper is to provide detailed mathematical proof to validate their claim.

**Strengths:**

Considering this significant contribution to the evolution strategies and other related ML communities, I personally suggest accepting this high-quality research paper.

In this paper, the author(s) also discussed the main limits of their method. These discussions about the possible limitations (hysteresis and complexity) are highly encouraged for academic research because this can better reflect the whole state of their method.

Furthermore, the author(s) pointed out that “whether there are better ways to leverage the sequential structure” beyond the isotropic Gaussian distribution. I agree that this is an interesting open question (worth to be investigated).

**Weaknesses:**

Some notes are given in order to further improve this paper:
For Section 5.3 Reinforcement Learning, the removal of “additional heuristic tricks” can help us focus on the underlying mechanism, which is very critical for algorithmic understanding, no matter from a practical or theoretical perspective. We often prefer general-purpose design principles, though these “additional heuristic tricks” sometimes work well in some cases. Given that only the linear policy was used in experiments, it is highly expected to include also the more challenging non-linear policy with higher dimensions.

**Questions:**

see above

**Limitations:**

No comments here.

---

> ### Author Rebuttal · Authors · 2023-08-09
>
> We thank the reviewer for appreciating the significance of our contribution.
>
> **Non-linear RL policy**. We initially experimented with linear policies on the Mujoco tasks because
> 1. it has been observed by Rajeswaran et al that using linear policies can yield performances comparable to state-of-the-art results on continuous control (Mujoco) tasks;
> 2. we want to keep our experiments consistent with prior works on ES (Mania et al, Vicol et al), which also experiment with linear policies.
>
> However, following your suggestion, we have conducted an additional experiment where we train a multi-layer perceptron policy used by Schulman et al on the Half Cheetah task, which thereby increases the total number of parameters by $6.5\times$. We tune each ES method’s SGD learning rate individually and report the result in Figure 3(b) in the attached pdf in the common response. Here we constrain all the methods to use the same number of total environment steps. We notice that among the four methods, only NRES has successfully solved the task. For the rest of the methods, although FullES comes close to solving the task, it requires $50\times$ more theoretical wall clock time than NRES. Besides, both PES and TES are far away from solving the task under the same budget. This result demonstrates NRES’s ability to learn non-linear, higher-dimensional policy for reinforcement learning and also provides additional evidence of NRES’s significant advantage over other ES methods. Thanks for the suggestion to explore this direction.
>
> Mania, H., Guy, A., & Recht, B. (2018). Simple random search provides a competitive approach to reinforcement learning. Advances in Neural Information Processing Systems, 31.
>
> Rajeswaran, A., Lowrey, K., Todorov, E. V., & Kakade, S. M. (2017). Towards generalization and simplicity in continuous control. Advances in Neural Information Processing Systems, 30.
>
>
> Schulman, J., Levine, S., Abbeel, P., Jordan, M., & Moritz, P. (2015, June). Trust region policy optimization. In International conference on machine learning (pp. 1889-1897). PMLR.
>
> Vicol, P., Metz, L., & Sohl-Dickstein, J. (2021, July). Unbiased gradient estimation in unrolled computation graphs with persistent evolution strategies. In International Conference on Machine Learning (pp. 10553-10563). PMLR.

---

### Official Review · Reviewer_Wtuw · 2023-07-26

**Soundness:** 3 good
**Presentation:** 3 good
**Contribution:** 3 good
**Rating:** 6
**Confidence:** 3

**Summary:**

This paper studies online evaluation strategies for unrolled computation graphs. Especially, the authors 1). propose a general class of unbiased online evolution strategies that generalizes Persistent Evolution Strategies (PES), named Generalized Persistent Evolution Strategies (GPES). The key idea is to share noise across truncation windows instead of sampling every round. 2). characterize the variance and variance reduction properties of their strategies. 3). study a special case of the general class strategies, named Noise Reuse Evolution Strategie (NRES), show the variance advantage of NRES over other estimators. 4). experimentally show the advantages of NRES across a variety of applications, including learning dynamical systems, meta-training learned optimizers, and reinforcement learning.

**Strengths:**

- The idea of reusing noise to reduce variance is interesting and a bit counterintuitive. For first order method, reusing noise always lead to a larger variance (If the noise is reused, averaging gradients cannot reduce the variance of the noise). However, it seems for evolution strategy, such a simple method could significantly reduce variance and improve the performance.
- Experimental results are comparable to the previous methods and show a remarkable improvement.
- The paper is well writen. The proof in the appendix is well organized and seems correct.

**Weaknesses:**

- No comparision between NRES and first order method.
- The authors did not explain the some parameter setting in the experiments. Especially, in Figure 5 and 6, why choose a different N when comparing online ES and FullES?
- For online ES, the algorithm update $\theta$ every truncation window, thus the loss will never be in the form of (7). In this case, it seems NRES is still biased.



**Questions:**

Given a time $T$, FullES returns one gradient g=[L([\theta+\epsilon]_{\times T})-L([\theta-\epsilon]_{\times T})]{\epsilon}/{2\sigma^2}, and NRES (Algorithm 2) returns $T/W$ gradients $g_1,\dots, g_{T/W}$, where g_k=[\sum_{t=Wk+1}^{Wk+W}L_t([\theta+\epsilon]_{\times t})-L([\theta-\epsilon]_{\times t})]{\epsilon}/{2\sigma^2 W}. By definition, there is $g=W/T \sum_{k=1}^{T/W} g_k$. In this case, if we choose learning rate $\eta_{FullES} = \eta_{NRES}T/W$, there should be no difference between FullES and NRES. So why is there such a big improvement in the experiments?

One minor typo: Algorithm 6 line 10: not $L_t^s$, should be $L_{self_.\tau +1}^s$.


**Limitations:**

Theoretical work. No limitations.

---

> ### Author Rebuttal · Authors · 2023-08-09
>
> **Comparison between NRES and first order methods**. We have provided discussions of automatic differentiation (first order) methods in Section B Additional Related Work in the Appendix. Besides, we have provided empirical comparisons between NRES and 4 different first order AD methods on the Lorenz task and the learned optimizer task in our paper. As these first order methods all perform worse than ES methods (line 241 and 261), we have deferred their results to Figure 8 and 9 in the Appendix.
>
> **How $N$ is chosen for the experiments**. As we have described on line 188-190, because a FullES worker’s gradient estimate uses $2T$ steps while an NRES worker uses $2W$ steps, we by default maintain a ratio of $T/W$ between the number of NRES and FullES workers to keep the per-update number of unrolls constant. This is how we choose N for online ES and FullES on the Lorenz task (Figure 4(b)), the Swimmer task (Figure 6(a)), and the Half Cheetah task (Figure 6(b)). For the learned optimizer task (Figure 5(a)), we empirically find that we can use significantly fewer online ES workers to achieve good performance  –  we only need $200$ total unrolls ($100$ NRES workers each antithetically unrolling for $1$ steps) per meta-gradient update. Because $200$ total unrolls is shorter than the minimum number of unrolls FullES needs per update (at least $2\times T = 2000$ unrolls), we have to relax the default number of worker ratio and allow FullES to use more workers (we choose $N=10$ after tuning it among $\\{1, 3, 10\\}$ (line 840-842).
>
> **Bias in NRES**.
> As we have discussed in the paragraph on _hysteresis_ (line 105-113), it is indeed correct that, as we update $\theta$ every truncation window, the loss in the current window has some historical dependence on past values of $\theta$ (hysteresis), making NRES gradient biased in practice. However, it is worth noting that all the prior works on online gradient estimation methods (including TES, PES and first order methods like UORO and DODGE) also have bias from hysteresis. Although we didn’t observe much impact of hysteresis in our experiments, we believe understanding and correcting hysteresis is an interesting direction of future work as we have discussed in line 323-325.
>
> **Understanding the improvement of NRES over FullES**.
> Given the same computation budget of $2T$ unrolls, we can use it either to produce one FullES gradient estimate or to produce $T/W$ i.i.d. NRES gradient estimates. The reviewer’s current description assumes the same NRES worker sequentially produces these $T/W$ gradient estimates – instead, these $T/W$ NRES gradient estimates are produced by independent workers and thus can happen simultaneously. In contrast, the single FullES worker has to unroll sequentially for $T$ steps from start to finish. Thus NRES can complete the same amount of total unrolls with $T/W \times$ less time than FullES (see line 191-195). This ability to achieve extra parallelization is the primary reason we see such a big improvement of NRES over FullES in the experiments.

---

> > ### Comment · Reviewer_Wtuw · 2023-08-16
> >
> > Thank you for the detailed response. It seems that the biggest improvement of NRES over previous works is the parallelization ability.
> >
> > Given the score indicates accept, I meantain this score.

---

> > > ### Author Response · Authors · 2023-08-17
> > >
> > > Thanks very much for your reply and your recommendation to accept the paper.
> > >
> > > Following the reviewer’s remark about the reason for the improvement of NRES, we briefly summarize the main benefits of NRES over prior works (also illustrated in the table in Figure 1(b)). NRES improves over prior ES methods for a number of reasons:
> > >
> > > 1. As the reviewer mentions, _NRES improves over FullES_ because NRES is an online method unlike the offline algorithm FullES and it thus has much __better parallelization__ ability than FullES. The gradient estimate from NRES can also have lower variance than FullES under the same total compute.
> > >
> > > 2. _NRES improves over the online method TES_ because TES suffers from truncation bias whereas NRES is __unbiased__.
> > >
> > > 3. _NRES improves over the online method PES_ because NRES has a significantly __lower variance__ than PES due to its noise-reuse property.
> > >
> > > These properties together make NRES a particularly compelling approach relative to prior ES methods.

---

### Author Rebuttal · Authors · 2023-08-10

We want to thank all the reviewers for your reviews and comments. We address each reviewer’s questions and feedback in an individual response. For new plots created for the rebuttal, we have included them in the uploaded pdf in this common response. Below is a summary of the new plots in the pdf:
* **Figure 1(a) (Reviewer hYKW)**: We compare ES gradient estimators on an $11 \times$ higher-dimensional ($d=19330$) learned optimizer task than that considered in Figure 5(a). To increase the parameter dimension, we increase the width of the multilayer perceptron used by the learned optimizer. For this higher-dimensional problem, NRES can still provide a $3.8\times$ wall clock time speed up over PES and a $9.7 \times$ wall clock time speed up over FullES.
* **Figure 1(b) (Reviewer hYKW)**: We compare ES gradient estimators on the learned optimizer task with a $10\times$ longer sequence length ($T=10000$) than that considered in Figure 5(a). For this problem, NRES still achieves a $2.1 \times$ wall clock speed up over PES and a $3.5 \times$ speed up over FullES.
* **Figure 2 (Reviewer FJdn)**: We perform an ablation study on the impact of the noise variance $\sigma^2$ on the performance of FullES and our proposed method NRES in solving the Mujoco Half Cheetah task. While setting $\sigma$ too small makes both methods fail to to solve the task, there still exists a range of larger $\sigma$ values under which both methods can solve the task successfully. For these cases, NRES always achieves a more than $50\times$ reduction in the number of sequential steps used over FullES.
* **Figure 3(a) (Reviewer FJdn)**: We perform an ablation study on the impact of the number of workers $N$ on the performance of NRES in solving the Mujoco Swimmer task. Here, increasing $N$ can help NRES use fewer sequential steps to solve the task but at a larger per sequential step compute cost.
* **Figure 3(b) (Reviewer 77pb)**: We compare ES gradient estimators on training a non-linear, ($6.5\times$) higher-dimensional policy network on the Mujoco Half Cheetah task under a fixed budget of total number of environment steps. Only our proposed method NRES has solved the task.

---

### Decision · Program_Chairs · 2023-09-21

**Decision:**

Accept (poster)

**Comment:**

This work introduces an evolution strategies method for unrolled computation graphs. In these problems usually the graph needs to be fully rolled out. Existing methods get around this problem by considering truncated optimization windows to optimize. This method instead (PES) introduces a way to compute an unbiased estimator by accumulating truncation noise during an online rollout. The new method is evaluated on a couple of experimental benchmarks. The reviewers agreed this is a meaningful contribution to the ES literature and thus I recommend its acceptance to the conference.